# Action initiation and punishment learning differ from childhood to adolescence while reward learning remains stable

Ruth Pauli [1] ✉, Inti A. Brazil[2], Gregor Kohls[3,4], Miriam C. Klein-Flügge [5,6], Jack C. Rogers[1,7], Dimitris Dikeos[8], Roberta Dochnal[9], Graeme Fairchild[10], Aranzazu Fernández-Rivas[11], Beate Herpertz-Dahlmann[3], Amaia Hervas[12], Kerstin Konrad[3,13], Arne Popma[14], Christina Stadler[15], Christine M. Freitag [16], Stephane A. De Brito[1,7] & Patricia L. Lockwood [1,5,6,7] ✉

Theoretical and empirical accounts suggest that adolescence is associated with heightened reward learning and impulsivity. Experimental tasks and computational models that can dissociate reward learning from the tendency to initiate actions impulsively (action initiation bias) are thus critical to characterise the mechanisms that drive developmental differences. However, existing work has rarely quantified both learning ability and action initiation, or it has relied on small samples. Here, using computational modelling of a learning task collected from a large sample ($N = 742$, 9-18 years, 11 countries), we test differences in reward and punishment learning and action initiation from childhood to adolescence. Computational modelling reveals that whilst punishment learning rates increase with age, reward learning remains stable. In parallel, action initiation biases decrease with age. Results are similar when considering pubertal stage instead of chronological age. We conclude that heightened reward responsivity in adolescence can reflect differences in action initiation rather than enhanced reward learning.

Adolescence is a time of great change, as young people navigate their way from the dependency of childhood to the independence of adulthood. Theoretical accounts suggest it is a period of risky, impulsive, and reward-seeking behaviour, which is hypothesised to reflect neurobiological changes that lead to heightened reward learning[1–5]. Adolescence is also a high-risk period for the onset of mental disorders[6], including disruptive behaviour disorders[7], which are strongly associated with impulsive behaviour and difficulties with reinforcement learning[8]. Internalising problems are likewise associated with difficulties in reinforcement learning[9,10], and social media

[1]Centre for Human Brain Health, School of Psychology, University of Birmingham, Birmingham, UK. [2]Radboud University, Donders Institute for Brain, Cognition and Behaviour, Nijmegen, The Netherlands. [3]Child Neuropsychology Section, Department of Child and Adolescent Psychiatry, Psychosomatics and Psychotherapy, RWTH Aachen University, Aachen, Germany. [4]Department of Child and Adolescent Psychiatry, Faculty of Medicine, TU, Dresden, Germany. [5]Department of Experimental Psychology, University of Oxford, Oxford, UK. [6]Wellcome Centre for Integrative Neuroimaging, University of Oxford, Oxford, UK. [7]Institute for Mental Health, School of Psychology, University of Birmingham, Birmingham, UK. [8]Department of Psychiatry, Medical School, National and Kapodistrian University of Athens, Athens, Greece. [9]Faculty of Medicine, Child and Adolescent Psychiatry, Department of the Child Health Center, Szeged University, Szeged, Hungary. [10]Department of Psychology, University of Bath, Bath, UK. [11]Basurto University Hospital, Bilbao, Spain. [12]University Hospital Mutua Terrassa, Barcelona, Spain. [13]JARA-Brain Institute II, Molecular Neuroscience and Neuroimaging, RWTH Aachen and Research Centre Jülich, Jülich, Germany. [14]Department of Child and Adolescent Psychiatry, VU University Medical Center, Amsterdam, Netherlands. [15]Department of Child and Adolescent Psychiatry, Psychiatric University Hospital, University of Basel, Basel, Switzerland. [16]Department of Child and Adolescent Psychiatry, Psychosomatics and Psychotherapy, University Hospital Frankfurt, Goethe University, Frankfurt am Main, Germany. ✉e-mail: r.pauli@bham.ac.uk; p.l.lockwood@bham.ac.uk

use (which can become problematic for some adolescents[11]) has recently been linked to reward learning mechanisms[12]. However, reward- and punishment-guided behaviour in adolescence is not yet fully understood. This is because distinct psychological processes can manifest in similar overt behaviour, and traditional data analysis techniques are usually not well suited to capturing these covert processes[13]. A myriad of terms have been developed to describe closely related concepts, such as reward learning, risk-taking, and impulsivity[14], which, though they might reflect similar behaviour, likely point to distinct psychological processes. Furthermore, these concepts are typically operationalised using questionnaires or summary performance measures from behavioural tasks, which cannot capture the temporally dynamic nature of learning processes[13]. In consequence, to understand differences in behaviour across adolescence, it is important to distinguish between learning processes and other mechanisms that might manifest in similar behaviour, such as response biases. Here, we use computational modelling to distinguish between learning processes (modifying future behaviour based on past experience of reward and punishment) and action initiation or 'go' biases (initiating actions impulsively, without regard for consequences). We test whether these different mechanisms are separable, and to what extent they exhibit normative developmental differences across late childhood and adolescence in a large sample collected from multiple countries.

Computational modelling of learning can be conducted using reinforcement learning models, which assume that actions and their outcomes become associated through experience, and the learned value of an action then influences the likelihood of repeating that action in the future[15]. There has been a relative paucity of computational modelling work focusing on learning in adolescence compared to the adult literature, and previous studies were not always designed to distinguish between learning processes and action initiation biases (a bias to make a response, regardless of its expected outcome value). Probabilistic learning tasks have suggested an adolescent peak in reward learning[16], better reward learning in adolescents[17], and relatively better reward (versus punishment) learning in adolescents compared to adults[18] (see also[19]). Reversal learning tasks (with changeable outcome probabilities) have pointed to increased punishment learning in adolescents compared to adults[20], a trough in punishment learning rates in mid-adolescence coupled with a sudden increase in reward learning rates in early adulthood[21], or peaks in both punishment and reward learning in late adolescence[22]. Together these studies suggest that reward and punishment learning might differ across adolescence, but they provide inconsistent evidence. Part of this variability is likely due to different learning contexts and task demands[23,24], but it could also reflect the reliance on smaller and non-diverse samples that are not fully representative of adolescents across different countries.

Only one study of adolescents to date, to our knowledge, has measured learning in a task design that incorporates requirements both to learn and also to inhibit actions[25] (see[26] for a study in adults). This study compared reward and punishment learning as well as an action initiation bias or tendency to 'go' (initiate an action) vs. 'no-go' (withhold an action) in children (8–12, $n = 20$), adolescents (13–17, $n = 20$), and adults (18–25 years, $n = 21$). Relative to both children and adults, adolescents exhibited attenuated 'go' and Pavlovian (action-consistent-with-valence) biases. Learning was best captured by a generic (not valence-specific) learning rate, and learning rate was not associated with age in this sample. This study suggests that, like learning rates in previous studies, action initiation biases might display developmental differences across adolescence.

In summary, adolescence has been associated with an enhanced ability to learn from reward and possible differences in learning from punishment, but evidence has been inconsistent. These differences could be due to small sample sizes, two-group designs (which cannot detect quadratic relationships), and lack of learning contexts designed

to assess action biases. Therefore, despite evidence that learning processes can undergo profound changes during adolescence, little is known about how learning mechanisms differ from action initiation biases during this crucial developmental period.

Here, we examined differences in reward and punishment learning and action initiation, using a large sample ($N = 742$) of youths aged 9-18 years recruited from across Europe. Participants viewed a series of abstract 3D objects and had to learn by trial-and-error whether to respond ('go', to win points) or withhold responding ('no-go', to avoid losing points) to each object[27,28] (see Fig. 1). We built a set of reinforcement learning models that were fitted to the data using a hierarchical expectation maximisation approach and compared them using Bayesian model comparison methods[29–31]. These models varied in terms of whether parameters were included for separate reward and punishment learning rates, action initiation biases, and sensitivity to the magnitude (number of points) gained or lost.

We find that a computational model including separate reward and punishment learning rates, and a constant action initiation bias (that measures the tendency to 'go' vs. 'no go' on each trial regardless of reward or punishment) best explains behaviour. In addition, we show an asymmetry in learning differences. While reward learning rates remain stable, punishment learning rates increase from childhood to adolescence. In parallel, despite stable reward learning, action initiation biases decrease with age. All results remain the same when replacing chronological age with pubertal stage. These findings point to normative developmental differences in punishment learning and action initiation. They suggest that theoretical accounts positing heightened responses to reward in adolescence should consider differences in impulsive action initiation rather than solely reward sensitivity or learning, since these can mimic reward learning in some contexts. Such findings are important for our understanding of learning and decision-making in adolescence as well as how learning and action initiation can go awry in the transition from childhood to adolescence.

## Results

We analysed behaviour from 742 participants (491 girls) aged 9-18 years (mean 13.99, SD = 2.48, median pubertal stage 'late pubertal') (see Methods) who completed a reward and punishment learning task (see Fig. 1). All participants were free from psychiatric disorders (see Methods). Pubertal status was measured using the self-report Pubertal Developmental Scale (PDS[32]; see "Methods"). After modelling the learning task data, we tested associations between age, pubertal status, and participants' model parameters, as well as behavioural responses. Age was treated as a continuous variable in all these analyses, although for presentation purposes, we divide age into three discrete bins. To test for quadratic associations between age and model parameters, we tested all models with age[2] included. We first examined whether there were associations between age or pubertal status and sex or IQ. As there were some associations between these measures (see Table S1), we included sex and IQ as covariates of no interest in all analyses of participants' behavioural responses and model parameters. For each of these analyses, we ran two models to assess developmental changes: one with participants' chronological age and one with pubertal status. Six participants who were included in the computational modelling were removed from subsequent analyses due to missing IQ data.

### Behavioural responses suggest age differences in reward and punishment learning
We first examined whether learning occurred during the task. A generalised linear mixed model (GLMM) (predicting correct responses from age, stimulus repetition number, outcome valence, and covariates; see Methods) revealed a significant main effect of stimulus repetition on the number of correct responses made, with performance improving throughout the task (Odds ratio (OR) = 1.19 [1.17, 1.21], z = 18.56, $p < 0.001$, 2-tailed). To confirm that learning occurred for both

reward and punishment stimuli, we repeated the analysis using only reward trials, and found a significant main effect of repetition on correct responses (OR = 1.13 [1.09, 1.16], z = 8.48, p <0.001, 2-tailed). We then repeated the analysis using only punishment trials and again found a significant main effect of repetition on correct responses (OR = 1.28 [1.26, 1.32], z = 19.65, p <0.001, 2-tailed). Thus, the task was able to capture learning behaviour in both reward and punishment conditions.

Next, we tested whether age was associated with behavioural responses over stimuli repetitions (Fig. 2; see also Supplementary Notes, Behavioural responses by stimulus point value, and Figs. S1–2 for behavioural responses broken down by stimulus point value and Figure S3 for relative reward bias over trials). Age was a significant positive predictor of overall learning (GLMM: age by stimulus repetition interaction: OR = 1.02 [1.01, 1.04], z = 2.49, p = .01, 2-tailed) and older participants also made more correct responses in total (OR = 1.08 [1.04, 1.11], z = 4.58, p <0.001, 2-tailed; see Figure S4), consistent with prior developmental learning studies[23]. However, this age-related learning improvement was specific to learning from punishment outcomes (age by repetition by valence interaction: OR = 1.09 [1.05, 1.13],

z = 4.65, p <0.001, 2-tailed). To check the direction of this interaction, we repeated the analysis using only reward trials and found no age by repetition interaction (OR = 0.98 [0.95, 1.01], z = −1.48, p = 0.14, 2-tailed), and repeated it again using only punishment trials and found a significant age by repetition interaction (OR = 1.07 [1.05, 1.11], z = 5.63, p <0.001, 2-tailed). To quantify the strength of evidence for this stable reward learning pattern, we calculated a Bayes factor using the BIC method[33], by repeating the GLMM model for reward trials only (and removing the valence term), then repeating this reward-only regression with the age by repetition interaction removed. This generated strong support for the stability of reward learning across age in this dataset ($BF_{01}$ = 57.80; very strong evidence in support of the null).

Next we assessed whether there were possible quadratic effects of age on behavioural responses. Although this slightly improved the model fit (ΔBIC = −27.79, p = .003, 2-tailed), the $age^2$ term was not a significant predictor of correct responses (OR = 0.99 [0.96, 1.02], z = −0.41, p = 0.68, 2-tailed) or of overall learning ($age^2$ by repetition interaction: OR = 0.99 [0.97, 1.01], z = −0.87, p = 0.39, 2-tailed). However, we did observe a significant $age^2$ by repetition by valence

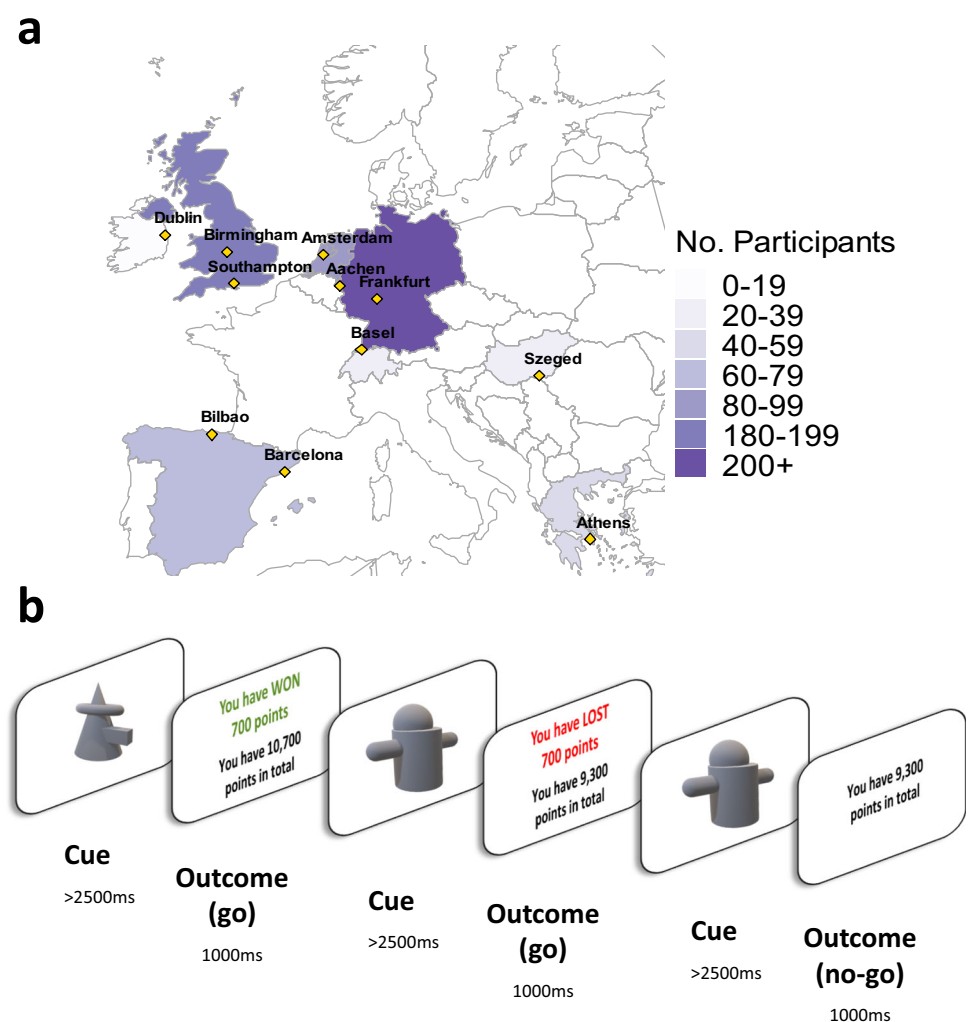

**Fig. 1 | Recruitment sites and learning task. a** Number of participants recruited from each country. Countries are coloured according to the total number of participants, with individual recruitment sites marked in yellow. **b** Details of the learning task (shown here in the English language version). The aim of the task was to learn whether to respond or withhold responses to stimuli in order to earn points. Participants learnt by trial and error whether to make or withhold a button press to obtain a reward (points) or avoid punishment (losing points). Eight unfamiliar stimuli were presented individually for 3000 ms or until a button press

response was made. Responses were followed by feedback on the outcome (1000 ms) or a running total alone if the participant did not respond. Each stimulus had a fixed value of +/− 1, 700, 1400, or 2000 points and was shown once per 'block' for 10 blocks, with a randomised order within blocks. Thus, four stimuli were associated with reward and four with punishment. Participants started the task with 10,000 points and could theoretically finish with a score between 51,010 and −31,010.

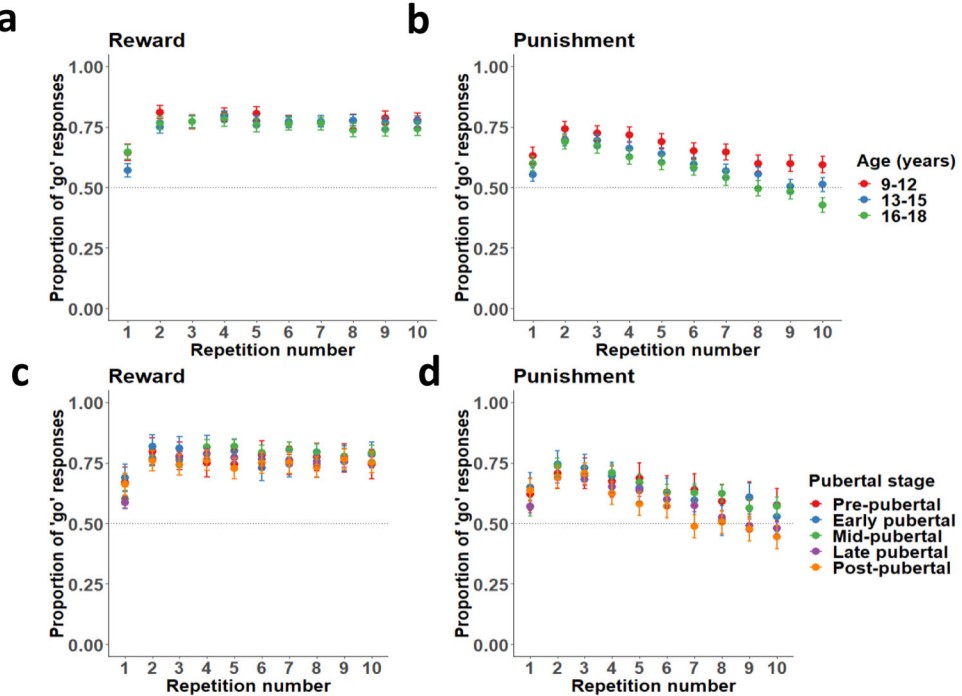

**Fig. 2 | Reward and punishment responding across stimulus repetitions, by age and pubertal stage. a** Proportion of 'go' responses to reward stimuli across repeated stimulus presentations, for three age groups. **b** Proportion of 'go' responses to punishment stimuli across repeated stimulus presentations, for three age groups. **c** Proportion of 'go' responses to reward stimuli across repetitions, by pubertal stage. **d** Proportion of 'go' responses to punishment stimuli across repetitions, by pubertal stage. In all panels, points represent means and error bars are 95% confidence intervals of the mean. *N* = 200 aged 9–12, 303 aged 13–15, and 239 aged 16–18. Dashed lines indicate chance performance. Note that 'go' responses are correct for reward and incorrect for punishment stimuli; thus, learning is demonstrated by increasing responses to reward and decreasing responses to punishment stimuli.

interaction (OR = 1.08 [1.04, 1.12], z = 3.98, *p* <0.001, 2-tailed) as well as the significant age by repetition by valence interaction (OR = 1.09 [1.05,1.13], z = 4.52, *p* <0.001, 2-tailed), suggesting that the punishment-specific improvement in learning was partially non-linear.

Since feedback during the task was given in the form of point scores, we also checked for age-related improvements in point score. As expected, older participants gained more points than younger participants overall (robust linear mixed effects regression: β = 0.16 [0.09, 0.23], z = 4.27, *p* <0.001, 2-tailed).

### Age-related improvement in punishment learning is not better explained by pubertal development

We next examined whether these age-related improvements in punishment learning were also observed for pubertal stage. Similar to age, pubertal stage was positively associated with overall performance (OR = 1.06 [1.03, 1.10], z = 3.71, *p* <0.001, 2-tailed), and with improved learning (OR = 1.03 [1.01, 1.05], z = 2.95, *p* = 0.003, 2-tailed). However, we did not observe a significant pubertal stage by repetition by valence interaction (OR = 1.04 [0.10, 1.07], z = 1.87, *p* = 0.06, 2-tailed). Furthermore, the model using age was a better fit to the data than the model using pubertal stage (ΔBIC = −137.22). To confirm that the lack of pubertal stage by repetition by valence interaction was not a reflection of the number of pubertal categories, we repeated the analysis with pubertal stage collapsed into three categories (pre/early, mid, and late/post-pubertal), and observed no differences (see Supplementary Notes, Behavioural responses by three pubertal stages).

### Computational modelling shows that a model with separate reward and punishment learning rates and an action initiation bias best explain behaviour

Behavioural analyses suggested that age related differences in learning were specific to punishment but not reward learning. However, to

quantify reward and punishment learning precisely, as well as other latent mechanisms that might differ across age, computational models of reinforcement learning are needed. We therefore compared a range of computational models of reinforcement learning to characterise participants' choice behaviour. In particular, we compared models that varied in terms of a single learning rate or separate learning rates for reward and punishment (influence of recent outcomes on future responses), initial or constant action initiation biases (bias to respond versus not respond on the first presentation of an object, or bias to respond versus not respond across all trials, respectively) and sensitivity to the magnitude of reward, punishment, or both (sensitivity to points gained or lost). Models were fitted using a hierarchical expectation maximisation approach and compared using Bayesian model comparison methods[29–31,34]. We constructed seven different models using an iterative procedure to appropriately constrain the model space (see Methods for full details):

1. αβ: single learning rate (α) and temperature parameter (β)
2. 2αβ: reward α, punishment α, β
3. αβb_i (1): single α, β, initial 'go' bias (*b_i*)
4. αβb_c (2): single α, β, constant 'go' bias (*b_c*)
5. 2αβb_i or 2αβb_c: reward α, punishment α, β, *b_i* or *b_c* (depending on winner from 3. & 4.)
6. 2αβb_iρ or 2αβb_cρ: reward α, punishment α, β, *b_i* or *b_c*, magnitude sensitivity (ρ)
7. 2αβb_i2ρ or 2αβb_c2ρ: reward α, punishment α, β, *b_i* or *b_c*, reward ρ, punishment ρ

Models were compared on exceedance probability, Log Model Evidence (LME), and the integrated Bayesian Information Criterion (BIC_int). Initially we found that model 6, which included separate learning rates for reward and punishment, a constant action initiation bias, and a single (valence-insensitive) magnitude sensitivity

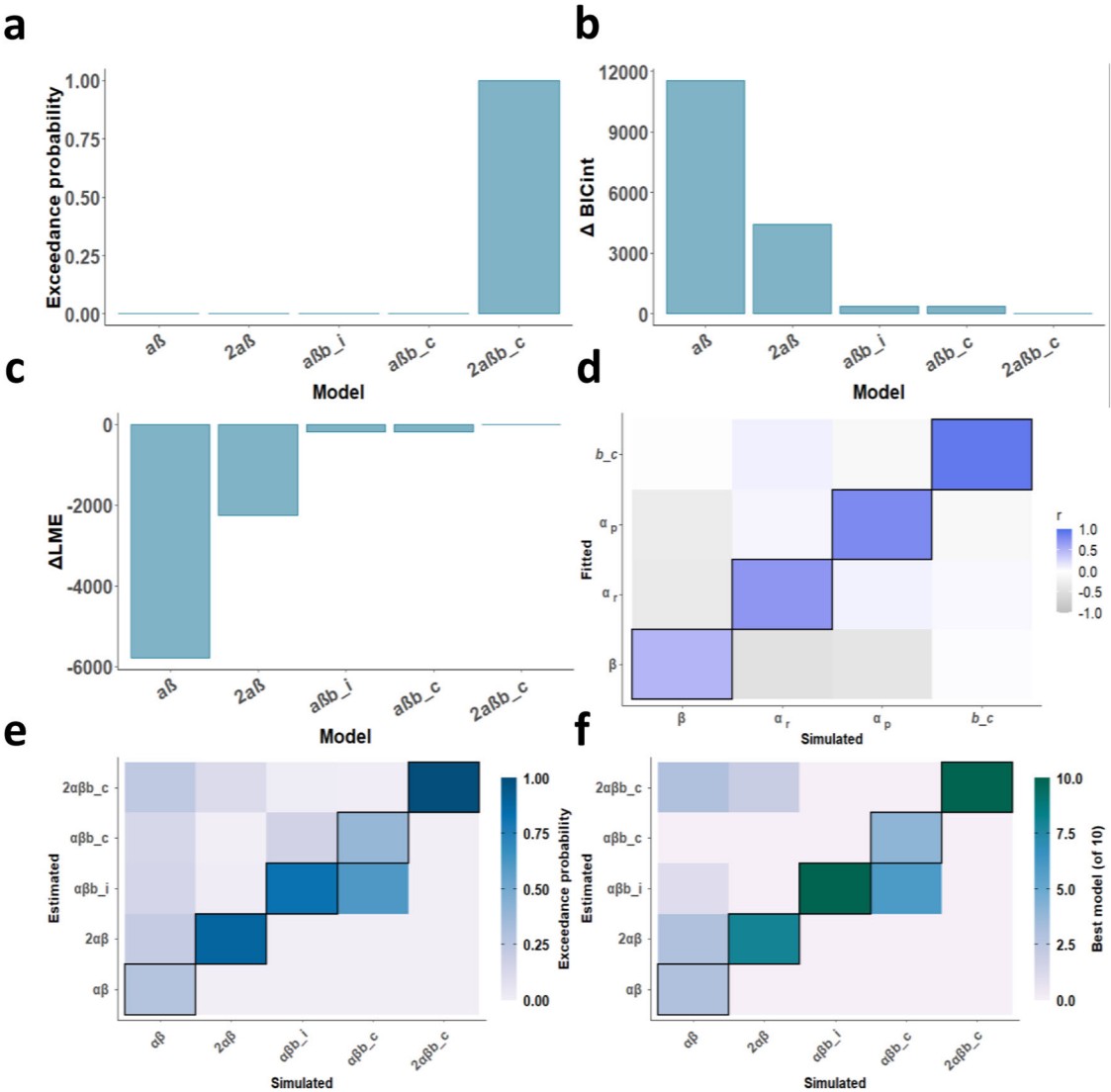

**Fig. 3 | Model performance and validation. a** Exceedance probability for the five computational models that comprised the final model space. The winning model was the 2αβ*b_c* model, with separate reward and punishment learning rates, and a constant action initiation bias. **b** ΔBIC_int, relative to the winning model (2αβ*b_c*). **c** ΔLME, relative to the winning model (2αβ*b_c*). **d** Parameter recovery. The confusion matrix represents Spearman correlations between simulated and fitted (recovered) parameters. Each parameter exhibited a significant positive correlation between its true and fitted values, with *r* values ranging from 0.49–0.92 (shown on the lower diagonal). **e** Exceedance probability from the model identifiability procedure. The diagonal represents the probability of each model having the best fit to its own synthetic data. The winning model (2αβ*b_c*) was highly identifiable from other models. **f** Number of runs where each model was selected as the best fit for data generated by each model in the model identifiability procedure. The diagonal represents the number of runs each model was selected as the best fit for its own data. The winning model (2αβ*b_c*) was the best fit to its own data on all 10 runs. This Figure also relates to Figs. S5 and S7.

parameter, best explained behaviour. This model had the highest exceedance probability (0.99) and the highest LME (−34066.81), and performed similarly to model 5 on BIC_int, which had the lowest absolute BIC_int (see Fig. S5). However, when we further validated the winning model using parameter recovery and model identifiability procedures (see Methods for details), the magnitude sensitivity parameter exhibited relatively poor recoverability (r = 0.11). We therefore selected model 5, with separate learning rates for reward and punishment and a constant action initiation bias, as our winning model (see Fig. 3). Except for LME, model 5 exhibited similar performance to model 6 (Fig. 3), and showed good recovery and identifiability for the winning model (Fig. 3 and Fig. S5). Moreover, all associations with age remained the same regardless of whether model 5 or 6 was selected (See Supplementary Notes, Associations with age are similar for model 6 parameters, and Table S2).

Additional modelling analyses in three separate age groups (9–12, 13–15, and 16–18 years) confirmed that model 5 won across the full age range (See Figure S6). As a further control analysis, we repeated the modelling procedure with the first presentation of each stimulus removed from the data, in case our main findings were skewed by the effect of stimulus unfamiliarity early on. For this analysis, action values were initialised as the mean 'go' response for the first set of presentations. Our results were unchanged when modelling only repetitions 2–10 (see Table S2, and Figs. S7 and S8).

### Punishment learning rates increase with age, while action initiation biases decline
Next we assessed whether the parameters from the winning computational model varied as a function of age, using GLMMs predicting parameter values from age and covariates.

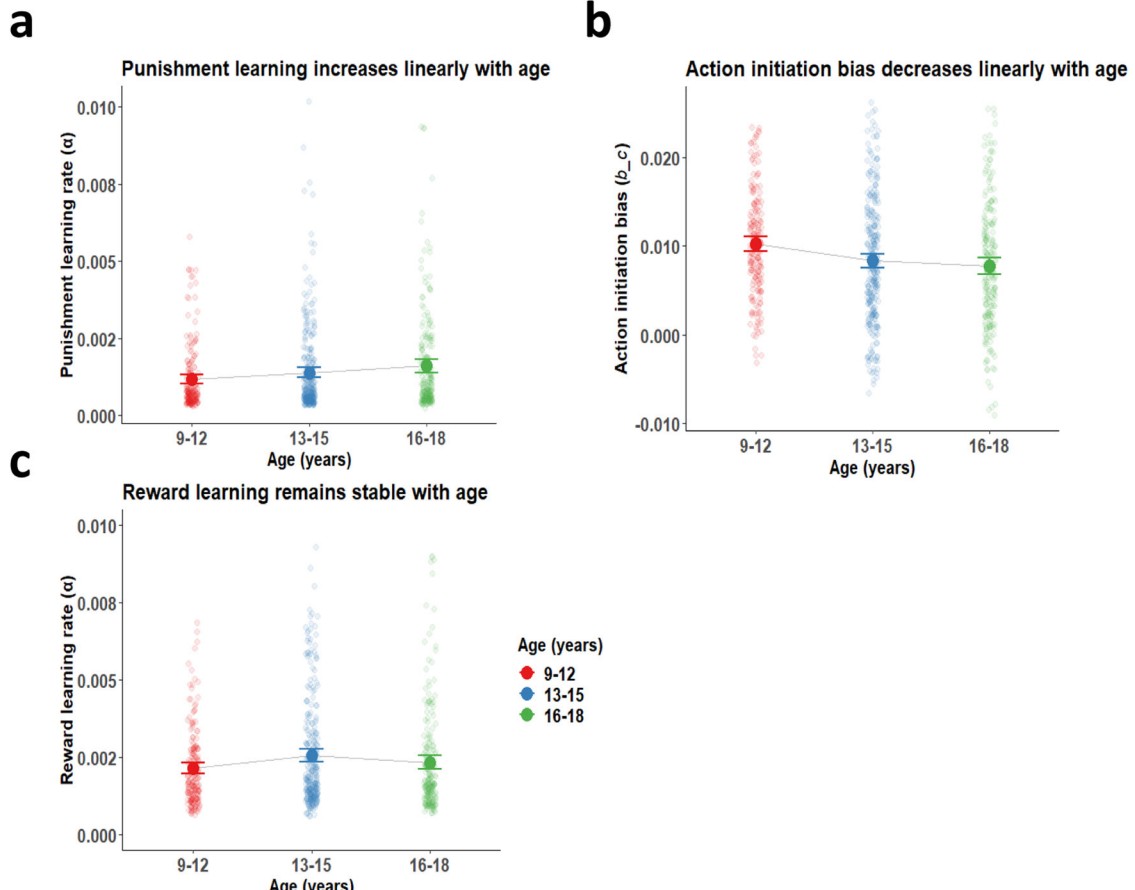

**Fig. 4 | Age differences in action initiation bias and punishment learning, but stable reward learning. a** Punishment learning rate across three age groups. Punishment learning rates increased linearly with age (GLMM; β = 0.10 [0.05, 0.15], z = 4.12, *p* <0.001 (2-sided), BF$_{10}$ = 167.25, BF$_{01}$ = 0.01). **b** Action initiation bias across three age groups. Action initiation biases declined linearly with age (GLMM; β = −0.20 [−0.28, −0.12], z = −4.91, *p* < 0.001 (2-sided), BF$_{10}$ = 8926.69, BF$_{01}$ = 0.0001). **c** Reward learning rates across three age groups. Reward learning

rates remained stable with age (GLMM; β = 0.01 [− 0.05, 0.07], z = 0.30, *p* = 0.77 (2-sided), BF$_{10}$ = 0.07, BF$_{01}$ = 13.60), including no significant quadratic effect (β = −0.84 [− 2.37, 0.69], z = −1.07, *p* = 0.28). Points and error bars represent means and 95% confidence intervals of the means for each group, with raw data represented by smaller points. *N* = 200 aged 9–12, 303 aged 13–15, and 239 aged 16–18. Division into age groups is for presentation purposes only; age was treated as a continuous variable in all analyses. This Figure also relates to Fig. S8.

Strikingly, age was strongly associated with increased punishment learning rates (β = 0.10 [0.05, 0.15], z = 4.12, *p* <0.001, 2-tailed), and lower action initiation biases (β = −0.20 [− 0.28, −0.12], z = −4.91, *p* < 0.001, 2-tailed; see Fig. 4). Importantly, reward learning rates did not differ significantly with age (β = 0.01 [− 0.05, 0.07], z = 0.30, *p* = 0.77, 2-tailed). To confirm the strength of these associations, and check the strength of evidence for any null effects, we calculated Bayes factors using the BIC method[33]. For each model parameter of interest, we compared two linear mixed effects regression models: our 'standard' model, predicting the model parameter from age and covariates, and a 'null' model, which predicted parameter values from the covariates only. We then calculated the BIC of each of these two models, and used the difference between the BICs to calculate a Bayes factor[33]. We observed decisive evidence for the associations between age and punishment learning rate (i.e., the model including age was a better fit; BF$_{10}$ = 167.25, BF$_{01}$ = 0.01) and between age and action initiation bias (BF$_{10}$ = 8926.69, BF$_{01}$ = 0.0001). In contrast, there was no evidence for associations between age and reward learning rate (BF$_{10}$ = 0.07, BF$_{01}$ = 13.60, strong evidence in support of the null). Continuous plots of age and these model parameters are provided in Fig. S10.

We observed no credible evidence for a relationship between age and temperature parameter (β = 0.002 [−0.07, 0.07], z = 0.08, *p* = 0.94, 2-tailed), and Bayes factors showed strong evidence for a lack of difference (BF$_{10}$ = 0.04, BF$_{1}$ = 26.11). All associations with age remained the

same when using fitted parameters from model 5 or model 6, which included the magnitude sensitivity parameter (see Table S2).

Lack of associations between age and model parameters might also reflect non-linear associations, especially for reward learning (see Fig. 4c). We therefore tested for quadratic effects of age by adding age[2] terms to the models. However, none of the model parameters exhibited significant quadratic associations with age (temperature parameter: β = 1.06 [−0.78, 2.90], z = 1.13, *p* = 0.26, 2-tailed. Reward learning rate: β = −0.84 [− 2.37, 0.69], z = −1.07, *p* = 0.28, 2-tailed. Punishment learning rate: β = 0.35 [− 0.87, 1.57], z = 0.56, *p* = .57, 2-tailed. Action initiation bias: β = −0.60 [−2.67, 1.45], z = −0.58, *p* = 0.56, 2-tailed).

Next, we assessed whether pubertal stage also predicted differences in punishment learning and action initiation (Fig. 5). These analyses revealed a similar positive association with punishment learning rate (β = 1.20 × 10$^{-4}$ [4.90 × 10$^{-5}$, 1.92 × 10$^{-4}$], z = 3.32, *p* <0.001, 2-tailed), a negative association with action initiation bias (β = −1.06×10$^{-3}$ [− 1.65 × 10$^{-3}$, −4.80 × 10$^{-4}$], z = −3.57, *p* <0.001, 2-tailed), and no significant association with reward learning rate (β = 6.30 × 10$^{-5}$ [− 3.90 × 10$^{-5}$, 1.65 × 10$^{-4}$], z = 1.21, *p* = 0.22, 2-tailed). There was no significant association with temperature parameter (β = −5.48×10$^{-7}$ [− 1.55 × 10$^{-6}$, 4.53 × 10$^{-7}$], t = −1.08, *p* = 0.28, 2-tailed; these statistics for temperature parameter are from a standard linear model without site, instead of GLMM as above, due to non-convergence of the more complex model).

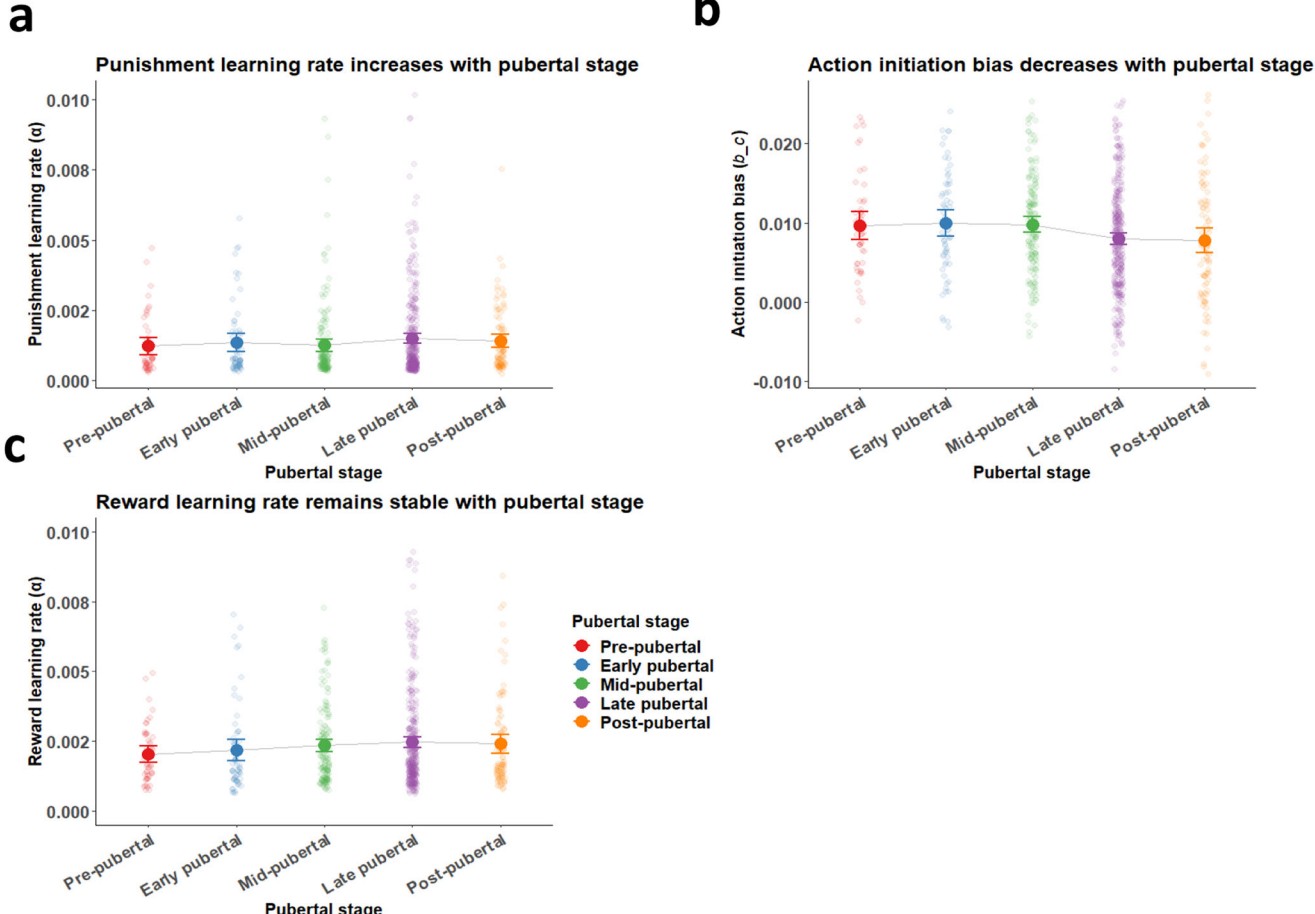

**Fig. 5 | Pubertal maturity differences in action initiation bias and punishment learning, but stable reward learning. a** Punishment learning rates across five pubertal stages. Punishment learning rates increased with pubertal stage (GLMM; $\beta = 1.20 \times 10^{-4}$ [$4.90 \times 10^{-5}$, $1.92 \times 10^{-4}$], z = 3.32, p < 0.001, 2-sided). **b** Action initiation bias across five pubertal stages. Action initiation biases decreased with pubertal stage (GLMM; $\beta = -1.06 \times 10^{-3}$ [$-1.65 \times 10^{-3}$, $-4.80 \times 10^{-4}$], z = -3.57, p < 0.001, 2-sided). **c** Reward learning rates across five pubertal stages. Reward learning rates were stable across puberty (GLMM; $\beta = 6.30 \times 10^{-5}$ [$-3.90 \times 10^{-5}$, $1.65 \times 10^{-4}$], z = 1.21, p = 0.22, 2-sided). N = 52 pre-pubertal, 65 early pubertal, 167 mid-pubertal, 356 late pubertal, and 102 post-pubertal. Points and error bars represent means and 95% confidence intervals of the means for each group, with raw data represented by smaller points.

## Model parameters predict task performance

We next assessed whether differences in model parameters were associated with task performance, using Spearman's rank correlations due to non-normal distributions of model parameters. Overall task performance (proportion of correct responses) was positively correlated with reward learning rate (Spearman's $r_{(696)} = 0.40$ [0.33, 0.46], p < 0.001, 2-tailed) and punishment learning rate (Spearman's $r_{(696)} = 0.68$ [0.63, 0.72], p < 0.001, 2-tailed) and negatively correlated with action initiation bias (Spearman's $r_{(696)} = -0.26$ [$-0.33$, $-0.19$], p < 0.001, 2-tailed). Temperature parameter values were also negatively correlated with task performance (Spearman's $r_{(696)} = -0.39$ [$-0.45$, $-0.32$], p <0.001, 2-tailed). These correlations demonstrate that the optimal strategy for this task is captured by higher learning rates for both reward and punishment, combined with a lower action initiation bias. Correlations between model parameters and reward and punishment task performance are shown in Supplementary Notes, Correlations between model parameters and task performance, and Table S3. Correlations between model parameters are shown in Table S4. Correlations between parameters were small to moderate.

## Winning computational model tracks learning from reward and punishment

To further confirm the accuracy of our model, we generated synthetic behavioural data for 742 'participants' using the winning model and its median parameter values. As a control analysis, we also generated synthetic data using the version of the winning model with only stimulus repetitions 2–10 included (Fig. 6). We then repeated the procedure separately for participants aged 9-12, 13–15, and 16–18 years to ensure that the model captured behaviour adequately across the full age range (see Fig. S11). In both the main 742-participant sample (with and without the first stimulation presentation) and the aged-based samples, the simulated responses fell within a similar range to the real responses. They followed a somewhat similar trajectory, particularly when omitting the first stimulus presentations and considering them as part of the practice phase (Fig. 6 and Fig. S11). Importantly, parameter associations with age were identical whether we included all trials or only repetitions 2–10 (Table S2 and Fig. S8).

## Discussion

Adolescence is often considered as a period in which reward sensitivity peaks[1–5]. Using a large, well-characterised, multi-country sample, we demonstrate that, in fact, reward learning rates in certain contexts remain stable across adolescence whilst the tendency to initiate actions decreases with age. Moreover, punishment learning rates increase across adolescence, with the oldest adolescents learning the most rapidly from punishment feedback. These findings remained the same when we replaced chronological age with pubertal status, and we found evidence that these differences in model parameters reflected

## All repetitions

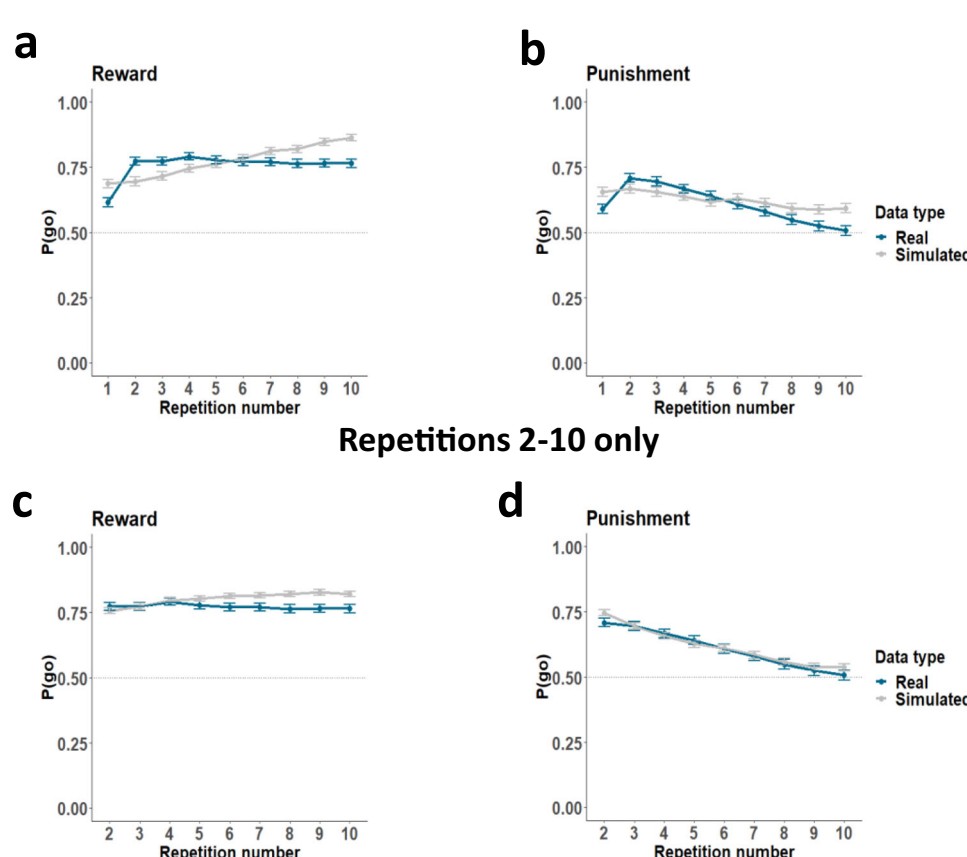

## Repetitions 2-10 only

**Fig. 6 | Simulated probability of 'go' response to reward and punishment stimuli across stimulus repetitions. a** Simulated and real probability of 'go' responses on reward trials, with simulated data generated using the winning model and its median parameter values for the full sample ($N = 742$) across all 10 stimulus repetitions. **b** Simulated and real probability of 'go' responses on punishment trials for the full sample ($N = 742$) across all 10 stimulus repetitions. **c** Simulated and real probability of 'go' responses on reward trials for the full sample ($N = 742$) across stimulus repetitions 2–10 only. **d** Simulated and real probability of 'go' responses on punishment trials for the full sample ($N = 742$) across stimulus repetitions 2–10 only. This Figure relates to Fig. S11. Points and error bars represent means and 95% confidence intervals of the means.

linear associations across adolescence rather than quadratic effects. Together, our findings suggest that the tendency to initiate actions and learn from punishment shifts from late childhood across adolescence and that future research should account for changes in action initiation when evaluating differences in valenced processing of reward and punishment. Our findings also demonstrate these associations robustly by testing a large and international, and therefore potentially more representative, sample.

These results highlight the importance of distinguishing between valenced learning mechanisms and action initiation biases. While previous research has demonstrated heightened reward learning in adolescence[16,18], we show that developmental differences in reward learning might in fact be specific to certain learning contexts, and apparent reward-oriented behaviour can reflect action initiation biases rather than reward learning processes. Knowledge of these developmental differences is an important prerequisite for understanding how adolescent development can go awry, for example in behavioural disorders, where there appear to be disruptions in reinforcement learning[8,35]. It is plausible that adolescent-onset psychopathologies, such as certain forms of conduct disorder, represent aberrant developmental pathways in which these normative increases in punishment learning and declines in action initiation biases could be disrupted. This would be consistent with the symptoms of conduct disorder, which include aberrant learning from punishment as well as impulsivity[36]. Life experience is also an important factor to consider

here, alongside biological changes[37]. Even in healthy adolescents, a better understanding of the difference between reward-oriented and impulsive behaviour could potentially facilitate behavioural interventions designed to reduce risky behaviour. For example, there may be contexts where it is more beneficial to focus on planning and impulse control than on learned behaviour, even when risky behaviour appears superficially to be driven by desirable outcomes (e.g., social status or material goods). These are important directions for future research.

One consideration is whether action initiation biases are themselves influenced by the prospect of a rewarding outcome, since there are forms of impulsivity that occur specifically in situations where a possible reward is anticipated[38,39]. Since 'go' responses in the current study necessarily occur in the context of possible reward, it is possible that the action initiation bias reflects a type of reward-related impulsivity. However, we have two reasons to suspect that this is not the case. First, in contrast to the classic go/no-go paradigm (where 'go' responses are required substantially more often than no-go responses), our task used equal numbers of go-for-reward and no-go-for-punishment trials. Participants who make 'go' responses blindly (i.e., in the absence of learning) are therefore equally likely to receive punishments as to receive rewards. Consequently, the classic go/no-go element of being 'primed' to make responses to gain points (because 'go' responses are more often correct) is missing from this task. Second, we tested a model that captured sensitivity to reward magnitude, but this model was outperformed by models with a generic or no

magnitude sensitivity. This further suggests that there was no sensitivity to reward driving behaviour other than that captured by the learning rate. These considerations do not support a role for reward in triggering the action initiation bias. Future studies could include 'go to avoid punishment' and 'no-go to gain reward' conditions to capture the full influence of action biases on reward or punishment responses in a large sample[25]. However, the action initiation bias we observed appears to be a genuine action bias, rather than a deliberate strategy or an indirect effect of reward facilitating action.

We also note that participants were not given practice trials, to encourage learning from the first stimulus onwards. However, as a control analysis, we repeated our analyses with these initial trials excluded. The analysis of stimulus repetitions 2–10 showed that the associations between age and model parameters remained unchanged, supporting the robustness of these associations. Furthermore, we assessed whether there were any age differences between excluded participants and the final sample, and obtained substantial evidence in support of the null hypothesis that data exclusions were not biased by age. Our findings therefore appear to be robust to data exclusions and across repetitions 2–10 as well the full learning task.

Previous research has painted a mixed picture of punishment learning in adolescence, with different studies reporting decreases[21,40] and increases in punishment learning during the adolescent period[20,22]. It is likely that these differences at least partially reflect variation in task design; in particular, having a higher or lower learning rate can be more or less beneficial depending on the task[23,24]. We observed a positive correlation between accuracy and punishment learning rates across age, suggesting that higher punishment learning rates as seen in older adolescents were more optimal for this task. Thus, the higher punishment learning rates exhibited by older adolescents are indicative of better overall performance. Importantly, however, we did not see increases in reward learning rates across adolescence, although these too were correlated with overall performance. Therefore, the higher punishment learning rates were not simply a reflection of higher general ability on the task, but rather seem to reflect a more specific ability to recall previous punishments and inhibit responses as a result. Crucially, we observe these results in a large sample of adolescents from multiple countries, providing substantive support for developmental differences in punishment learning.

Although there have been previous reports of heightened reward learning in adolescence[16,18], the only other study to use a go/no-go design did not observe separate learning rates for reward and punishment[25]. By contrast, our winning model did contain separate learning rates for reward and punishment, demonstrating an asymmetry in learning. However, the lack of an age effect for reward learning in the current study and the lack of a separate learning rate for reward in previous studies[25] both suggest that reward learning rates are not related to age in a context where action initiation biases can occur. It is theoretically possible that a strong action initiation bias would remove the need for reward learning, since participants could 'default to go' and then simply learn from punishment. Again, however, there was a clear association between the reward learning parameter and task performance, and when action initiation biases were lowest in older participants, there was no increase in reward learning rates. This suggests that reward learning was necessary for better performance, even if it did not improve with age. Moreover, for all parameters where we observed differences across development, we saw the same associations when considering pubertal stage, suggesting that our findings were robust across different measures of development capturing pubertal stage as well as chronological age.

Our study has several strengths. It is among the first to test how action initiation biases and learning differ concurrently across the full spectrum of adolescence, using a learning context that manipulates the requirement for learning and action initiation, something that has often been neglected in computational modelling studies of learning.

We used a large ($N = 742$), mixed-sex sample, collected from multiple different countries and speaking different languages, carefully screened to be typically developing in terms of psychiatric functioning, and well characterised in terms of social background. The international nature of our sample helps our findings to represent adolescent behaviour more universally. In all analyses, we modelled test site as a random effect, and our results remain robust to this modelling. Given the fundamental nature of reward and punishment learning[41], we did not anticipate or set out to test country-level differences, but future studies could seek to include an even larger and more geographically diverse sample to further investigate possible differences between countries and cultures in learning rates across adolescence. We built and tested several different plausible models of learning and used multiple measures to validate them, and we also used measures of pubertal stage as well as chronological age to further elucidate developmental differences in learning.

However, we note some limitations to the study. First, our learning task did not contain 'no-go to gain reward' and 'go to avoid punishment' conditions, meaning that we were unable to assess Pavlovian action biases[25]. It would be interesting for future studies to implement probabilistic learning tasks with the full action-valence crossover. Second, outcomes were deterministic, which has generally not been the case in previous studies (except[42]). It is possible that the relationship between learning rates and performance in this context is different from that observed when using the more common probabilistic and reversal learning tasks[23,24]. In addition, while our task was able to separate action initiation biases from learning processes, there were several other possible parameters that it was not practical or theoretically meaningful to assess using this design. For example, we did not include variable learning rates[43], choice stickiness parameters[44], or the role of forgetting[42], which could be measured in future research using a suitable probabilistic learning task, possibly with drifting rewards to capture behavioural variability. Such a task may also help to separate correlations between model parameters more clearly, which were in some cases moderately correlated, although the differential association with age supports their distinction.

In summary, we tested developmental differences in learning and action initiation biases in a large, cross-sectional sample of typically developing adolescents aged 9–18 years. Behaviour was best explained by a model with separate learning rates for reward and punishment as well as a constant action initiation bias, and we observed normative developmental differences in these parameters, associated with both chronological age and (to a lesser extent) pubertal stage. Specifically, we observed linear declines in action initiation biases and increases in punishment learning across adolescence, combined with stable levels of reward learning. We conclude that adolescents develop an increasing ability to inhibit actions, learn from negative outcomes, and make more selective behavioural responses as they transition through adolescence and approach adulthood. These findings help add to theoretical and empirical accounts that largely focus on enhanced reward processing and suggest that action biases and punishment learning are crucial processes to understand across adolescence.

## Methods
### Participants
Participants were selected from the FemNAT-CD project database[45]. All participants included in the present analyses had completed the reinforcement learning task, were 9-18 years old, and were classed as typically developing, with no current psychiatric diagnoses (including autism), learning disability, serious physical illness, or histories of disruptive behaviour disorders including ADHD (see Questionnaire measures below). Eight hundred and thirty-two participants were eligible for inclusion. We screened the data to exclude participants with poor task performance. Four participants never responded, two responded to every trial, six scored below zero points on the task

(indicating deliberate punishment-seeking and reward-avoidance), and 78 responded to fewer than half of the reward trials (i.e., trials where responding was the correct behaviour). The final sample thus consisted of 742 youths (491 girls, 251 boys; sex is defined here as self/parent-reported biological sex. The sex imbalance here reflects a deliberate over-sampling of girls in the larger study from which these data were taken). Of these, 52 were classed as pre-pubertal, 65 as early pubertal, 167 as mid-pubertal, 48 as late pubertal, and 16 as post-pubertal. There was no difference in age between the sexes in the final sample (t-test: t$_{(488)}$ = 1.00, $p$ = 0.20, 2-tailed, BF$_{01}$ = 5.03, moderate evidence in support of the null). The participants were recruited from 11 sites across Europe (Aachen: 139, Frankfurt: 140, Birmingham: 103, Amsterdam: 90, Southampton: 89, Bilbao: 55, Athens: 49, Szeged: 33, Basel: 28, Barcelona: 12, Dublin: 4). For LMM and GLMM (i.e., non-modelling) analyses only, we excluded an additional six participants who were missing IQ data. For the analyses of model parameters and age, we excluded 38 participants with values more than three standard deviations from the mean on one or more model parameters. Reanalysing the data with all eligible 832 participants did not change our results (see Supplementary Notes, Reanalysis with all 832 eligible participants and no exclusion of outliers, Table S2 and Fig. S12).

All participants provided written informed consent (if over the age of consent in their country) or written informed assent, with written informed consent provided by a parent or guardian. Participants completed the learning task as part of the larger study and received a small monetary or voucher reimbursement in line with local ethical approvals[46]. This payment was not linked to the learning task specifically, and was therefore not associated with task performance.

## Inclusion and ethics
The data used in this study were collected from several European countries as part of the FemNAT-CD project. Local researchers from each site have been included as co-authors in line with FemNAT-CD guidelines and authorship requirements. Roles and responsibilities were determined in advance by FemNAT-CD members. Local ethical approval was obtained from the relevant authorities at each site, as detailed above. The data reported here did not involve risks to the safety, wellbeing, or security of participants or researchers.

## Ethics declarations
The FemNAT-CD project received ethical approval from the relevant local ethics committees, as follows: Aachen: Ethik Kommission Medizinische Fakultät der Rheinisch Westfälischen Technischen Hochschule Aachen (EK027/14). Amsterdam: Medisch Etische Toetsingscommissie (2014.188). Athens: Election Committee of the First Department of Psychiatry, Eginition University Hospital (641/9.11.2015). Barcelona: Child and Adolescent Mental Health−University Hospital Mutua Terrassa (acta 12/13). Basel: Ethik Kommission Nordwest- und Zentralschweiz (EKNZ 336/13). Bilbao: Hospital del Basurto. Birmingham and Southampton: University Ethics Committee and National Health Service Research Ethics Committee (NRES Committee West Midlands, Edgbaston; REC reference 3/WM/0483). Dublin: SJH/AMNCH Research Ethics Committee (2014/04/Chairman (3)). Frankfurt: Ethik Kommission Medizinische Fakultät Goethe Universität Frankfurt am Main (445/13). Szeged (Hungary): Egészségügyi Tudományos Tanács Humán Reprodukciós Bizottság (CSR/039/00392−3/2014). This study was conducted in accordance with the ethical standards of the 1964 Declaration of Helsinki and its later amendments.

## Questionnaire and interview measures
Participants were assessed for current and past psychiatric and behavioural disorders using the K-SADS-PL clinical interview[47]. The K-SADS-PL is a semi-structured diagnostic interview used to assess psychopathology in children and adolescents. We conducted interviews separately with participants and with their parents, or another

responsible adult informant if a parent was not available. All researchers administering the interview had been trained in its use. After completing the two interviews (parent and participant), we then generated combined parent and child summary ratings of all symptoms (past, present, and lifetime). Where assessors gave discrepant ratings for a symptom, they discussed all available information until an agreement was reached for the summary rating. Except for CD, ODD, and ADHD, where DSM-5 criteria were used, all diagnoses were generated based on the DSM-IV-TR diagnostic criteria, which were current at the outset of the project[48]. For the current study, participants were excluded if they met the diagnostic criteria for any current disorder, or (due to broader project requirements) any history of externalising behavioural disorders.

IQ was assessed with the vocabulary and matrix reasoning subscales of the Wechsler Abbreviated Scale of Intelligence[49] at English-speaking sites, or with the vocabulary, block design, and matrix reasoning subscales of the Wechsler Scale for Children (participants <17 years) or Wechsler Adult Intelligence Scale (17–18 years[50]).

Pubertal stage was assessed using the self-report Pubertal Developmental Scale (PDS[32]), which assesses growth of body and facial hair, change of voice, and menstruation. Each item is rated on a scale from 1 (*not yet* started) to 4 (*seems* complete). These subscales are then summed to yield an overall pubertal stage score: pre-pubertal (1), early pubertal (2), mid-pubertal (3), late pubertal (4) or post-pubertal (5).

Socioeconomic status (SES) was assessed based on parental income, education, and occupation. Assessments were based on the International Standard Classification of Occupations (International Labour Organization; www.ilo.org/public/english/bureau/stat/isco/) and the International Classification of Education (UNESCO; uis.unesco.org/en/topic/international-standard-classification-education-isced). Human ratings and computer-based ratings were combined into a factor score using principal component analysis. A clear one-dimensional structure underlying the different measures could be corroborated using confirmatory factor analysis (comparative fit index = 0.995; root mean square error of approximation = 0.035). Reliability of the composite SES score was acceptable (Cronbach's α = 0.74). To account for economic variation between countries, the final SES score was scaled and mean-centred within each country, providing a measure of relative SES. Missing data were imputed by statisticians at the Institute of Medical Biometry and Statistics (Freiburg, Germany), as described below.

## Imputation of missing data
Missing data were imputed by statisticians at the Institute of Medical Biometry and Statistics (IMBI), a member of the FemNAT-CD project. Missing data for the PDS were imputed separately, before the decision was made to impute missing values for other measures. The procedure for the PDS imputation is thus described separately from the other measures. The following description is a standard text provided by IMBI, for use in all FemNAT-CD project publications.

Missing values of the PDS score were imputed based on the whole FemNAT-CD sample. It has been shown that missing data in a multi-item instrument is best handled by imputation at the item level[51]. Thus, missing values of the single items were imputed first, and the scores were calculated based on the imputed items. The imputation was done in SAS® version 9.4 using the procedure PROC MI. Imputation by fully conditional specification (FCS) is used, which offers a flexible method to specify the multivariate imputation model for arbitrary missing patterns including both categorical and continuous variables[52]. As the items are measured at an ordinal level, the logistic regression method is specified in the FCS statement. For imputation diagnostics, distribution of the observed and imputed items and scores were checked. The imputation of the PDS items was done separately in males and in females because of sex specific items: item 2 (females and males) and items 4, 5a of the form for females or items 4, 5 of the form for males

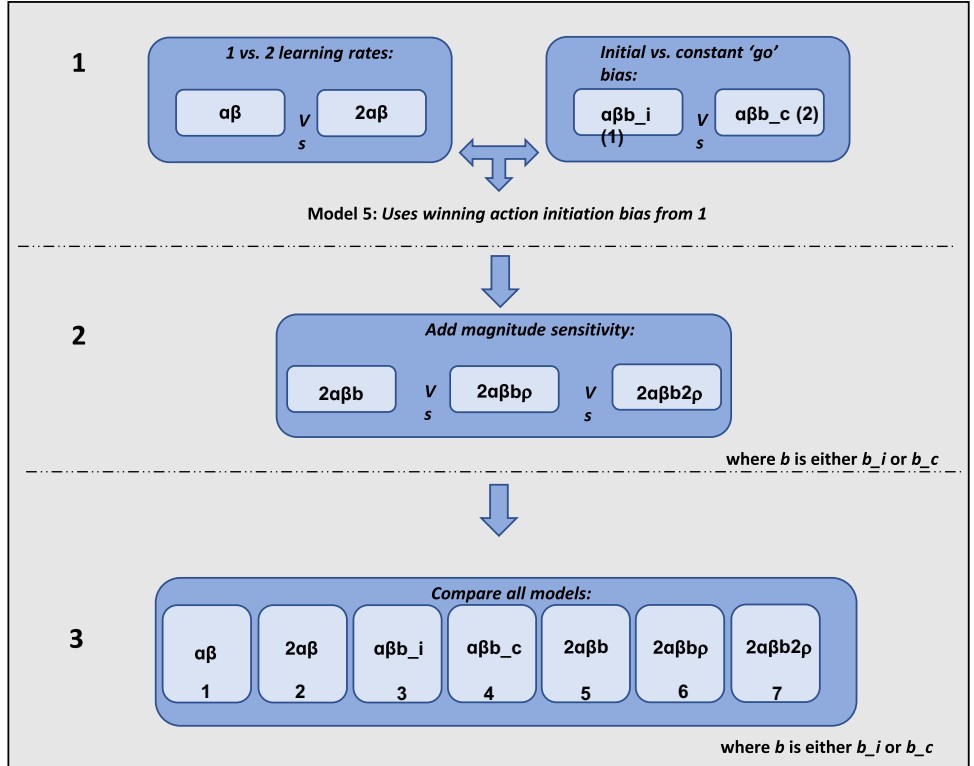

**Fig. 7 | Steps in model construction procedure.** In the first step (**1**), models with one versus two learning rates were compared, and separately, models with an initial versus constant action initiation bias were compared. A fifth model was then constructed by combining all parameters from the winning models in step **1** (i.e., one versus two learning rates and the winning action initiation bias). In step **2**, we tested whether model 5 was improved by adding a single magnitude sensitivity parameter (model 6) or separate magnitude sensitivity parameters for reward versus punishment outcomes (model 7). Finally, to confirm that the winning model from step **2** was the best overall model, we compared models 1–7 directly in step **3**.

were imputed respectively. The following variables were included in the imputation model: sex specific items of the PDS as mentioned above and the two remaining PDS items (items 1 and 3), age at PDS and age at informed consent, to impute age at PDS if missing, weight, case/control status, site, and migration status.

Imputation for the remaining measures was conducted separately, following the same procedure as above. The following variables were included in the imputation model: all items of the respective questionnaire, age, IQ, group (case/control), sex (male/female), site, comorbidities (post-traumatic stress disorder (PTSD), attention-deficit/hyperactivity disorder (ADHD), oppositional defiant disorder (ODD), depression, anxiety), and items of other questionnaires if correlated with at least one of the items with ≥.4. For imputation diagnostics, distribution of the observed and imputed items and scores were checked. Data were not imputed for the learning task itself.

### Learning task
Participants completed a 'passive avoidance' reinforcement learning task on a computer in a quiet testing room. The task was adapted from two previous studies[53,54] and presented in E-Prime[55]. The aim of the task was to gain points by pressing a button when presented with 'good' objects (to earn points) and withholding responses when presented with 'bad' objects (to avoid losing points). Participants were carefully instructed that some of the objects were 'good' (would earn them points) and some were 'bad' (would lose them points), and that they must learn which objects were good and which were bad so that they could respond only to the good objects. The researcher checked that this was clear to each participant. There were no practice trials before the main experiment so that participants would be learning for the first time from the first trial onwards. In order to maximise their point score, participants thus had to learn through trial-and-error which

objects were associated with reward and which with punishment. There were eight different objects in total, four associated with rewards and four with punishment, with values of +/−1, +/−700, +/−1400, or +/−2000 points. The point value associated with each object was fixed and did not change throughout the task. The eight objects were each presented 10 times in a random order (thus 80 trials in total). Each response was followed by feedback on the number of points gained or lost plus the running total; when participants did not respond, the value of the object was not revealed (see Fig. 1). Stimuli were displayed for 3000 ms or until the participant responded, and feedback (or the running total alone) was then displayed for 1000 ms. We checked that 3000 ms was sufficient for participants to make a response if they elected to do so. The mean reaction time was 1000 ms (SD = 231.47) for the whole sample, and 1050 ms (SD = 252.61) in the youngest participants (aged 9-12 years). Therefore, participants were able to respond in the allotted time (see Supplementary Notes, Reaction time across stimulus repetitions by age, and Figures S13-14 for additional analysis of reaction times). Participants started the task with 10,000 points and could theoretically obtain final scores between 51,010 and −31,010, although the maximum score obtainable through learning (rather than 'lucky guesses') was 46,909. Since scores below zero could only be obtained by systematically responding to punishment instead of reward, participants with scores below zero points were excluded (see Participants above, and Figures S15-16 for more information on exclusions and the final sample).

### Model fitting and comparison procedure
Seven different reinforcement learning models were constructed. For each model, rewards were coded as 1, neutral outcomes (when no response was made) as 0, and punishments as −1. Expected values for 'go' and 'no-go' actions were initialised at 0 at

the task outset, which is the midpoint between the possible outcomes 1 and −1 and reflects participants' initial lack of knowledge about stimuli values. (An exception to this was for models with an initial action initiation bias; see below). First, we constructed a basic reinforcement learning model, in which learning was captured by a single learning rate ($\alpha$) parameter and a temperature parameter $\beta$, which captures noisiness in responding. In this model, the expected value $V$ of a response on trial $t$ is updated with a reward prediction error $PE$ scaled by the learning rate $\alpha$, where the prediction error is the discrepancy between the outcome $r$ (1, 0, or −1) and the expected value:

$$\text{If go}: V_{(t+1)} = V_{(t)} + (\alpha * PE_{(t)})$$
$$\text{If no} - \text{go}: V_{(t+1)} = V_{(t)}$$

where

$$PE_{(t)} = r_{(t)} - V_{(t)}$$

(1): basic model

The expected values are then converted to response probabilities using the Softmax equation, where the temperature parameter $\beta$ adds noise:

$$\text{Probability of observed response} = e^{Vgo(t)/\beta} / (e^{Vgo(t)/\beta} + e^{Vnogo(t)/\beta})$$

(2): softmax

Using the model comparison procedure illustrated in Fig. 7, we constructed six further models with combinations of additional parameters. These parameters were separate learning rates for reward versus punishment outcomes (3), two versions of an action initiation bias towards responding regardless of anticipated outcome (4–5), and one or two magnitude sensitivity parameters, which accounted for sensitivity to the actual point value obtained (6–7).

$$\text{For reward outcomes}: V_{(t+1)} = V_{(t)} + (\alpha_r * PE_{(t)})$$
$$\text{For punishment outcomes}: V_{(t+1)} = V_{(t)} + (\alpha_p * PE_{(t)})$$

(3): two learning rates

For models that included the initial 'go' bias, the starting value of responding to each object was increased (or decreased) by an amount $b\_i$ on the first presentation of the object only:

$$V_{(1)} = b\_i$$

(4): initial 'go' bias

For models that included the constant 'go' bias, the value of responding to each object was increased (or decreased) by an amount $b\_c$ on each presentation of the object:

$$V_{\text{biased}(t)} = V_{(t)} + b\_c$$

(5): constant 'go' bias

$V_{biased}$ was used only to calculate the response probability for the current trial, so that the bias did not accumulate over repeated presentations of the object.

For models that included a single magnitude sensitivity parameter, the absolute point score obtained on each trial (re-scaled to be between 0 – 1) was multiplied by a magnitude sensitivity parameter $\rho$ and added to the outcome (which was itself still coded as 1, 0, or −1):

$$\text{Outcome}_{(t)} = r_{(t)} + \text{magnitude}_{(t)} * \rho_{(t)}$$

(6): magnitude sensitivity parameter

Finally, models that included two magnitude sensitivity parameters applied different magnitude sensitivities to reward and punishment outcomes:

$$\text{If reward}: \text{Outcome}_{(t)} = r_{(t)} + \text{magnitude}_{(t)} * \rho_{r(t)}$$
$$\text{If punishment}: \text{Outcome}_{(t)} = r_{(t)} + \text{magnitude}_{(t)} * \rho_{p(t)}$$

(7): two magnitude sensitivity parameters

Model fitting and comparison were conducted in MATLAB 2019b (TheMathWorksInc[56]). We used an iterative maximum a posteriori (MAP) approach for all model fitting, in line with previous work using reinforcement learning models[29–31,34]. This procedure computes the maximum posterior probability ($PP_i$) estimate obtained with parameter vector $h_i$, given the observed choices and given the prior computed from group-level Gaussian distributions over the parameters with a mean vector $\mu$ and standard deviation $\sigma^2$. This is a conservative approach whereby any differences in resulting parameters can be seen as robustly capturing latent differences across people, for example across ages. It is ideally suited for studies of reinforcement learning where group-level estimates can improve the reliability of the resulting fitted parameters[57]. First, we initialised Gaussian distributions as uninformative priors with a mean of 0.1 (plus noise) and variance of 100. Next, during the expectation step, we estimated the model parameters for each participant using maximum likelihood estimation (MLE), calculating the log-likelihood of the participants' set of responses given the model being fitted. We then computed the maximum posterior probability estimate, given the participants' responses and the prior probability from the Gaussian distribution, and recomputed the Gaussian distribution over parameters during the maximisation step. These alternating expectation and maximisation steps were repeated iteratively until convergence of the posterior likelihood, or for a maximum of 800 iterations. Convergence was defined as a change in the posterior likelihood of less than 0.001 between successive maximisation steps (see[31] for full details). Bounded free parameters were transformed from the Gaussian space into native model space using link functions. We used a sigmoid function to bound learning rates between 0 and 1, an exponential function for the temperature parameter to ensure positive values, and a hyperbolic tangent transfer (tansig) function for the action initiation bias to allow the parameter to have positive or negative values consistent with previous modelling of the action initiation bias[25].

To compare models, we used Laplace approximation of log model evidence (more positive values indicating better fit[58]) in a random-effects analysis using spm_bms[59] from SPM8 (www.fil.ion.ucl.ac.uk/spm/software/spm8/). This calculates the exceedance probability, i.e., the posterior probability that each model is the most likely. An exceedance probability over 0.95 provides strong evidence for the best-fitting model. We also calculated the integrated BIC score ($BIC_{int}$) for each model, which penalises more complex models. Lower $BIC_{int}$ scores indicate better performance. MATLAB code for models and model fitting and comparison procedures is available at https://osf.io/d2zp4/.

In addition to the main model fitting and comparison procedure with the full sample, we also repeated the procedure separately for participants aged 9-12 years, those aged 13–15 years, and those aged 16–18 years, to confirm that the same model won across the age range (see Figure S6). An additional control model, with a declining rather than constant action initiation bias, underperformed relative to our winning model and was therefore not considered further (see Supplementary Notes, Declining action initiation bias model).

### Parameter recovery and model identifiability

We used a parameter recovery procedure to ensure that the parameters from the winning model were dissociable from each other,

and a model identifiability procedure to ensure that the reinforcement learning models were dissociable from each other[29]. For the parameter recovery procedure, we simulated participant response data only for the winning model, using a range of parameter values between the minimum and maximum values for that parameter. Parameter values were selected from a vector for each parameter. The vectors consisted of 10 equal steps between the minimum and maximum values that were observed for that parameter when modelling the real participant data, with some deviation for noise. This procedure created a grid of all possible values for each parameter to be used for recovery. Data were simulated for 10,000 synthetic participants. The winning model was then fitted again to its simulated data using the MAP procedure, and correlations between the parameters used to simulate the data and the recovered parameters (estimated from the simulated data) were checked for correspondence. For the model identifiability procedure, we simulated participant response data for each model in turn, using a range of parameter values within the observed range from the real data. For each of these models, the full set of seven models was then fitted to the simulated data from that model, using the MAP procedure, and this was repeated 10 times. We then created confusion matrices for mean exceedance probability and for the number of times each model won, to check that for each model and its simulated data, the winning model was the one that had been used to generate the data. This procedure confirms that each model is reliably associated with a different pattern of responses from the competing models.

We also generated synthetic behavioural responses using our winning model and its median parameter values, to check that the real and simulated responses were broadly similar. Behavioural responses for 742 synthetic participants were generated using the whole-sample median value for each parameter, and then plotted for comparison with the actual observed behavioural responses from the 742 (non-synthetic) participants. We then repeated this procedure for the winning model using stimulus repetitions 2–10 only, and separately for participants aged 9-12, 13–15, and 16–18 years to check that our winning model captured behaviour adequately across the sample's full age range. Finally, as an additional test of the validity of our winning model, we conducted correlations between task performance (number of overall correct responses and correct responses for reward and punishment separately) and each model parameter (Spearman's correlations, R's correlation package cor_test function).

## Statistical analysis

All statistical analyses were conducted in R (v. 4.1.1 and v. 4.1.2) through RStudio. First, we investigated associations between age or pubertal stage and the model parameters from the winning model. Since parameter values were not normally distributed, we used robust linear mixed effects regression models using the rlmer function in R. We tested whether each parameter was predicted by age, with IQ and sex as covariates (fixed effects) and varying intercepts for different sites of data collection (random effects). We then checked for quadratic associations with age by adding an $age^2$ term to each model. Discrete variables were recoded so that contrasts summed to zero, and continuous variables were z-scored.

To confirm these learning effects matched participants' behavioural responses, we next used nested linear mixed effects models to assess whether age was related to participants' changing responses to reward and punishment stimuli over the course of the task. These analyses were conducted using R's lme4 package glmer function[60]. Participants' responses were coded as 1 (active response) or 0 (no response) and were predicted from age, sex (0 = male, 1 = female), object repetition number (1-10), and object valence (0 = reward, 1 = punishment) (fixed effects), with varying intercepts

allowed for responses grouped by participant nested within site (random effects). All continuous variables were z-scored, and discrete variables (participant response, sex) were recoded so that the two levels summed to zero (e.g., 0 and 1 becomes −0.5 and 0.5). The same analysis was then repeated for pubertal stage, using PDS score as the dependent variable instead of age. In all analyses, IQ, and sex were included as covariates. The strength of null effects was interpreted using Bayes factors calculated with the BIC method[33] and R's lme4 BIC function[60], using standard priors, and the language suggested by Jeffreys[61].

## Reporting summary

Further information on research design is available in the Nature Portfolio Reporting Summary linked to this article.

## Data availability

The data included in this study were collected as part of the FemNAT-CD project[45]. Raw and processed data have been deposited in the OSF repository and are available at https://doi.org/10.17605/OSF.IO/D2ZP4.

## Code availability

MATLAB code for models and model fitting and comparison procedures is available at https://doi.org/10.17605/OSF.IO/D2ZP4.

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

## Acknowledgements

R.P was supported by an ESRC post-doctoral fellowship award (ES/V011324/1). P.L was supported by a Medical Research Council Fellowship (MR/P014097/1 and MR/P014097/2), a Sir Henry Dale Fellowship funded by the Wellcome Trust and the Royal Society (223264/Z/21/Z), and a Jacobs Foundation Research Fellowship. S.A.B.D was supported by an ESRC grant (ES/V003526/1). The FemNAT-CD project was funded by the European Commission under the 7th Framework Health Program, Grant Agreement no. 602407. We are grateful to all our participants and their families, to other members of the FemNAT-CD project, and to Jo Cutler, Anthony Gabay, Tobias Hauser, Marco Wittmann, and Stefano Palminteri for helpful discussions and advice.

## Author contributions

R.P. conducted all analyses and wrote the manuscript. P.L. assisted with coding for the computational modelling analyses, contributed to manuscript preparation, and provided guidance and oversight on all aspects of the analyses. G.K adapted the learning task for use in this study. G.K. and I.B. collated the learning task data and conducted preliminary data pre-processing and quality checks. R.P., G.K., and J.C.R. collected the data. M.K.-F. assisted with computational modelling. D.D., R.D., G.F., A. F.-R., B. H.-D., A.H., K.K., A.P., C.S., C.M.F., and S.A.D.B. were responsible for the original study design of the FemNAT-CD45 project and for overseeing data collection. All authors read and approved the final manuscript.

## Competing interests

C.M.F. receives royalties for books on attention-deficit/hyperactivity disorder and autism spectrum disorder. She has served as consultant to Desitin and Roche. No other authors report any conflicts of interest.
