## [Peer Review File · Nature Communications]

REVIEWER COMMENTS

Reviewer #1 (Remarks to the Author):

This is an interesting paper on the development of reinforcement learning in adolescence. The paper goes beyond previous work by including a relatively large sample of adolescents. And including a previously developed tasks that can also measure response biases. The results are interesting and deepen our knowledge of how learning develops. I do have a few questions about the modeling choices, which is central to the paper.

The model fitting and comparison is really well-done, including the parameter and model recovery. It does also reveal that the rho parameter does not really recover really well. It may be significant but the effect size may also not be considered relevant at .1? This also makes the superior model fit of 5 over 6 less convincing. Overall it would be good to the parameter distributions for all parameters.

In the model recovery it is very salient that the winning model is performing so much better compared to the other models. Do the authors have any idea why? Also I was surprised to see that for instance model 5 shows rather terrible recovery, and that often model 6 fits model 5 simulated data better. One could see how model 6 could completely mimic model 5 with a linear reward magnitude function, but how it could better fit that data given that the model in all respects is the same but at the same time also gets punished by an additional parameter? Maybe I am missing something but my intuition is that this actually analytically not possible, so maybe it is worth checking the overall recovery procedure?

In general, the supplement and methods could provide a bit more detail on the fitting and recovery. For instance, which parameter bounds are used (if any)? And from what type of distributions are parameters drawn from for the simulations. At what Q-values were the models initialized, and how does that fit the instructions of the task?

What seemed to be missing from the model analyses is a nice overview of the posterior predictives, how well does the model predict the behavior on the task? Overall it would be nice to see how people (of different ages) perform and how the model does or does not predict this. Maybe you could show the model predictions like you do for different ages in Figure 5? After writing this I did find something step by step:

This in methods" We also generated synthetic behavioural responses using our winning model and its mean parameter values, to check that the real and simulated responses were broadly similar."

But this did not point to the supplement where I finally found it. This is such a key to model evaluation that this should really be in the main text. Also I would not use mean values given the very skewed parameter distributions, in that case median seems the better option. This may also help you with the reward learning part, because in my opinion this does not "broadly mimics real behaviour" as is claimed in the figure legend. If that does not help, I would re-consider you evaluation model. Also like you do in the rest of the paper, you should show this for the three age bins separately.

Also here it states that for the model recovery you only have 125 synthetics subjects left. I was compelled by the large sample size in the paper and the argument that smaller samples could lead to unstable results between different studies. So the question is, why such small samples in these steps, specifically because these subjects are free. Of course there is computational time, but that could not be so much?

" For the parameter recovery procedure, we simulated participant response data only for the winning model, using a range of parameter values between the minimum and maximum possible values for that parameter. Data were simulated for 243 synthetic participants."

Can more details be provided here ? What were max and min values? were these theoretically or empirically determined? Were these values drawn from uniform distributions? Why only 243 (very specific but also low number) synthetic subjects, and did you take into account the correlational structure (correlation between parameters) when constructing the 243 subjects? If not you might have created parameter combinations that never occur in the real data.

I start to wonder whether it is possible with your data to get some insights whether the winning model is really the winning model for each age group. Of course age is continuous but if you would do the model fitting for the three groups you now used, would it really show the same winning model? Because if not, comparing parameters of a model that does not equally well describe behavior of each group is problematic. It would be good if the authors can show something in this direction.

This also made me wonder if the current approach is the best. It seems the MAP procedure takes the parameter distribution of the whole group and then uses this in iterative steps to re-estimate the parameters for each individual. What you assume here is thus that all participants have parameters

values that are generated by the same generative distribution. And this procedure will pull parameters to the center of mass of that distribution. However, your hypotheses are that there will be age differences in these parameters. So in that sense your current fitting procedure goes against your main hypotheses. The solutions seems to have age in the hierarchical structure that you are fitting. Please clarify your motivation for how the model was set up and consider other options.

Finally, I do like the task because it has the go, no-go, element in it, I like that there is also an amount of points participants start with, this really helps feel losses as losses, the differences in reward size do not seem particularly helpful. Overall this makes the task similar but also very different from previous studies, and therefore I think it makes an interesting contribution to the existing literature on learning but I do not see how it can or does help providing a framework for unifying the diversity of results.

Some smaller points

“For the analyses of model parameters and age, we excluded 41 participants with values more than three standard deviations from the mean on one or more model parameters.”

please move this statement to main text and report how this effects the analyses. Was this a pre-registered exclusion criterium?

“This payment was not linked to task performance.”

please move to main text. Would you think this affects participants of different ages differently?

Smaller things, but would you think it is a good idea to include test-site as a variable in your models?

Can you explain in more simple words what the stopping criterium is for the iterative MAP procedure? Which measure is used in each step and what it is compared too.

Reviewer #2 (Remarks to the Author):

The goal of this study was to examine reward and punishment learning and action initiation (impulsivity) in children and adolescents. Strengths of the study include a strong theoretical rationale, large sample size (n=700+ from 11 countries) and computational modelling approach. Areas of concern include lack of clarity regarding data exclusion, lack of detail regarding demographics of participants whose data were included in the analyses, and interpretations of the results.

It is surprising that key adolescent learning papers are not cited, including Peters & Crone, 2017 and Davidow et al 2016.

The authors did not include data from 96 adolescents who responded to fewer than half of the reward trials (i.e. trials where responding was the correct behavior). How can the authors be certain that they did not inadvertently remove meaningful learning (i.e. non-learning) data? Do the results differ significantly if these 96 participants are included?

It is also curious that the final sample consisted of twice as many girls as boys. Was the majority of data that was excluded from boys?

Was there a bias in terms of age for excluded data? (i.e. was the likelihood of not including data higher among younger participants?). The authors found that "reward learning rates did not differ significantly by age" but if the younger participants' data who did not understand the task was excluded, then only the younger participants who were more cognitively sophisticated would have been included; this would, unsurprisingly, lead to a seemingly (and perhaps not representative) lack of age difference.

How many participants were included in each of the 5 pubertal groups?

The current study finds stability in reward learning rates across adolescence. In the Discussion the authors claim that this result is in contrast to findings showing that adolescence is a period of heightened sensitivity to reward. However, this is not a reasonable comparison because the majority of studies that report heightened reward sensitivity/learning in adolescence compared adolescents to age-distinct children and adults rather than across the period of adolescence as is done here. This is further confounded by the possibility that the majority of data that was excluded is from the

younger (e.g. 9, 10, 11 year olds) participants, then the age range included in the analysis is much more narrow than in the previous studies that found a quadratic effect.

Please provide a more clear description of how action initiation bias was measured.

The authors fail to describe the nuances in the 11 countries included—how do these countries differ in a manner that might have impacted the findings?

It is unclear if country was included as a covariate.

An explication of the relevance and implications of these findings to real-world adolescent decision-making would be helpful to the reader.

The authors present a study on a large developmental sample that suggests that behavior is not only guided by reinforcement learning (i.e., effects of previous outcomes on future behavior), but also by an “action initiation bias” (i.e., the tendency to respond [“go”] rather than withhold a response [“nogo”]), and that this action initiation bias changes with age. These findings might have implications for clinical research, given that the action initiation bias might be related to impulsivity, and both reward learning and impulsivity have been argued to underlie clinical conditions, especially during adolescence.

The study has several strengths, including the large sample size and the clear language in describing the results. However, it also has limitations that need to be addressed. Most prevalently, the computational model needs to be improved because it does not appropriately capture human behavior, which is most evident in the fact that simulated behavior does not capture the crucial aspects of human behavior, i.e., a failure in qualitative model fit. Previous computational models have provided better fits for this task, and the authors could use these as inspiration to improve their model.

There also might be issues in terms of the model fitting procedure, which lead to a wide spread of estimated parameters across participants and low parameter recovery (lowest with $r=0.1$). Further, the action bias parameter is commonly used in similar tasks and models (e.g., Guitart-Masip et al., 2021), and I would recommend that the authors state that more clearly. It sometimes sounds like this study is the first to introduce the action bias parameter, but the novelty of this study seems instead to be the application of a known model to a large developmental sample.

Second, the behavioral analyses of the paper could be strengthened, e.g., by conducting more specific tests that relate to the hypotheses. The interpretation of some statistical models also needs to be refined, and some claims that are not currently backed up by tests could be.

From a presentation perspective, the paper sometimes seems to imply that because the results of this study are not the same as the results other labs have found in previous studies, the results of previous studies might be faulty. I would recommend providing more nuance when comparing this study to previous ones. For example, specific parameter values are often a function of the underlying task (e.g., fitted reward learning rates are likely higher in a task in which rewards are informative than in a task in which rewards are not informative, reflecting participants’ adaptive strategies). In other words, a previous study with a different result does not necessarily contradict the current results because both reflect the specific task used.

I provide more details for each of these points below:

Major

About the task / dataset

My main concern was if participants—especially the youngest ones—fully understood the task. Could the authors provide more details about the training procedure, including instructions given to participants and whether participants received any test trials before the main experiment? What did participants know about the task beforehand? E.g., did they know it was deterministic, and that there was an equal number of go-reward and avoid-punishment objects?

I am particularly interested in this because the data show a sudden increase in “go” responses between the first and second repetition number (I am pasting the behavioral plot below). The fact that this increase is present both for reward and for punishment trials implies that this was not based on task feedback, and participants might instead still be learning how to do the task: For example, in the first iteration, participants might have been relatively cautious and avoided potential negative outcomes by inhibiting “go” responses. But by the second repetition, they might have realized that they can only obtain feedback—i.e., the crucial learning information—if they choose the “go” response. Alternatively, participants who are unfamiliar with the specific task might simply have missed more trials on the first repetition, which would be counted as “nogo” responses, and explain the lower proportion of “go” responses specific to the first iteration.

In both cases, younger participants might be more affected than older participants. Do the authors have evidence that 2.5 seconds was enough time for the youngest participants? For example, were more younger than older participants excluded due to a lack of task performance or understanding? Another argument for a lack of task understanding was the fact that older

participants gained more points than younger participants overall, as you show using the robust linear mixed effects regression.

I was also wondering if the youngest participants already understood the large numbers used for this study, which are in the thousands. I am wondering at what time children typically learn these numbers at school, and whether children who have not covered them yet might process the feedback differently, due to a lack of understanding what it meant. (Could behavior be plotted separately for children who have not covered large numbers yet in school, compared to those who have, compared to adults?) For example, the lack of differences in learning speed for stimuli with different reward values suggests that participants either ignored that information, or were unable to process it. Especially the interesting pattern with learning being fastest for the 700-point stimulus, compared to both 1400 and 2000, made me wonder if younger participants thought 700 is more than 2000 because 7 is more than 2? To see if this might be the case, the authors could, e.g., plot the average “mean response” of repetitions 5-10 over age. This would show if the youngest participants indeed thought 700 is better than 2000, and might also reveal whether older participants were more sensitive to the actual numbers.

The data suggest that participants did not treat values as a continuous variable: The figure below shows that all the negative values tend to cluster together, that “1” seems to be treated separately, and the positive values are quite mixed. Can the authors discuss potential factors that relate to this finding? (To help interpret the figure, it might also be helpful to use a continuous color scheme; maybe neutral in the middle, red for positive values, blue for negative ones).

The dataset consists of 491 girls and 251 boys. Given that there were sex effects on behavioral measures, I was wondering if the sex composition differed between different ages, and might explain some age-related effects. I was also wondering if the country of origin affected the behavior. E.g., did the authors conduct ANOVAs to test for differences in testing sites?

From a purely methodological perspective, I was curious to know why the authors chose to not display any feedback after “nogo” trials, rather than, e.g., saying “You have not gained or lost any points,” for consistency with the visual feedback of “go” trials?

Behavioral analyses

From a purely rhetorical point of view, I would suggest presenting the behavioral results before the modeling results, to base the interpretation of the more abstract parameters on a solid understanding of participant behavior.

I was also missing more in-depth behavioral analyses. For example, I was interested whether participants showed win-stay-lose-shift behavior (i.e., a higher tendency to repeat the previous choice for a given stimulus when that choice was previously rewarded, and higher tendency to switch the choice when it was punished). In this case, because only the “go” choice leads to rewards and punishments, the analysis could compare participants' stay versus switch choices in the conditions “previous go and reward”, “previous go and punishment”, and “previous nogo”.

I was also interested in whether response times could shed additional light on task behavior. For example, do the authors observe response-time slowing after punishment outcomes? How do response times change over repetition? And over age?

On a different note, it appears that the main difference between the reward and punishment conditions seems to be that go responses stay high in the reward condition, whereas they drop in the punishment condition. It would be interesting to see within-participant differences between these two conditions in light of this possibility, and to consider potential implications about cognitive processing (e.g., the authors mention a default strategy of responding “go”, which is solely modified by punishment feedback).

Lastly, I was interested in how the delay length between the presentation of the same stimulus affected performance. For example, was performance better when the same stimulus was presented twice in quick succession (i.e., with only a small number of other stimuli interleaved)? This would suggest that participants held stimuli in working memory, and might suggest the addition of a working-memory or forgetting parameter to the task model.

I was also wondering how raw accuracy on the task developed with age. Could the authors provide a plot that shows that?

Further, there were a few issues with existing analyses that the authors should address:

- The authors state: "... we did not observe a significant pubertal stage by repetition by valence interaction (OR = 1.04 [0.10, 1.07], $z = 1.87$, $p = .06$), suggesting that the punishment-specific improvement in learning was better captured by age than by pubertal stage." It is not valid to claim here that the effect better captured by age than by pubertal stage; if the authors wish to make a claim about specific differences between the effects of age and puberty, both need to be compared directly. The lack of a significant effect in model does not support this claim (e.g., see Makin & Orban de Xivry, 2019)
- The authors state: "... this age-related learning improvement was specific to learning from punishment outcomes (age by repetition by valence interaction: OR = 1.09 [1.05, 1.13], $z = 4.65$, $p < .001$)." To claim that the improvement was specific to the punishment condition, it would be necessary to see a plot that shows both the punishment and the reward condition and reveals a difference in the means, or to conduct another follow-up test. By itself, the significant interaction could also imply the inverse patterns (effect for reward, but not punishment) or a simple difference in the magnitudes of the effects for both conditions.
- The authors state: "A generalised linear mixed model (GLMM) (predicting correct responses from age, stimulus repetition number, outcome valence, and covariates; see Methods) revealed a significant main effect of stimulus repetition on the number of correct responses made, with performance improving throughout the task (Odds ratio (OR) = 1.19 [1.17, 1.21], $z = 18.56$, $p < .001$). Thus, participants exhibited learning." Did the model show any significant interactions? Interactions always need to be interpreted before main effects (For example, imagine a scenario in which two conditions A and B have average values of "4" before an intervention; after intervention, condition B rose to "20", but condition A stayed at "4"; the main effect of intervention would be significant; however, we would not want to claim that the intervention was overall successful; we need to interpret the interaction between intervention and condition first.) For the current data, potential interactions with age might imply that younger participants exhibited less or no learning. Interactions with outcome valence might mean that there was learning for one, but not the other valence (e.g., plots suggest more learning from punishment than reward).

Lastly, due to the qualitative difference in responding to repetition 1 compared to later repetitions, I am wondering if repetition 1 might be considered part of participants' learning phase, and if it would clean up some results if the analyses were restricted to repetitions 2-10 only. This would likely improve the fits of the GLMs that contain repetition as a predictor because in the current data, the effect of repetition is highly non-linear, but it seems linear after trial 2. It might also help the computational model, which currently does not capture the quantitative aspects of human behavior. Furthermore, reanalyzing data from repetitions 2-10 would provide evidence for the robustness of the current results in the spirit of leave-one-(repetition)-out validation.

Modeling

Model does not capture human behavior adequately

My central concern with the current study is that the model does not seem to capture human behavior adequately. As shown in the plot below from the supplemental material, one of the most noteworthy patterns in participant behavior is the initial increase in “P(go),” which is completely absent from the model behavior. Whereas the model thereafter shows a similar decline in go responses in the punishment condition to humans, it shows a steady increase of go responses in the reward condition, which is absent in the human data. (Indeed, the behavioral analyses show a lack of learning from reward, which directly contradicts the model.) This suggests that the model did not indeed capture human behavior.

Furthermore, the generate-and-recover analyses show poor recoverability for some parameters, especially parameter ρ with a correlation of just $r=0.1$. That the model was not able to recover the underlying parameters of simulated data suggests that it also did not recover the parameters of human data very well. To determine this issue, it would be instructive to see scatterplots of the generated and recovered values of each parameter (e.g., see Wilson & Collins, 2019). It would also be helpful to see a correlation matrix of all the model parameters, to see if parameters traded off with each other within the model. For example, learning rates and decision temperature parameters are known to trade off with each other, with important caveats for model interpretation.

These patterns suggest that the models are misspecified and might have to be parameterized differently to adequately capture human behavior (e.g., Katahira, 2015, JoMP). The extremely large variability of parameter values between participants (despite hierarchical fitting) also supports the interpretation of a lack of model fit and potential model misspecification.

I was also wondering if models fit participants of different ages equally well? If there are age-related differences in model fit, parameter differences over age would be hard to interpret. To answer this question, the authors could plot simulated behavior separately for participants of

different ages, and assess the fit between the model and participants' behavior at each age group. For a more quantitative analyses, they could calculate the NLL for each participant's behavior under the winning model and parameters, and plot it over age.

As for the behavioral analyses, I am wondering what the results would look like if the models were refit to the data starting at repetition 2. This would provide a (very mild) version of split-half reliability (i.e., fitting the model on half the dataset and validating it on the other half). If this leads to very different results, the validity of the current claims should be questioned. To capture the overall patterns better, I think the authors have two choices: If they think the prominent increase in "go" responses between repetition 1 and 2 is a crucial aspect of their dataset, they should aim to create a model that captures this increase. If they think that this sudden increase might be an artifact (e.g., related to a lack of previous training with the task), they could consider removing these trials and fitting the model to the remaining data. However, I don't think the model captures human behavior adequately in the current form.

If this is the case, i.e., if the current model does not provide an adequate fit to the behavioral data, the fitted model parameters cannot be interpreted. In other words, even though the current model wins against other competing models in terms of numerical model comparison, as soon as at least two different models are compared, there always will be a winning model, and this does not show that the winning model fits the data adequately. Continued search of the model space will likely lead to a model that fits the data better and whose fitted parameters will be interpretable.

Improvement of the model

There are several ways in which the model could be improved. First, additional parameters could help explain additional variance, and resolve issues of model misspecification (e.g., Katahira, 2015, JoMP). For example, similar tasks that involve a number of different stimuli whose presentation is spread out in time often involve additional memory parameters (e.g., episodic memory: Bornstein; working-memory: Collins), or forgetting parameters (e.g., Xia & Collins, 2021). If the authors find that the delay between presentations of the same stimulus explains variance in behavior (as suggested above), then adding a forgetting parameter will likely improve the fit of the computational model.

I was also wondering how the go bias parameters in the current model compares to the more common persistence / choice stickiness / perseverance parameter, which increases the chances of selecting the action that was shown on the previous trial. It seems like the go bias parameter is the same as a typical persistence parameter, except that it only applies to go trials, and that it is always active, rather than just when the previous action was also a "go" action. I am wondering if a more traditional persistence parameter would potentially fare better in the current dataset. This idea is especially tempting given that most behavioral studies have found higher proportions of stay trials in children compared to adults behaviorally (and more persistence computationally). Have the authors explored this possibility?

I was wondering if the initiation bias was related to a behavioral measure of staying?

I was also wondering what the optimal strategy was for the current task, and whether there was a way to quantify how close participants were to this optimal strategy? Apparently, because feedback is deterministic, an optimal strategy would set both reward and punishment learning to the maximum values of 1. (Why are participants' parameters so different?)

And what is the optimal setting of the go parameter? It seems like it should be 1 at the first iteration, so that the outcome of each stimulus can be observed once (and then memorized); after the first repetition, the go bias should then drop to 0, so that the correct action can be chosen consistently and without bias. If indeed, the optimal strategy requires an adjustment of the go bias over time, this might be a valid mechanism to integrate into the model.

Also related to the go bias parameter, it seems like the main function of this parameter is to shift the balance between go/nogo toward more go responses. I was wondering if other options had been explored to capture this effect? For example, it might be more succinct to think of this shift as a shift of the entire softmax function.

Given the low recoverability of the current parameter ρ , the authors might consider either removing it from the model (and it would be interesting if doing so would improve the recoverability of other parameters as well), or modeling reward magnitude sensitivity differently. For example, it seems like negative outcomes cluster together, positive outcomes cluster together, and outcomes of 1 are distinct. Would recoding the outcomes in this way improve the model fit?

Many studies have also reported that learning rates change over time (e.g., Jepma, 2020) or depending on previous trial outcomes. Have the authors explored this possibility in the current study?

Overall, I was wondering how the authors determined the initial search space for their model, and the list of parameters to include. Was this based on a literature search of computational models for similar tasks? E.g., how is the current model related to the model in Guitart-Masip et al., (2012), which seems to provide a better fit to participant data in a similar task (see below)?

Fig. 2. Observed and modeled behavioral performance. (A–D) Learning time courses for all four conditions. Each row of the raster images shows the choices of one of the 47 subjects in each of the four conditions. Go responses are depicted in white and no-go responses are depicted in grey. The overlaid black lines depict the time varying probabilities, across subjects, of making a go response. The colored lines show the same time-varying probabilities, but evaluated on choices sampled from the model (see Materials and methods). (E) Mean percentage of correct responses in each of the four conditions. Green error bars depict the 95% confidence interval (CI) and the red error bars depict standard error of the mean (SEM). Post hoc comparisons were implemented by means of repeated measures *t*-test: **p* < 0.005. (F) Integrated Bayesian Information Criterion (BIC) score for all models tested. All models are modified Q-learning model with two pairs of action-values (go and no-go) for each state (fractal image). The winning model includes as free parameters a learning rate, a slope of the softmax rule, irreducible noise, a constant bias factor added to the action-value for go, and a Pavlovian factor that adds a fraction of the current state value to the action-value for go.

I am also wondering if the go bias might change over time rather than be a stable value, given that the ratio of go / nogo actions declines with repetitions. Have the authors tested this possibility?

The authors also note that “one consideration is whether action initiation biases are themselves influenced by the prospect of a rewarding outcome, since there are forms of impulsivity that occur specifically in situations where a possible reward is anticipated.” I think this is a very interesting hypothesis—has this been tested in the model? E.g., the prospect of a rewarding outcomes seems to be the value (i.e., the expected long-term cumulative reward) of the “go” action. In other words, the authors hypothesize that the go bias is a function of action value. This could be explored both behaviorally and computationally.

Going beyond individual parameters, I was wondering if the structure of the model might have to be adapted to capture human behavior better. For example, the authors use a classic RL update when participants choose the “go” action, but no update is performed after the “nogo” action. In words, this model assumes that participants do not learn in “nogo” trials. I was wondering if the model space could be explored more to establish whether this assumption is true. To explore this idea, the authors could first determine whether participant behavior still changes even when participants choose the “nogo” action; if so, the model could capture behavioral changes, e.g., by including an update based on the outcome “0”.

Model fitting

I also have a few concerns regarding the model fitting procedure. For one, individual differences in fitted parameters are very large, and I was wondering if full hierarchical Bayesian fitting would work better than the iterative approximation chosen here. Why did the authors choose the hierarchical expectation maximization approach? There is now a huge literature that shows that models improve tremendously when fitted hierarchically (e.g., Lee, 2011, JoMP; Katahira, 2016; JoMP), and empirical evidence is steadily growing as well (e.g., Brown, 2020, Biological Psychiatry; Pratt, 2020). For example, the issues in parameter recoverability might be resolved by choosing this fitting method (e.g., see Eckstein et al., 2022 DCN, supplemental section, comparing ML fit to hierarchical fit). An additional advantage would be that age differences in model parameters could be tested in an unbiased way using this approach.

Furthermore, the issue of large between-participant parameter variability (Fig. 3 and 4), and the need to exclude 41 participants with values more than three standard deviations from the mean, might be alleviated using this approach.

In terms of exposure of results, I was missing a plot that shows raw parameters over continuous age.

Writing / presentation

Relationship to prior work

The authors state that: “very little is known [...] about the robustness of previous findings.” Several studies have come out just in the last year with regard to this question that might resolve the issue, including Nussenbaum & Hartley (2019) and Eckstein et al. (2021, COBS), which address this particular issue for developmental populations, and are already cited in other places. Another paper that addresses this issue in a developmental population is Eckstein et al., (2021, biorxiv). More studies targeting the reliability of RL models more generally include Weidinger & van den Bos (2019), Brown et al., (2020), Shahar et al., (2019), Pratt et al., (2020), and Waltmann et al. (2022)

The authors state: “To our knowledge, only one study to date has measured learning in a task design that incorporates requirements both to learn and also to inhibit actions.” This is slightly misleading as it implies that no study at all has studied this, whereas it only refers to developing populations. E.g., Guitart-Masip et al. have published several studies using a similar task and model.

In a similar vein, the authors state: “There has been a relative paucity of computational modelling work focusing on learning in adolescence” and “reward- and punishment-guided behaviour in adolescence is not well understood.” I would soften these statements. It is unclear at what these behaviors would be “understood”, and there have now been several reviews on

this topic (including Nussenbaum & Hartley), there is a dedicated journal (DCN), and a conference (Flux) that likely provide ample references.

Another concern regards statements that seem to “oversell” computational modeling. For example: “Here, we use computational modelling to distinguish between learning processes (...) and action initiation or ‘go’ biases (...). We test whether these different mechanisms are separable...” Recent work has suggested that computational modeling might not be able to answer questions of this kind (e.g., Eckstein et al., 2021 COBH), and I would encourage the authors to provide more nuance here and in other parts of the paper that share this sentiment.

Similarly, the paper sometimes seems to imply that the action initiation bias should replace previous modeling approaches or that previous studies were problematic because they did not include such a parameter (e.g., “They suggest that theoretical accounts positing heightened responses to reward in adolescence should consider differences in impulsive action initiation rather than reward sensitivity or learning.”). I would encourage the authors to consider that most prior RL studies did not include conditions in which such a parameter could be added (e.g., go versus nogo trials), just like the current study did not include a variety of conditions in which other parameters could be added that have been shown to be critical in prior studies (e.g., attention, episodic memory, model-based RL, working memory, etc.).

I do not see how the conclusions follow from the premises in the following statements:

- “Moreover, for all parameters where we observed differences across development, we saw the same associations when considering pubertal stage. This further suggests that these differences are part of the developmental process, rather than only a reflection of chronological age.” If both models fit the data, how can one be ruled out at the cost of the other?
- “First, in contrast to the classic go/no-go paradigm (where ‘go’ responses are required substantially more often than no-go responses), our task used equal numbers of go-for-reward and no-go-for-punishment trials. This means that ‘go’ responses were not particularly associated with reward in this context.” If “go” is the only response that can possibly lead to reward, why is there no association between go and reward?

Potential for more in-depth discussion

I was also interested to hear why the authors think that the go bias parameter in this study might be related to impulsivity. It seems that a go bias is necessary to perform well in this task (i.e., because only the go action can lead to feedback, it is rational and advisable to systematically prefer the go action over the nogo, especially in the beginning of the task).

In terms of the overall conclusion drawn about the results of this study, it seems inconsistent to state that there is learning from reward when behavioral analyses indicate no learning from reward. Improving the computational model will hopefully resolve this issue, such that behavioral and modeling analyses will lead to the same conclusion.

There were a few section in which I wish the authors addressed the implications of their findings, e.g.:

- “However, the action initiation bias we observed appears to be a genuine action bias, rather than a deliberate strategy or an indirect effect of reward facilitating action. “ How did the authors reach that conclusion?
- “Together, our findings suggest that the tendency to initiate actions and learn from punishment shifts from late childhood across adolescence” Can the authors say more about the mechanism behind this transition?
- “It is plausible that adolescent-onset psychopathologies represent aberrant developmental pathways, in which these normative increases in punishment learning and declines in action initiation biases are disrupted.” It would be interesting to explain more explicitly, or with examples, what the authors had in mind here. Which (neural) pathways? Which psychopathologies?
- “Our findings also demonstrate these associations robustly by testing a large and geographically diverse sample.” I am wondering if testing a large sample automatically implies robust findings. The error bars in model estimates might suggest otherwise. Including participants from many different backgrounds might also induce additional noise. Can the author explain this sentiment a little more?

Minor

In the introduction, can the authors define “action initiation biases”, just like they define RL?

I was wondering why the authors chose to divide participants into 3 age bins, but 5 puberty bins? Do the results depend on the number of bins chosen? It might aid direct comparison to have the same number of bins both analyses.

In the sentence: “Part of this variability could be due to different task demands”, Eckstein et al. (2021, COBS) and Eckstein et al., (2021, biorxiv) might be other relevant citations.

I did not fully understand which methods were used in the following analyses: “To confirm the strength of these associations, and obtain strength of evidence for any null effects, we calculated Bayes factors using the BIC method and linear mixed effects regression models, with age removed from the null model. We observed very strong evidence for the associations between age and punishment learning rate ($BF_{10} = 336.00$, $BF_{01} = 0.003$) and between age and action initiation bias ($BF_{10} = 6545.00$, $BF_{01} = 0.0002$)” Could the authors provide additional explanation? Specifically, how did they test age effects after removing age?

For Figures 3 and 4, I was wondering if the data might be summarized better in terms of the median and bootstrapped confidence intervals, rather than the mean and standard deviation, given that data points are not distributed normally (see Figure 4 below)? This would also make the analysis more resistant to outlier values.

Why does Table S1 use Pearson’s correlation, but table S2 Spearman’s?

This sentence seems to miss a verb? “Internalising problems are likewise associated with difficulties in reinforcement learning, and social media use, which can become problematic for some adolescents, and has recently been linked to reward learning mechanisms.”

“Computational modelling of learning typically uses reinforcement learning models”: There also is a huge (equally-sized?) literature on Bayesian Inference, e.g., Gopnik, Griffiths, Gershman

The above figure should include information about trial timing.

The reference list includes both “Sutton RS, Barto AG. Reinforcement Learning: An Introduction. MIT Press; 2011.” and “Sutton RS, Barto AG. Reinforcement Learning, Second Edition: An Introduction. MIT Press; 2018. “

The following figure might be better served using a continuous color scheme:

Response to Reviewers' comments: "Action initiation and punishment learning differ from childhood to adolescence while reward learning remains stable"

Reviewer #1 (Remarks to the Author):

This is an interesting paper on the development of reinforcement learning in adolescence. The paper goes beyond previous work by including a relatively large sample of adolescents. And including a previously developed tasks that can also measure response biases. The results are interesting and deepen our knowledge of how learning develops. I do have a few questions about the modeling choices, which is central to the paper.

Response: Thank you so much for your positive feedback about our work. We are delighted that you found our manuscript interesting and that it deepened our knowledge of how learning develops. We really appreciate your time reviewing our work and your valuable comments, which have significantly helped us to improve our manuscript further.

The model fitting and comparison is really well-done, including the parameter and model recovery. It does also reveal that the rho parameter does not really recover really well. It may be significant but the effect size may also not be considered relevant at .1? This also makes the superior model fit of 5 over 6 less convincing. Overall it would be good to the parameter distributions for all parameters.

Response: We agree that the recovery of the rho/magnitude sensitivity parameter is low, albeit significant. For this reason, as well as no specific hypotheses about how magnitude sensitivity differs with age, we refrained in the manuscript from interpreting the age differences in magnitude sensitivity that we observed. However, the inclusion of this parameter did nonetheless improve the model fit and the resulting model showed good identifiability, so we accepted model 6 as the best model.

To address the concerns about the low recovery of the rho parameter we have now run new analyses showing that both model 5 (without the rho) and model 6 (with the rho) show the same age-related differences in action initiation/go bias and punishment learning. In addition, given the poor recovery of the rho, we have now omitted it from our final model space and updated our winning model throughout the paper to be model 5 without the rho parameter. The parameters from model 5 show excellent recovery and model 5 also shows good identifiability. The original analysis with model 6 can now be found summarised in Supplementary Table S2 and Supplementary Figure S5. Figure 3 showing recovery and model identifiability has now been updated as below:

Figure 3. Model performance and validation. (a) Exceedance probability for the five computational models that comprised the final model space. The winning model was the $2a\beta b_c$ model, with separate reward and punishment learning rates, and a constant action initiation bias. (b) ΔBIC_{int} , relative to the winning model ($2a\beta b_c$). (c) ΔLME , relative to the winning model ($2a\beta b_c$). (d) Parameter recovery. The confusion matrix represents Spearman correlations between simulated and fitted (recovered) parameters. Each parameter exhibited a significant positive correlation between its true and fitted values, with r values ranging from 0.49-0.92 (shown on the lower diagonal) (e) Exceedance probability from the model identifiability procedure. The diagonal represents the probability of each model having the best fit to its own synthetic data. The winning model ($2a\beta b_c$) was highly identifiable from other models. (f) Number of runs where each model was selected as the best fit for data generated by each model in the model identifiability procedure. The diagonal represents the number of runs each model was selected as the best fit for its own data. The winning model ($2a\beta b_c$) was the best fit to its own data on all 10 runs.

We have now also included further information about parameter distributions:

“First, we initialised Gaussian distributions as uninformative priors with a mean of 0.1 (plus noise) and variance of 100. Next, during the expectation step, we estimated the

model parameters for each participant using maximum likelihood estimation (MLE), calculating the log-likelihood of the participants' set of responses given the model being fitted. We then computed the maximum posterior probability estimate, given the participants' responses and the prior probability from the Gaussian distribution, and recomputed the Gaussian distribution over parameters during the maximisation step. These alternating expectation and maximisation steps were repeated iteratively until convergence of the posterior likelihood, or for a maximum of 800 iterations. Bounded free parameters were transformed from the Gaussian space into native model space using link functions. We used a sigmoid function to bound learning rates between 0 and 1, an exponential function for the temperature parameter to ensure positive values, and a hyperbolic tangent transfer (tansig) function for the action initiation bias to allow the parameter to have positive or negative values consistent with previous modelling of the action initiation bias²⁵."

In the model recovery it is very salient that the winning model is performing so much better compared to the other models. Do the authors have any idea why?

Response: Thank you for this observation. The model identifiability is based on the exceedance probability and although the winning model clearly won when examining the exceedance probability ($XP = 1$), it was very similar to other models in some of the other fit statistics. Since the exceedance probability most clearly distinguished between different models in the main model fitting procedure, it also showed model 6 was the best for the model identifiability. However, in response to your previous point, we have now updated the winning model to be model 5 throughout, and show that model 5 has good identifiability.

Also I was surprised to see that for instance model 5 shows rather terrible recovery, and that often model 6 fits model 5 simulated data better. One could see how model 6 could completely mimic model 5 with a linear reward magnitude function, but how it could better fit that data given that the model in all respects is the same but at the same time also gets punished by an additional parameter? Maybe I am missing something but my intuition is that this actually analytically not possible, so maybe it is worth checking the overall recovery procedure?

Response: Thank you for this suggestion. We have carefully checked our code again and confirm that it is correct. In the original manuscript, although model 6 did sometimes fit model 5 better than model 5's fit to itself, model 6 was still not being chosen as the best fit for model 5 very often. However, in line with your earlier comment, we have updated our model space to remove model 6 due to the low recovery of the rho parameter. An updated model identifiability shows good identifiability for model 5. We have updated the figures in the manuscript as shown above in response to your previous comment.

In general, the supplement and methods could provide a bit more detail on the fitting and recovery. For instance, which parameter bounds are used (if any)? And from what type of distributions are parameters drawn from for the simulations. At what Q-values were the models initialized, and how does that fit the instructions of the task?

Response: Thank you for this suggestion. We have now added additional information about the parameter space, distributions and fitting procedure, and Q values under **Model fitting and comparison procedure:**

“Expected values for ‘go’ and ‘no-go’ actions were initialised at 0 at the task outset, which is the midpoint between the possible outcomes 1 and -1 and reflects participants’ initial lack of knowledge about stimuli values. (An exception to this was for models with an initial action initiation bias; see below).”

“First, we initialised Gaussian distributions as uninformative priors with a mean of 0.1 (plus noise) and variance of 100. Next, during the expectation step, we estimated the model parameters for each participant using maximum likelihood estimation (MLE), calculating the log-likelihood of the participants’ set of responses given the model being fitted. We then computed the maximum posterior probability estimate, given the participants’ responses and the prior probability from the Gaussian distribution, and recomputed the Gaussian distribution over parameters during the maximisation step. These alternating expectation and maximisation steps were repeated iteratively until convergence of the posterior likelihood, or for a maximum of 800 iterations. Bounded free parameters were transformed from the Gaussian space into native model space using link functions. We used a sigmoid function to bound learning rates between 0 and 1, an exponential function for the temperature parameter to ensure positive values, and a hyperbolic tangent transfer (tansig) function for the action initiation bias to allow the parameter to have positive or negative values consistent with previous modelling of the action initiation bias²⁵.”

Under the heading **Parameter recovery and model identifiability**, we have added more information about the selection of parameter values:

“Parameter values were selected from a vector for each parameter. The vectors consisted of 10 equal steps between the minimum and maximum values that were observed for that parameter when modelling the real participant data, with some deviation for noise. This procedure created a grid of all possible values for each parameter to be used for recovery.”

What seemed to be missing from the model analyses is a nice overview of the posterior predictives, how well does the model predict the behavior on the task? Overall it would be nice to see how people (of different ages) perform and how the model does or does not predict this. Maybe you could show the model predictions like you do for different ages in Figure 5? After writing this I did find something step by step: This in methods" We also generated synthetic behavioural responses using our winning model and its mean parameter values, to check that the real and simulated responses were broadly similar."

But this did not point to the supplement where I finally found it. This is such a key to model evaluation that this should really be in the main text. Also I would not use mean values given the very skewed parameter distributions, in that case median seems the better option. This may also help you with the reward learning part, because in my opinion this does not "broadly mimics real behaviour" as is claimed in the figure legend. If that does not help, I would re-consider you evaluation model. Also like you do in the rest of the paper, you should show this for the three age bins separately.

Response: We apologise for this omission. We have now reported these analyses in the main manuscript rather than the supplementary materials, and we have conducted the analysis separately for the three age groups as suggested, using the updated model 5 as in response to your other comments. These new analyses show good prediction of behaviour that is reasonable across all three age bins. As we note throughout the manuscript, all of our analyses are based on continuous associations with age rather than discreet age bins, which are only used for visualisation, so these graphs should be interpreted with some caution. We have also increased the number of synthetic participants to match the sample size of our tested participants, which has also contributed to improved simulated behaviour based on our winning model. Thank you also for the suggestion to use median instead of mean values, which we have now implemented. Together, these new analyses robustly show the suitability of our selected winning model and more closely mimics real participant behaviour.

The text under **Methods: Parameter recovery and model identifiability** has been updated as follows:

“We also generated synthetic behavioural responses using our winning model and its median parameter values, to check that the real and simulated responses were broadly similar. Behavioural responses for 742 synthetic participants were generated using the whole-sample median value for each parameter, and then plotted for comparison with the actual observed behavioural responses from the 742 (non-synthetic) participants. We then repeated this procedure separately for participants aged 9-12, 13-15, and 16-18 years to check that our winning model captured behaviour adequately across the sample’s full age range.”

The following section and figure have been updated and moved from the supplementary materials to the main manuscript as suggested:

“Winning computational model tracks learning from reward and punishment

To further confirm the accuracy of our model, we generated synthetic behavioural data for 742 ‘participants’ using the winning model and its median parameter values. We then repeated the procedure separately for participants aged 9-12, 13-15, and 16-18 years to ensure that the model captured behaviour adequately across the full age range. In both the 742-participant sample and the aged-based samples, the simulated responses fell within a similar range to the real responses and, especially on punishment trials, followed a broadly similar trajectory (Figure 6). Thus, the learning rates observed in the winning model are compatible with the behavioural evidence of learning we observed in the participants’ response patterns.”

Figure 6. Simulated probability of ‘go’ response to reward and punishment stimuli across 10 stimulus repetitions. (a) Simulated and real probability of ‘go’ responses on reward trials, with simulated data generated using the winning model and its median parameter values for the full sample (N = 742). **(b)** Simulated and real probability of ‘go’ responses on punishment trials for the full sample (N = 742). **(c)-(d)** Simulated and real probability of ‘go’ responses on reward **(c)** and punishment **(d)** trials for participants aged 9-12 years. **(e)-(f)** Simulated and real probability of ‘go’ responses on reward **(e)** and punishment **(f)** trials for participants aged 13-15 years.

(g)-(h) Simulated and real probability of 'go' responses on reward **(g)** and punishment **(h)** trials for participants aged 16-18 years.

Also here it states that for the model recovery you only have 125 synthetic subjects left. I was compelled by the large sample size in the paper and the argument that smaller samples could lead to unstable results between different studies. So the question is, why such small samples in these steps, specifically because these subjects are free. Of course there is computational time, but that could not be so much?

Response: Thank you for this query and apologies for the lack of clarity. The 125 synthetic subjects were not for the model identifiability, but for the generation of synthetic responses using the mean parameter values of the winning model. We have now increased the number of synthetic subjects to 742 to match the number in the main sample (see our response to your previous comment).

“For the parameter recovery procedure, we simulated participant response data only for the winning model, using a range of parameter values between the minimum and maximum possible values for that parameter. Data were simulated for 243 synthetic participants.”

Can more details be provided here? What were max and min values? were these theoretically or empirically determined? Were these values drawn from uniform distributions? Why only 243 (very specific but also low number) synthetic subjects, and did you take into account the correlational structure (correlation between parameters) when constructing the 243 subjects? If not you might have created parameter combinations that never occur in the real data.

Response: In response to your previous comments, we have now added additional information on the generation of parameter values. Parameter bounds were theoretically determined, based where possible on previous literature. Parameter values were selected from a vector with a fixed number of steps between the highest and lowest possible values. Due to computational time, we had initially selected a smaller number of steps, resulting in 243 synthetic participants based on the possible combinations of parameters and steps. We have now increased the number of steps to 10 per parameter for the new winning model (model 5), resulting in 10,000 synthetic participants, and we have also used the minimum and maximum observed parameter values from the main modelling when running the parameter recovery, to avoid creating values that never occurred in the real data. Together these changes show good parameter recovery (Figure 3, see above). We have added this information to the modelling section of the manuscript as well as a new figure:

“For the parameter recovery procedure, we simulated participant response data only for the winning model, using a range of parameter values between the minimum and maximum values for that parameter. Parameter values were selected from a vector for each parameter. The vectors consisted of 10 equal steps between the minimum and maximum values that were observed for that parameter when modelling the real participant data, with some deviation for noise. This procedure created a grid of all possible values for each parameter to be used for recovery. Data were simulated for 10,000 synthetic participants.”

I start to wonder whether it is possible with your data to get some insights whether the winning model is really the winning model for each age group. Of course age is continuous but if you would do the model fitting for the three groups you now used, would it really show the same winning model? Because if not, comparing parameters of a model that does not equally well describe behavior of each group is problematic. It would be good if the authors can show something in this direction.

Response: Thank you for this suggestion. We have now run the modelling procedure separately based on the three age groups we used for visualisation (9-12, 13-15, and 16-18 years). This new analysis robustly supported that the same winning model (model 5) is selected for each age group. We have added a new supplementary figure to confirm this and mentioned it in the main text:

“Additional modelling analyses in three separate age groups (9-12, 13-15, and 16-18 years) confirmed that the model 5 won across the full age range (See Figure S6).”

“In addition to the main model fitting and comparison procedure with the full sample, we also repeated the procedure separately for participants aged 9-12 years, those aged 13-15 years, and those aged 16-18 years, to confirm that the same model won across the age range (See Figure S6).”

Figure S6. Model comparison for three separate age groups. The final step in the modelling procedure (i.e., comparison of all five models) was replicated in three separate age groups: 9–12 years, 13–15 years, and 16–18 years. **(a)** Exceedance probability for models with 9–12-year-old participants. **(b)** $\Delta\text{BIC}_{\text{int}}$ for models with 9–12-year-old participants. **(c)** ΔLME for models with 9–12-year-old participants. **(d)** Exceedance probability for models with 13–15-year-old participants. **(e)** $\Delta\text{BIC}_{\text{int}}$ for models with 13–15-year-old participants. **(f)** ΔLME for models with 13–15-year-old participants. **(g)** Exceedance probability for models with 16–18-year-old participants. **(h)** $\Delta\text{BIC}_{\text{int}}$ for models with 16–18-year-old participants. **(i)** ΔLME for models with 16–18-year-old participants. In each age group, the winning model was the same as for the overall sample on all three measures of performance.

This also made me wonder if the current approach is the best. It seems the MAP procedure takes the parameter distribution of the whole group and then uses this in iterative steps to re-estimate the parameters for each individual. What you assume here is thus that all participants have parameters values that are generated by the same generative distribution. And this procedure will pull parameters to the center of mass of that distribution. However, your hypotheses are that there will be age differences in these parameters. So in that sense your current fitting procedure goes against your main hypotheses. The solutions seems to have age in the hierarchical structure that you are fitting. Please clarify your motivation for how the model was set up and consider other options.

Response: Thank you for the opportunity to clarify. We deliberately opted for the more conservative MAP fitting approach so that we could be confident that any age differences we saw were robust and could not be dismissed as simply an artefact of the fitting procedure. We again emphasise that it is more powerful, and biologically plausible, to consider age as a continuous variable which does not simply fit into a hierarchical structure where these additional variables often need to be modelled as discrete group differences. Critically in response to your earlier comment, we have now demonstrated that the same model wins in all three age groups. Therefore, we show that even with a very conservative approach to model fitting (which, as the reviewer points out, will tend to make parameters estimates more similar across participants), we still see age related differences in key model parameters. We have now added additional information to clarify our decision for using the MAP fitting procedure:

“We used an iterative maximum a posteriori (MAP) approach for all model fitting, in line with previous work using reinforcement learning models^{29–31,34}. This procedure computes the maximum posterior probability (PP_i) estimate obtained with parameter vector h_i , given the observed choices and given the prior computed from group-level Gaussian distributions over the parameters with a mean vector μ and standard deviation σ^2 . This is a conservative approach whereby any differences in resulting parameters can be seen as robustly capturing latent differences across people, for example across ages. It is ideally suited for studies of reinforcement learning where group-level estimates can improve the reliability of the resulting fitted parameters⁵⁴.”

Finally, I do like the task because it has the go, no-go, element in it, I like that there is also an amount of points participants start with, this really helps feel losses as losses, the

differences in reward size do not seem particularly helpful. Overall this makes the task similar but also very different from previous studies, and therefore I think it makes an interesting contribution to the existing literature on learning but I do not see how it can or does help providing a framework for unifying the diversity of results.

Response: Thank you for this helpful comment. We agree with your assessment, and it was not our intention to suggest that this study supersedes or unifies earlier work which, as you say, has mostly used a very different learning context. We have now modified the text in various places to emphasise that this is a different learning context and does not necessarily contradict previous findings, but rather highlights that in certain contexts, superficially reward-oriented behaviour might actually be impulsive. For example:

“Adolescence is often considered as a period in which reward sensitivity peaks¹⁻⁵. Using a large, well-characterised, multi-country sample, we demonstrate that, in fact, reward learning rates in certain contexts remain stable across adolescence whilst the tendency to initiate actions decreases.”

“These results highlight the importance of distinguishing between valenced learning mechanisms and action initiation biases. While previous research has demonstrated heightened reward learning in adolescence^{17,19}, we show that developmental differences in reward learning might in fact be specific to certain learning contexts, and apparent reward-oriented behaviour can reflect action initiation biases rather than reward learning processes.”

Some smaller points

“For the analyses of model parameters and age, we excluded 41 participants with values more than three standard deviations from the mean on one or more model parameters.” please move this statement to main text and report how this effects the analyses. Was this a pre-registered exclusion criterium?

Response: Apologies for the lack of clarity. This statement is indeed already in the main text. (In the revised manuscript, the number is now 38 rather than 41; this is because we updated our winning model from 6 to 5). This study could not be pre-registered because the data were collected as part of a previous study, but we have now redone all of our analyses with outliers and other excluded data included, and can confirm that this does not affect any of our results or interpretation. These analyses have been added to the supplementary materials and are summarised in Table S2:

Table S2. Summary of additional analyses

	Main analyses	With model 6	With all eligible participants	With all eligible and outliers	With repetitions 2-10 only
Sample size					
N	742	742	832	832	742
Behavioural analysis: GLMM predicting no. correct responses from age					
Main effect stimulus repetition	OR = 1.19 [1.17, 1.21], $z = 18.56, p < .001$)	N/A	OR = 1.17 [1.15, 1.19], $z = 19.01, p < .001$	N/A	OR = 1.15 [1.13, 1.17], $z = 13.87, p < .001$
Age*repetition	OR = 1.02 [1.01, 1.04], $z = 2.49, p = .01$)	N/A	OR = 1.03 [1.01, 1.04], $z = 2.97, p = .003$	N/A	OR = 1.02 [1.00, 1.04], $z = 2.06, p = .04$
Main effect age	(OR = 1.08 [1.04, 1.11], $z = 4.58, p < .001$)	N/A	OR = 1.07 [1.04, 1.10], $z = 4.92, p < .001$	N/A	OR = 1.08 [1.04, 1.12], $z = 4.48, p < .001$
Age*repetition*valence	OR = 1.09 [1.05, 1.13], $z = 4.65, p < .001$)	N/A	OR = 1.10 [1.06, 1.14], $z = 5.56, p < .001$	N/A	OR = 1.07 [1.03, 1.12], $z = 3.32, p < .001$
BF ₀₁ for (no) age difference in reward learning	57.80	N/A	29.11	N/A	112.00
Modelling: model comparison					
Overall winning model	5	N/A (6)	5	5	5
Modelling: GLMMs predicting parameter values from age					
Main effect of age on punishment learning	$\beta = 0.10$ [0.05, 0.15], $z = 4.12, p < .001$	$\beta = 0.10$ [0.05, 0.15], $z = 4.26, p < .001$	$\beta = 0.11$ [0.07, 0.15], $z = 4.86, p < .001$	$\beta = 0.05$ [0.03, 0.07], $z = 5.40, p < .001$	$\beta = 0.12$ [0.07, 0.17], $z = 4.59, p < .001$
Main effect of age on action initiation bias	$\beta = -0.20$ [-0.28, -0.12], $z = -4.91, p < .001$	$\beta = -0.20$ [-0.28, -0.12], $z = -4.78, p < .001$	$\beta = -0.20$ [-0.27, -0.12], $z = -5.30, p < .001$	$\beta = -0.17$ [-0.23, -0.10], $z = -5.14, p < .001$	$\beta = -0.15$ [-0.23, -0.06], $z = -3.44, p < .001$
Main effect of age on reward learning rate	$\beta = 0.01$ [-0.05, 0.07], $z = 0.30, p = .77$	$\beta = 0.01$ [-0.06, 0.07], $z = 0.17, p = .86$	$\beta = 0.02$ [-0.03, 0.08], $z = 0.80, p = .42$	$\beta = 0.03$ [-0.01, 0.07], $z = 1.29, p = .20$	$\beta = 0.03$ [-0.05, 0.11], $t = 0.73, p = .46^*$
BF ₀₁ for (no) age difference in reward learning rate	13.60	12.62	12.62	11.57	20.00
Main effect of age on temperature parameter	$\beta = 0.002$ [-0.07, 0.07], $z = 0.08, p = 0.94$	$\beta = 0.002$ [-0.07, 0.08], $z = 0.06, p = 0.95$	$\beta = -0.03$ [-0.10, 0.03], $z = -0.96, p = 0.34$	$\beta = -0.03$ [-0.10, 0.04], $t = -0.85, p = 0.39^*$	$\beta = 0.06$ [-0.01, 0.14], $z = 1.70, p = .09$

*These statistics are from a standard linear model without site, due to non-convergence of the mixed (GLMM) model due to model complexity

“This payment was not linked to task performance.”

Please move to main text. Would you think this affects participants of different ages differently?

Response: This statement is also in the main text. The payment was given for participation in a larger study rather than for this task specifically, and therefore we think it is unlikely that participants would have made an association between the final study payment and this specific task (i.e., the lack of association between task performance and payment would have seemed natural). We have amended the text to make this clearer:

“Participants completed a learning task as part of the larger study and received a small monetary or voucher reimbursement in line with local ethical approvals⁴⁴. This payment was not linked to the learning task specifically, and was therefore not associated with task performance.”

Smaller things, but would you think it is a good idea to include test-site as a variable in your models?

Response: Thank you for this query. We do include test site as a random effect in our models, which is the best way to handle these kinds of grouping variables that might affect variability but are not informative in their own right. This is stated in the **Statistical Analysis** section, which we have amended slightly for clarity:

“We tested whether each parameter was predicted by age, with IQ, and sex as covariates (fixed effects) and varying intercepts for different test sites of data collection (random effects).”

“These analyses were conducted using R’s lme4 package glmer function⁵⁵. Participants’ responses were coded as 1 (active response) or 0 (no response) and were predicted from age, sex (0 = male, 1 = female), object repetition number (1-10), and object valence (0 = reward, 1 = punishment) (fixed effects), with varying intercepts allowed for responses grouped by participant nested within test site (random effects).”

Can you explain in more simple words what the stopping criterium is for the iterative MAP procedure? Which measure is used in each step and what it is compared too.

Response: We have added this information to the manuscript:

“Convergence was defined as a change in the posterior likelihood of less than 0.001 between successive maximisation steps (see³¹ for full details).”

Reviewer #2 (Remarks to the Author):

The goal of this study was to examine reward and punishment learning and action initiation (impulsivity) in children and adolescents. Strengths of the study include a strong theoretical rationale, large sample size (n=700+ from 11 countries) and computational modelling approach.

Response: Thank you so much for your positive feedback about our work. We are delighted that you found that our manuscript provided a strong theoretical rationale, large sample size and computational modelling approach. We really appreciate your time to review our work and your valuable comments, which have significantly helped us to improve our manuscript further.

Areas of concern include lack of clarity regarding data exclusion, lack of detail regarding demographics of participants whose data were included in the analyses, and interpretations of the results.

It is surprising that key adolescent learning papers are not cited, including Peters & Crone, 2017 and Davidow et al 2016.

Response: Thank you for pointing out these additional manuscripts that are relevant to the current work. We have added the citations to our introduction:

“Probabilistic learning tasks have suggested an adolescent peak in reward learning¹⁶, better reward learning in adolescents¹⁷, and relatively better reward (versus punishment) learning in adolescents compared to adults¹⁸ (see also¹⁹).”

The authors did not include data from 96 adolescents who responded to fewer than half of the reward trials (i.e. trials where responding was the correct behavior). How can the authors be certain that they did not inadvertently remove meaningful learning (i.e. non-learning) data? Do the results differ significantly if these 96 participants are included?

Response: We made the decision to exclude these participants because we wanted to ensure that all participants were engaged and performing the task correctly, and this criterion ensured that participants were paying attention to the task. We have now run a new analysis including all these excluded participants, which confirms that our results remain the same. We have added a new section to the supplementary materials and an extensive table (Table S2) showing the impact of any exclusions on our key results. In implementing these changes, we noticed we had inadvertently double counted some of our excluded participants. The total number of participants excluded due to responding to less than half of the reward trials is now correctly stated as 78 in the updated manuscript:

“Reanalysis with all 832 eligible participants and no exclusion of outliers

To confirm that our results were not driven by the exclusion of poor-quality data or outliers on model parameters, we repeated our main analyses using the data from all 832 eligible participants (i.e., including those who never responded (n = 4), responded on every trial (n = 2), scored below zero points (n = 6), or responded on fewer than half of the reward trials (n = 78). Analysis of model parameter and age associations were repeated both with and without outliers, i.e., model parameter

values more than three standard deviations from the mean. These analyses revealed virtually identical findings to our main analyses (see Table S2).”

Table S2. Summary of additional analyses

	Main analyses	With model 6	With all eligible participants	With all eligible and outliers	With repetitions 2-10 only
Sample size					
N	742	742	832	832	742
Behavioural analysis: GLMM predicting no. correct responses from age					
Main effect stimulus repetition	OR = 1.19 [1.17, 1.21], $z = 18.56, p < .001$)	N/A	OR = 1.17 [1.15, 1.19], $z = 19.01, p < .001$	N/A	OR = 1.15 [1.13, 1.17], $z = 13.87, p < .001$
Age*repetition	OR = 1.02 [1.01, 1.04], $z = 2.49, p = .01$)	N/A	OR = 1.03 [1.01, 1.04], $z = 2.97, p = .003$	N/A	OR = 1.02 [1.00, 1.04], $z = 2.06, p = .04$
Main effect age	(OR = 1.08 [1.04, 1.11], $z = 4.58, p < .001$)	N/A	OR = 1.07 [1.04, 1.10], $z = 4.92, p < .001$	N/A	OR = 1.08 [1.04, 1.12], $z = 4.48, p < .001$
Age*repetition*valence	OR = 1.09 [1.05, 1.13], $z = 4.65, p < .001$)	N/A	OR = 1.10 [1.06, 1.14], $z = 5.56, p < .001$	N/A	OR = 1.07 [1.03, 1.12], $z = 3.32, p < .001$
BF ₀₁ for (no) age difference in reward learning	57.80	N/A	29.11	N/A	112.00
Modelling: model comparison					
Overall winning model	5	N/A (6)	5	5	5
Modelling: GLMMs predicting parameter values from age					
Main effect of age on punishment learning	$\beta = 0.10$ [0.05, 0.15], $z = 4.12, p < .001$	$\beta = 0.10$ [0.05, 0.15], $z = 4.26, p < .001$	$\beta = 0.11$ [0.07, 0.15], $z = 4.86, p < .001$	$\beta = 0.05$ [0.03, 0.07], $z = 5.40, p < .001$	$\beta = 0.12$ [0.07, 0.17], $z = 4.59, p < .001$
Main effect of age on action initiation bias	$\beta = -0.20$ [-0.28, -0.12], $z = -4.91, p < .001$	$\beta = -0.20$ [-0.28, -0.12], $z = -4.78, p < .001$	$\beta = -0.20$ [-0.27, -0.12], $z = -5.30, p < .001$	$\beta = -0.17$ [-0.23, -0.10], $z = -5.14, p < .001$	$\beta = -0.15$ [-0.23, -0.06], $z = -3.44, p < .001$
Main effect of age on reward learning rate	$\beta = 0.01$ [-0.05, 0.07], $z = 0.30, p = .77$	$\beta = 0.01$ [-0.06, 0.07], $z = 0.17, p = .86$	$\beta = 0.02$ [-0.03, 0.08], $z = 0.80, p = .42$	$\beta = 0.03$ [-0.01, 0.07], $z = 1.29, p = .20$	$\beta = 0.03$ [-0.05, 0.11], $t = 0.73, p = .46^*$
BF ₀₁ for (no) age difference in reward learning rate	13.60	12.62	12.62	11.57	20.00
Main effect of age on temperature parameter	$\beta = 0.002$ [-0.07, 0.07], $z = 0.08, p = 0.94$	$\beta = 0.002$ [-0.07, 0.08], $z = 0.06, p = 0.95$	$\beta = -0.03$ [-0.10, 0.03], $z = -0.96, p = 0.34$	$\beta = -0.03$ [-0.10, 0.04], $t = -0.85, p = 0.39^*$	$\beta = 0.06$ [-0.01, 0.14], $z = 1.70, p = .09$

*These statistics are from a standard linear model without site, due to non-convergence of the mixed (GLMM) model due to model complexity

It is also curious that the final sample consisted of twice as many girls as boys. Was the majority of data that was excluded from boys?

Response: Thank you for highlighting this. The sex imbalance reflects that these data were taken from a larger study, which deliberately over-sampled females. We have now stated this in the text:

“The final sample thus consisted of 742 youths (491 girls, 251 boys; the sex imbalance here reflects a deliberate over-sampling of girls in the larger study from which these data were taken).”

Was there a bias in terms of age for excluded data? (i.e. was the likelihood of not including data higher among younger participants?). The authors found that “reward learning rates did not differ significantly by age” but if the younger participants’ data who did not understand the task was excluded, then only the younger participants who were more cognitively sophisticated would have been included; this would, unsurprisingly, lead to a seemingly (and perhaps not representative) lack of age difference.

Response: Thank you for the opportunity to clarify. We ran additional analyses to confirm that there was no age bias in the number of excluded participants. A Bayesian analysis showed substantial evidence for the null (no age differences in excluded versus included participants). We have added this information to the supplementary materials:

“We also note that there was no significant age difference between the excluded participants and those included in the final sample (2-tailed t-test: $t_{(110.94)} = 0.25$, $p = .81$, $BF_{01} = 7.87$, substantial evidence for the null).”

Furthermore, we have now repeated all of our analyses with these data included, and can confirm that our results were unchanged (see Table S2 in our response to your previous comment).

How many participants were included in each of the 5 pubertal groups?

Response: The numbers are as follows: pre-pubertal = 52, early pubertal = 65, mid-pubertal = 167, late pubertal = 356, post-pubertal = 102. We have now added this under **Participants** in the main text:

“Of these, 52 were classed as pre-pubertal, 65 as early pubertal, 167 as mid-pubertal, 48 as late pubertal, and 16 as post-pubertal.”

The current student finds stability in reward learning rates across adolescence. In the Discussion the authors claim that this result in contrast to findings showing that adolescence is a period of heightened sensitivity to reward. However, this is not a reasonable comparison because the majority of studies that report heightened reward sensitivity/learning in adolescence compared adolescents to age-distinct children and adults rather than across the period of adolescence as is done here.

Response: Thank you for highlighting this. We have rephrased the opening sentence of our discussion to avoid making this comparison:

“Adolescence is often considered as a period in which reward sensitivity peaks¹⁻⁵. Using a large, well-characterised, multi-country sample, we demonstrate that, in fact, reward learning rates in certain contexts remain stable across adolescence whilst the tendency to initiate actions decreases.”

Our intention here was to show that, between late childhood (pre-pubertal/pre-teen status) and the end of adolescence (post-pubertal/legal adulthood), there is no peak in reward learning in this type of learning context. Although we do not compare distinct age groups, our sample covers a broad age range, and we think it is unlikely that an adolescent peak in reward learning (relative to children/adults) would not have manifested in at least some evidence for a peak within our age range. Otherwise, we would have to assume that the adolescent peak has already been achieved by eight years of age and does not begin to decline until at least a year or two after the completion of puberty. For comparison, a previous study covering a wide age range reported a peak in reward learning between 13-15 years old, which is comfortably within our range (Xia et al., 2021). We think it is more likely that the discrepancy with previous studies reflects the different learning context in this study, and we have now also inserted the words “in certain contexts” (see above) to make this clearer.

This is further confounded by the possibility that the majority of data that was excluded is from the younger (e.g. 9, 10, 11 year olds) participants, then the age range included in the analysis is much more narrow than in the previous studies that found a quadratic effect.

Response: We can confirm that there was no age bias in the exclusion criteria (see our response to your previous comments).

Please provide a more clear description of how action initiation bias was measured.

Response: We have added more detail in the introduction where we first introduce the action initiation bias:

“... action initiation biases (a bias to make a response, regardless of its expected outcome value)”

The authors fail to describe the nuances in the 11 countries included—how do these countries differ in a manner that might have impacted the findings? It is unclear if country was included as a covariate.

Response: In all our analyses we indeed included testing site as a covariate. Therefore, all of our results are robust to any country-level differences. We have deliberately avoided making statements about national differences, because our study was not designed to detect national differences. By using a nationally diverse sample we hope that our results are more representative of adolescents generally, or at least those living in high-income Western countries. However, we would note that processes such as reward and punishment learning are fundamental aspects of human behaviour, and animal behaviour more broadly (see O’Doherty et al., 2017). Given this, we believe it is unlikely that we would see large country level differences. We have reworded the information we provide in our statistical analysis section to

make it clear that test sites were modelled as random effects to account for any site differences:

“We tested whether each parameter was predicted by age, with IQ and sex as covariates (fixed effects) and varying intercepts for different test sites of data collection (random effects).”

“Participants’ responses were coded as 1 (active response) or 0 (no response) and were predicted from age, sex (0 = male, 1 = female), object repetition number (1-10), and object valence (0 = reward, 1 = punishment) (fixed effects), with varying intercepts allowed for responses grouped by participant nested within test site (random effects).”

We have also added a note to our discussion about the strengths and limitation of our diverse sample:

“We used a large (N = 742), mixed-sex sample, which was nationally and linguistically diverse, carefully screened to be typically developing in terms of psychiatric functioning, and well characterised in terms of social background. The national and linguistic diversity helps our findings to represent adolescent behaviour more universally. In all analyses, we modelled test site as a random effect, and our results remain robust to this modelling. Given the fundamental nature of reward and punishment learning⁴⁰, we did not anticipate or set out to test country-level differences, but future studies could seek to include an even larger and more geographically diverse sample to further investigate possible differences between countries and cultures in learning rates across adolescence”

An explication of the relevance and implications of these findings to real-world adolescent decision-making would be helpful to the reader.

Response: Thank you for this suggestion. We have added the following to the discussion to address this point in more detail:

“Even in healthy adolescents, a better understanding of the difference between reward-oriented and impulsive behaviour could potentially facilitate behavioural interventions designed to reduce risky behaviour. For example, there may be contexts where it is more beneficial to focus on planning and impulse control than on learned behaviour, even when risky behaviour appears superficially to be driven by desirable outcomes (e.g., social status or material goods).”

Reviewer 3

The authors present a study on a large developmental sample that suggests that behavior is not only guided by reinforcement learning (i.e., effects of previous outcomes on future behavior), but also by an “action initiation bias” (i.e., the tendency to respond [“go”] rather than withhold a response [“nogo”]), and that this action initiation bias changes with age. These findings might have implications for clinical research, given that the action initiation bias might be related to impulsivity, and both reward learning and impulsivity have been argued to underlie clinical conditions, especially during adolescence.

The study has several strengths, including the large sample size and the clear language in describing the results. However, it also has limitations that need to be addressed. Most prevalently, the computational model needs to be improved because it does not appropriately capture human behavior, which is most evident in the fact that simulated behavior does not capture the crucial aspects of human behavior, i.e., a failure in qualitative model fit. Previous computational models have provided better fits for this task, and the authors could use these as inspiration to improve their model.

Response: Thank you for your positive feedback about our work. We are delighted that you found our manuscript well written and were impressed with the sample size. We really appreciate your time reviewing our work and your valuable comments, which have significantly helped us to improve our manuscript further. We address your concerns in detail below.

There also might be issues in terms of the model fitting procedure, which lead to a wide spread of estimated parameters across participants and low parameter recovery (lowest with $r=0.1$).

Response: We agree that the recovery of the rho/magnitude sensitivity parameter is low, albeit significant. For this reason, as well as no specific hypotheses about how magnitude sensitivity differs with age, we refrained in the manuscript from interpreting the age differences in magnitude sensitivity that we observed. However, the inclusion of this parameter did nonetheless improve the model fit and the resulting model showed good identifiability, so we accepted model 6 as the best model.

To address the concerns about the low recovery of the rho parameter we have now run new analyses showing that both model 5 (without the rho) and model 6 (with the rho) show the same age-related differences in action initiation/go bias and punishment learning. In addition, given the poor recovery of the rho, we have now omitted it from our final model space and updated our winning model throughout the paper to be model 5 without the rho parameter. The parameters from model 5 show excellent recovery and model 5 also shows good identifiability. The original analysis with model 6 can now be found summarised in Supplementary Table S2 and Supplementary Figure S5. Figure 3 showing recovery and model identifiability has now been updated as below. Importantly, the relationships between model parameters and age were unchanged between model 5 and 6, which highlights the robustness of these associations.

Table S2. Summary of additional analyses

	Main analyses	With model 6	With all eligible participants	With all eligible and outliers	With repetitions 2-10 only
Sample size					
N	742	742	832	832	742
Behavioural analysis: GLMM predicting no. correct responses from age					
Main effect stimulus repetition	OR = 1.19 [1.17, 1.21], $z = 18.56$, $p < .001$)	N/A	OR = 1.17 [1.15, 1.19], $z = 19.01$, $p < .001$	N/A	OR = 1.15 [1.13, 1.17], $z = 13.87$, $p < .001$
Age*repetition	OR = 1.02 [1.01, 1.04], $z = 2.49$, $p = .01$)	N/A	OR = 1.03 [1.01, 1.04], $z = 2.97$, $p = .003$	N/A	OR = 1.02 [1.00, 1.04], $z = 2.06$, $p = .04$
Main effect age	(OR = 1.08 [1.04, 1.11], $z = 4.58$, $p < .001$)	N/A	OR = 1.07 [1.04, 1.10], $z = 4.92$, $p < .001$	N/A	OR = 1.08 [1.04, 1.12], $z = 4.48$, $p < .001$
Age*repetition*valence	OR = 1.09 [1.05, 1.13], $z = 4.65$, $p < .001$)	N/A	OR = 1.10 [1.06, 1.14], $z = 5.56$, $p < .001$	N/A	OR = 1.07 [1.03, 1.12], $z = 3.32$, $p < .001$
BF ₀₁ for (no) age difference in reward learning	57.80	N/A	29.11	N/A	112.00
Modelling: model comparison					
Overall winning model	5	N/A (6)	5	5	5
Modelling: GLMMs predicting parameter values from age					
Main effect of age on punishment learning	$\beta = 0.10$ [0.05, 0.15], $z = 4.12$, $p < .001$	$\beta = 0.10$ [0.05, 0.15], $z = 4.26$, $p < .001$	$\beta = 0.11$ [0.07, 0.15], $z = 4.86$, $p < .001$	$\beta = 0.05$ [0.03, 0.07], $z = 5.40$, $p < .001$	$\beta = 0.12$ [0.07, 0.17], $z = 4.59$, $p < .001$
Main effect of age on action initiation bias	$\beta = -0.20$ [-0.28, -0.12], $z = -4.91$, $p < .001$	$\beta = -0.20$ [-0.28, -0.12], $z = -4.78$, $p < .001$	$\beta = -0.20$ [-0.27, -0.12], $z = -5.30$, $p < .001$	$\beta = -0.17$ [-0.23, -0.10], $z = -5.14$, $p < .001$	$\beta = -0.15$ [-0.23, -0.06], $z = -3.44$, $p < .001$
Main effect of age on reward learning rate	$\beta = 0.01$ [-0.05, 0.07], $z = 0.30$, $p = .77$	$\beta = 0.01$ [-0.06, 0.07], $z = 0.17$, $p = .86$	$\beta = 0.02$ [-0.03, 0.08], $z = 0.80$, $p = .42$	$\beta = 0.03$ [-0.01, 0.07], $z = 1.29$, $p = .20$	$\beta = 0.03$ [-0.05, 0.11], $t = 0.73$, $p = .46^*$
BF ₀₁ for (no) age difference in reward learning rate	13.60	12.62	12.62	11.57	20.00
Main effect of age on temperature parameter	$\beta = 0.002$ [-0.07, 0.07], $z = 0.08$, $p = 0.94$	$\beta = 0.002$ [-0.07, 0.08], $z = 0.06$, $p = 0.95$	$\beta = -0.03$ [-0.10, 0.03], $z = -0.96$, $p = 0.34$	$\beta = -0.03$ [-0.10, 0.04], $t = -0.85$, $p = 0.39^*$	$\beta = 0.06$ [-0.01, 0.14], $z = 1.70$, $p = .09$

*These statistics are from a standard linear model without site, due to non-convergence of the mixed (GLMM) model due to model complexity

Figure 3. Model performance and validation. (a) Exceedance probability for the five computational models that comprised the final model space. The winning model was the $2\alpha\beta b_c$ model, with separate reward and punishment learning rates, and a constant action initiation bias. (b) ΔBIC_{int} , relative to the winning model ($2\alpha\beta b_c$). (c) ΔLME , relative to the winning model ($2\alpha\beta b_c$). (d) Parameter recovery. The confusion matrix represents Spearman correlations between simulated and fitted (recovered) parameters. Each parameter exhibited a significant positive correlation between its true and fitted values, with r values ranging from 0.49-0.92 (shown on the lower diagonal) (e) Exceedance probability from the model identifiability procedure. The diagonal represents the probability of each model having the best fit to its own synthetic data. The winning model ($2\alpha\beta b_c$) was highly identifiable from other models. (f) Number of runs where each model was selected as the best fit for data generated by each model in the model identifiability procedure. The diagonal represents the number of runs each model was selected as the best fit for its own data. The winning model ($2\alpha\beta b_c$) was the best fit to its own data on all 10 runs.

Further, the action bias parameter is commonly used in similar tasks and models (e.g., Guitart-Masip et al., 2021), and I would recommend that the authors state that more clearly. It sometimes sounds like this study is the first to introduce the action bias parameter, but the novelty of this study seems instead to be the application of a known model to a large developmental sample.

Response: Thank you for highlighting this. We are happy to clarify this in the text, as it was not our intention to imply that this parameter had not been used before. We have amended the following sections in the text:

“There has been a relative paucity of computational modelling work focusing on learning in adolescence, and previous studies were usually not designed to distinguish between learning processes and action initiation biases...”

“To our knowledge, only one study of adolescents to date has measured learning in a task design that incorporates requirements both to learn and also to inhibit actions²⁵ (see²⁶ for a study in adults). This study compared reward and punishment learning as well as an action initiation bias or tendency to ‘go’ (initiate an action) vs. ‘no-go’ (withhold an action) in children (8-12, n = 20), adolescents (13-17, n = 20), and adults (18-25 years, n = 21).”

Second, the behavioral analyses of the paper could be strengthened, e.g., by conducting more specific tests that relate to the hypotheses. The interpretation of some statistical models also needs to be refined, and some claims that are not currently backed up by tests could be.

Response: We have addressed these issues below, following your more detailed comments.

From a presentation perspective, the paper sometimes seems to imply that because the results of this study are not the same as the results other labs have found in previous studies, the results of previous studies might be faulty. I would recommend providing more nuance when comparing this study to previous ones. For example, specific parameter values are often a function of the underlying task (e.g., fitted reward learning rates are likely higher in a task in which rewards are informative than in a task in which rewards are not informative, reflecting participants’ adaptive strategies). In other words, a previous study with a different result does not necessarily contradict the current results because both reflect the specific task used.

Response: Thank you for highlighting this. It was not our intention to suggest previous studies were faulty. We have amended the opening part of our discussion to acknowledge more clearly that the learning context in our study is different from previous studies. Our aim was also to highlight the importance and additional robustness from being able to analyse data from a large and geographically diverse sample of adolescents, which is something that has been rarely done before:

“Using a large, well-characterised, multi-country sample, we demonstrate that, in fact, reward learning rates in certain contexts remain stable across adolescence whilst the tendency to initiate actions decreases.”

We have also emphasised this much more strongly in the discussion:

“While previous research has demonstrated heightened reward learning in adolescence^{16,18}, we show that developmental differences in reward learning might in fact be specific to certain learning contexts, and apparent reward-oriented behaviour can reflect action initiation biases rather than reward learning processes.”

I provide more details for each of these points below:

Major

About the task / dataset

My main concern was if participants—especially the youngest ones—fully understood the task. Could the authors provide more details about the training procedure, including instructions given to participants and whether participants received any test trials before the main experiment? What did participants know about the task beforehand? E.g., did they know it was deterministic, and that there was an equal number of go-reward and avoid-punishment objects?

Response: Participants were not explicitly told that the task was deterministic, although this was clearly implied in the instructions they received. This procedure is the same as studies where outcomes are probabilistic and this is not explicitly instructed, to encourage learning (e.g. Lockwood et al., 2016; 2021). We are confident that the youngest participants fully understood the task for several reasons. First, there was no age bias when excluding participants (i.e., younger children were no more likely to be excluded due to poor quality data). We have added this information to the manuscript:

“We also note that there was no significant age difference between the excluded participants and those included in the final sample (2-tailed t-test: $t_{(110.94)} = 0.25$, $p = .81$, $BF_{01} = 7.87$, substantial evidence for the null).”

Second, the same model wins even in the youngest participants when we divide our sample into three age groups. We have added a new supplementary figure to demonstrate this:

Figure S6. Model comparison for three separate age groups. The final step in the modelling procedure (i.e., comparison of all five models) was replicated in three separate age groups: 9–12 years, 13–15 years, and 16–18 years. **(a)** Exceedance probability for models with 9–12-year-old participants. **(b)** $\Delta\text{BIC}_{\text{int}}$ for models with 9–12-year-old participants. **(c)** ΔLME for models with 9–12-year-old participants. **(d)** Exceedance probability for models with 13–15-year-old participants. **(e)** $\Delta\text{BIC}_{\text{int}}$ for models with 13–15-year-old participants. **(f)** ΔLME for models with 13–15-year-old participants. **(g)** Exceedance probability for models with 16–18-year-old participants. **(h)** $\Delta\text{BIC}_{\text{int}}$ for models with 16–18-year-old participants. **(i)** ΔLME for models with 16–18-year-old participants. In each age group, the winning model was the same as for the overall sample on all three measures of performance.

Third, even when we exclude the first presentation of each object, the associations with age also remained the same, as shown in the new Table S2.

In addition, we have updated our description of the task to make the instructions given to participants clearer:

“Participants were carefully instructed that some of the objects were ‘good’ (would earn them points) and some were ‘bad’ (would lose them points), and that they must learn which objects were good and which were bad so that they could respond only to the good objects. The researcher checked that this was clear to each participant. There were no practice trials before the main experiment so that participants would be learning for the first time from the first trial onwards.”

Participants were not informed about the number of reward and punishment (or total) stimuli. We suspect that doing so would have prompted the participants to begin explicitly counting the objects, and thus might have complicated the task unnecessarily by introducing a stronger working memory component.

In addition, we have also added further information about the analyses with and without data exclusions and analyses of trials 2-10 to the **Discussion** in the main text:

“We also note that participants were not given practice trials, to encourage learning from the first stimulus onwards. However, as a control analysis, we repeated our analyses with these initial trials excluded. The analysis of trials 2-10 showed that the associations between age and model parameters remained unchanged, supporting the robustness of these associations. Furthermore, we assessed whether there were any age differences between excluded participants and the final sample, and obtained substantial evidence in support of the null hypothesis that data exclusions were not biased by age. Our findings therefore appear to be robust to data exclusions and across repetitions 2-10 as well the full learning task.”

I am particularly interested in this because the data show a sudden increase in “go” responses between the first and second repetition number (I am pasting the behavioral plot below). The fact that this increase is present both for reward and for punishment trials implies that this was not based on task feedback, and participants might instead still be learning how to do the task: For example, in the first iteration, participants might have been relatively cautious and avoided potential negative outcomes by inhibiting “go” responses. But by the second repetition, they might have realized that they can only obtain feedback—i.e., the crucial learning information—if they choose the “go” response. Alternatively, participants who are unfamiliar with the specific task might simply have missed more trials on the first repetition, which would be counted as “nogo” responses, and explain the lower proportion of “go” responses specific to the first iteration.

Response: Thank you for this query. We agree it is a possibility that the initial low response levels are due to unfamiliarity with the task. However, this is not unusual in reinforcement learning tasks where on the first trial the value of all options is completely unknown. We have now conducted a new analysis including only trials 2-10 so that we can demonstrate that our findings are robust to any differences in behaviour on the first object presentation. This analysis showed exactly the same associations with age as in analyses with all trials included. Since it is important that participants learn from the beginning and very difficult to accurately model the

stimulus values from the second object presentation, we present these new results in our supplementary materials only (see Table S2 and Figure S7 reproduced below):

Figure S7. Model comparison with stimulus repetitions 2-10 only. (a) Exceedance probability for the four computational models that comprised the model space when including data from repetitions 2-10 only. The winning model on this measure was the $2a\beta b_c$ model. **(b)** ΔBIC_{int} , relative to the winning model ($2a\beta b_c$). **(c)** ΔLME , relative to the winning model ($2a\beta b_c$). Note model 3 is not included in the model space since there can be no initial action initiation bias when omitting the first trial.

In both cases, younger participants might be more affected than older participants. Do the authors have evidence that 2.5 seconds was enough time for the youngest participants? For example, were more younger than older participants excluded due to a lack of task performance or understanding? Another argument for a lack of task understanding was the fact that older participants gained more points than younger participants overall, as you show using the robust linear mixed effects regression.

Response: We would expect older adolescents to earn more points than younger participants, in line with their better performance on this task (and this also seems to be the case in the literature generally, e.g., Raab & Hartley, 2020). However, as noted in response to your previous comments, there was no age bias in exclusions for poor data quality and therefore we are confident that task understanding was not a particular problem for our younger participants.

We ran an additional control analysis to confirm that 3000ms was sufficient for participants to make a response if they elected to do so. This analysis showed that the mean RT was 1000ms, and in the youngest participants (aged 9-12) it was 1050ms. Therefore, all participants were able to respond in the allotted time. We have added this to the methods section:

“We checked that 3000ms was sufficient for participants to make a response if they elected to do so. The mean reaction time was 1000ms for the whole sample, and 1050ms in the youngest participants (aged 9-12 years). Therefore, all participants were able to respond in the allotted time (see Supplementary materials for additional analysis of reaction times).”

I was also wondering if the youngest participants already understood the large numbers used for this study, which are in the thousands. I am wondering at what time children typically learn these numbers at school, and whether children who have not covered them yet might process the feedback differently, due to a lack of understanding what it meant. (Could behavior be plotted separately for children who have not covered large numbers yet in school, compared to those who have, compared to adults?) For example, the lack of differences in learning speed for stimuli with different reward values suggests that participants either ignored that information, or were unable to process it. Especially the interesting pattern with learning being fastest for the 700-point stimulus, compared to both 1400 and 2000, made me wonder if younger participants thought 700 is more than 2000 because 7 is more than 2? To see if this might be the case, the authors could, e.g., plot the average “mean response” of repetitions 5-10 over age. This would show if the youngest participants indeed thought 700 is better than 2000, and might also reveal whether older participants were more sensitive to the actual numbers.

Response: Based on standard mathematics curricula for this age range, all the participants in our sample should understand place value in numbers up to four digits (e.g., understand that 700 is less than 2000 even though 7 is more than 2), be able to count in multiples of 1000, and order and compare numbers beyond 1000 (see for example p.24 in:

https://assets.publishing.service.gov.uk/government/uploads/system/uploads/attachment_data/file/335158/PRIMARY_national_curriculum_-_Mathematics_220714.pdf).

However, the subjective value of rewards and losses is frequently non-linear, with increases in objective reward value having less subjective value as the magnitude of reward increases (e.g., Yang & Qi, 2021; <https://doi.org/10.3390/jtaer16040041>, DiPalantino & Vojnovic, 2009; <https://doi.org/10.1145/1566374.1566392>; Kahneman & Tversky, 1979; <https://doi.org/10.2307/1914185>). Our lack of a linear relationship between point value and learning, especially at the higher magnitudes, would seem to fit with this literature rather than imply a lack of numerical understanding.

We have now added an additional plot of the behavioural responses by point magnitude broken down by age group in Supplementary materials:

Figure S2. Responding across repetitions by object value for three age groups. (a) Responses by participants aged 9-12 years. (b) Responses by participants aged 13-15 years. (c) Responses by participants aged 16-18 years. The pattern of responding to all reward magnitudes was broadly similar across all three age groups, while older participants showed decreased responding to all punishment magnitudes compared to younger participants.

The data suggest that participants did not treat values as a continuous variable: The figure

below shows that all the negative values tend to cluster together, that “1” seems to be treated separately, and the positive values are quite mixed. Can the authors discuss potential factors that relate to this finding? (To help interpret the figure, it might also be helpful to use a continuous color scheme; maybe neutral in the middle, red for positive values, blue for negative ones).

Response: Thank you for the suggestion regarding the colour scheme, which we have now changed (see for example the figures in response to your previous comment). As we also noted in response to your previous comment, it is common for objective values not to map onto subjective values/learning in this way. We are also aware that with only four objects each for reward and punishment, and no a-priori hypotheses about point magnitude, it would be difficult for us to draw conclusions about the effect of point magnitude on behaviour. We have added a brief discussion about these points to the supplementary materials:

Behavioural responses by stimulus point value

We plotted participants' behavioural responses over stimuli repetitions, by the point value of the stimuli (see Figure S1). Punishment stimuli clustered together, as did higher value reward stimuli. This suggests that participants' subjective response to the point values was non-linear, which is commonly observed in the literature (e.g., 1–3). However, we note that with only four stimuli each for reward and punishment trials, it would be premature to draw conclusions about the effect of point value on participants' behaviour. In addition, our computational models suggested that magnitude information was not a strong driver of participants' behaviour.

The dataset consists of 491 girls and 251 boys. Given that there were sex effects on behavioural measures, I was wondering if the sex composition differed between different ages, and might explain some age-related effects.

Response: Thank you for pointing this out. There was no significant association between age and sex, and Bayesian evidence confirmed substantial evidence in support of the null. We have now added this information to the manuscript:

“There was no association between age and sex in the final sample (2-tailed t-test: $t_{(488)} = 1.00$, $p = .20$, $BF_{01} = 5.03$, substantial evidence in support of the null).”

I was also wondering if the country of origin affected the behavior. E.g., did the authors conduct ANOVAs to test for differences in testing sites?

Response: We use site of data collection as a random effect in our mixed models, so that we can account for variation due to site differences without treating this as theoretically informative in its own right (because our study was not designed to detect national differences). This information is detailed under the **Statistical Analysis**, section of the manuscript. We have now slightly reworded it for clarity:

“We tested whether each parameter was predicted by age, with IQ and sex as covariates (fixed effects) and varying intercepts for different test sites of data collection (random effects).”

“Participants’ responses were coded as 1 (active response) or 0 (no response) and were predicted from age, sex (0 = male, 1 = female), object repetition number (1-10), and object valence (0 = reward, 1 = punishment) (fixed effects), with varying intercepts allowed for responses grouped by participant nested within test site (random effects).”

From a purely methodological perspective, I was curious to know why the authors chose to not display any feedback after “nogo” trials, rather than, e.g., saying “You have not gained or lost any points,” for consistency with the visual feedback of “go” trials?

Response: This task was developed by the FemNAT-CD consortium (<http://www.femnat-cd.eu/>). We believe the decision was made not to include feedback so that the task would be comparable to previous versions, and to avoid any ambiguity about whether this kind of feedback was really perceived as neutral by the participants. In addition, the lack of feedback reduces the cognitive demands from reading extra text on the screen.

Behavioral analyses

From a purely rhetorical point of view, I would suggest presenting the behavioral results before the modeling results, to base the interpretation of the more abstract parameters on a solid understanding of participant behavior.

Response: We have now presented the behavioural results first.

I was also missing more in-depth behavioral analyses. For example, I was interested whether participants showed win-stay-lose-shift behavior (i.e., a higher tendency to repeat the previous choice for a given stimulus when that choice was previously rewarded, and higher tendency to switch the choice when it was punished). In this case, because only the “go” choice leads to rewards and punishments, the analysis could compare participants' stay versus switch choices in the conditions “previous go and reward”, “previous go and punishment”, and “previous nogo”.

Response: Thank you for this suggestion. We have now done this analysis, but it is difficult to interpret given the task design. Specifically, because the same stimulus very rarely appears consecutively, punishment stimuli are slightly more likely to be followed by reward stimuli than by another punishment stimulus, and vice versa for rewards. We would therefore expect participants to make more responses on average following a punishment stimulus than a reward stimulus, and this is indeed what we observe (mean response following punishment: 0.81, SD = 0.39; mean response following reward: 0.64, SD = 0.48; mean response following neutral/no-go outcomes: 0.62, SD = 0.48; the difference in response rates following the different trial types was significant; GLMM: OR = 0.66 [0.64, 0.68], $z = -34.51$, $p < .001$). Thus, behaviour that appears to be irrational when interpreted in the context of win-stay lose-shift (i.e., responding more after punishment) is actually ‘correct’ in the context of this task.

The other option, to conduct the same analysis at the level of individual stimuli (e.g., so that the 'previous trial' was the previous presentation of the same stimulus rather than the previous trial in time), is also not possible because there is no variation in outcome for 'go' trials at the stimulus level. Given the difficulty of defining or interpreting win-stay lose-shift behaviour in this learning context, we feel that it would be confusing to include these analyses in the manuscript. However, we are happy to add them if you (the reviewer) prefer.

I was also interested in whether response times could shed additional light on task behavior. For example, do the authors observe response-time slowing after punishment outcomes? How do response times change over repetition? And over age?

Response: Similar to our response to your previous comment, it is challenging to interpret trial-order effects in this task, including reaction time slowing following punishment. When we conducted this analysis, we observed faster reaction times following punishment, which again, likely reflects the greater probability of the subsequent trial being a reward trial ($\beta = 46.43 [40.61, 52.25]$, $t_{(38930)} = 15.63$, $p < .001$. Mean RT following punishment: 0.93s, SD = 0.54s; following reward: 1.02s, SD = 0.56s; following neutral/no-go outcomes: 1.00s, SD = 0.53s). Due to the difficulty in interpreting these differences, we have not added this analysis to the manuscript.

To address your second point, we have added an analysis of reaction times across repetitions by age to the Supplementary materials:

“Reaction time across stimulus repetitions by age

We analysed whether reaction times for 'go' responses across the task were related to age. A linear mixed effects model predicting reaction time from stimulus repetition, sex, and IQ (fixed effects) and participant nested in site of data collection (random effects) revealed a significant main effect of age on reaction times, with older participants reacting faster ($\beta = -23.37 [-40.55, -6.18]$, $t_{(40084)} = -2.67$, $p = .008$). There was also a main effect of stimulus repetition on reaction times, with faster reaction times for later repetitions ($\beta = -53.67 [-58.62, -48.73]$, $t_{(40084)} = -21.29$, $p < .001$), and a significant age by repetition interaction ($\beta = -9.69 [-14.63, -4.75]$, $t_{(40084)} = -3.85$, $p < .001$), with reaction times decreasing more rapidly for older participants (see Figure S10). The age-related patterns in reaction times are similar to previous learning experiments where reaction times decrease when actions are learned, and fit with previous developmental studies suggesting age related reaction time decreases in general^{4,5}.”

Reaction time across repetitions by age

Figure S10. Reaction time for ‘go’ responses across stimulus repetitions, by age. Reaction times decreased over the course of the task, and this decrease was greater for older participants ($\beta = -9.69$ $[-14.63, -4.75]$, $t_{(40084)} = -3.85$, $p < .001$). Age was treated as continuous in this analysis; age bins are for plotting only.

On a different note, it appears that the main difference between the reward and punishment conditions seems to be that go responses stay high in the reward condition, whereas they drop in the punishment condition. It would be interesting to see within-participant differences between these two conditions in light of this possibility, and to consider potential implications about cognitive processing (e.g., the authors mention a default strategy of responding “go”, which is solely modified by punishment feedback).

Response: An analysis of differences between reward and punishment responses would be challenging to interpret, because differences could be driven by changes in either reward or punishment responding. To address any possible differences in an interpretable manner, we have calculated a ‘reward bias’ for each participant, by dividing the number of responses to reward by the total number of responses for each repetition. This analysis showed, consistent with our main analysis, a significant age by repetition interaction, such that the response bias for reward over punishment increased with age and repetition:

Figure S3. Reward response bias across stimulus repetitions by age. There was a significant negative association between age and overall reward response bias ($\beta = -0.12 [-0.16, -0.08]$, $t_{(58352)} = -6.31$, $p < .001$) as well as a significant age by repetition interaction ($\beta = -0.05 [-0.06, -0.05]$, $t_{(58352)} = -15.84$, $p < .001$). Age was continuous in the analysis; age bins are for presentation purposes only.

Lastly, I was interested in how the delay length between the presentation of the same stimulus affected performance. For example, was performance better when the same stimulus was presented twice in quick succession (i.e., with only a small number of other stimuli interleaved)? This would suggest that participants held stimuli in working memory, and might suggest the addition of a working-memory or forgetting parameter to the task model.

Response: Thank you for this suggestion. We had considered adding a forgetting parameter to the model, but a behavioural analysis revealed no effect of the delay time between presentations on the number of correct responses (OR = 0.99 [0.97, 1.01], $z = -0.85$, $p = .40$, $BF_{01} = 161.00$, decisive evidence in support of the null), and no interaction between delay time and age (OR = 1.00 [0.99, 1.02], $z = 0.50$, $p = .61$, $BF_{01} = 203.00$, decisive evidence in support of the null). Therefore, delay time was not a significant predictor of behaviour.

I was also wondering how raw accuracy on the task developed with age. Could the authors provide a plot that shows that?

Response: Raw accuracy improved with age, consistent with prior developmental reinforcement learning studies (reviewed in Nussenbaum & Hartley, 2019). This is mentioned in the manuscript before we describe the age-related differences in learning:

“...and older participants also made more correct responses in total (OR = 1.08 [1.04, 1.11], $z = 4.58$, $p < .001$; see Figure S4), consistent with prior developmental learning studies²³.”

We have now also added a plot showing this in Supplementary materials as requested:

Figure S4. Overall task performance by three age groups. Performance improved with age (OR = 1.08 [1.04, 1.11], $z = 4.58$, $p < .001$). Age was continuous in the analysis; age bins are for presentation purposes only.

Further, there were a few issues with existing analyses that the authors should address:

- The authors state: "... we did not observe a significant pubertal stage by repetition by valence interaction (OR = 1.04 [0.10, 1.07], $z = 1.87$, $p = .06$), suggesting that the punishment-specific improvement in learning was better captured by age than by pubertal stage." It is not valid to claim here that the effect better captured by age than by pubertal stage; if the authors wish to make a claim about specific differences between the effects of age and puberty, both need to be compared directly. The lack of a significant effect in model does not support this claim (e.g., see Makin & Orban de Xivry, 2019)

Response: Thank you for highlighting this. We wanted to express that the model fit for the age model was better than for the puberty model, but the wording was clumsy. We have now rephrased as follows:

"However, we did not observe a significant pubertal stage by repetition by valence interaction (OR = 1.04 [0.10, 1.07], $z = 1.87$, $p = .06$). Furthermore, the model using age was a better fit to the data than the model using pubertal stage ($\Delta\text{BIC} = -137.22$).

- The authors state: "... this age-related learning improvement was specific to learning from punishment outcomes (age by repetition by valence interaction: OR = 1.09 [1.05, 1.13], $z = 4.65$, $p < .001$)." To claim that the improvement was specific to the punishment

condition, it would be necessary to see a plot that shows both the punishment and the reward condition and reveals a difference in the means, or to conduct another follow-up test. By itself, the significant interaction could also imply the inverse patterns (effect for reward, but not punishment) or a simple differences in the magnitudes of the effects for both conditions.

Response: We have now conducted additional analyses to confirm the direction of the interaction and added the following text:

“To check the direction of this interaction, we repeated the analysis using only reward trials and found no age by repetition interaction (OR = 0.98 [0.95, 1.01], $z = -1.48$, $p = .14$), and repeated it again using only punishment trials and found a significant age by repetition interaction (OR = 1.07 [1.05, 1.11], $z = 5.63$, $p < .001$).”

- The authors state: “A generalised linear mixed model (GLMM) (predicting correct responses from age, stimulus repetition number, outcome valence, and covariates; see Methods) revealed a significant main effect of stimulus repetition on the number of correct responses made, with performance improving throughout the task (Odds ratio (OR) = 1.19 [1.17, 1.21], $z = 18.56$, $p < .001$). Thus, participants exhibited learning.” Did the model show any significant interactions? Interactions always need to be interpreted before main effects (For example, imagine a scenario in which two conditions A and B have average values of “4” before an intervention; after intervention, condition B rose to “20”, but condition A stayed at “4”; the main effect of intervention would be significant; however, we would not want to claim that the intervention was overall successful; we need to interpret the interaction between intervention and condition first.) For the current data, potential interactions with age might imply that younger participants exhibited less or no learning. Interactions with outcome valence might mean that there was learning for one, but not the other valence (e.g., plots suggest more learning from punishment than reward).

Response: Thank you for highlighting this. At this point we simply wanted to demonstrate that learning was possible, before going into the interactions in more detail later. We have now moved the behavioural results section so that it comes before the modelling results in the manuscript. Thus, this sentence about learning now comes directly before we discuss the interactions, which we hope will clarify our purpose in reporting this main effect. In addition, we have reworded this section:

“We first examined whether learning occurred during the task. A generalised linear mixed model (GLMM) (predicting correct responses from age, stimulus repetition number, outcome valence, and covariates; see Methods) revealed a significant main effect of stimulus repetition on the number of correct responses made, with performance improving throughout the task (Odds ratio (OR) = 1.19 [1.17, 1.21], $z = 18.56$, $p < .001$). To confirm that learning occurred for both reward and punishment stimuli, we repeated the analysis using only reward trials, and found a significant main effect of repetition on correct responses (OR = 1.13 [1.09, 1.16], $z = 8.48$, $p < .001$). We then repeated the analysis using only punishment trials and again found a significant main effect of repetition on correct responses (OR = 1.28 [1.26, 1.32], $z =$

19.65, $p < .001$). Thus, the task was able to capture learning behaviour in both reward and punishment conditions.”

Lastly, due to the qualitative difference in responding to repetition 1 compared to later repetitions, I am wondering if repetition 1 might be considered part of participants’ learning phase, and if it would clean up some results if the analyses were restricted to repetitions 2-10 only. This would likely improve the fits of the GLMs that contain repetition as a predictor because in the current data, the effect of repetition is highly non-linear, but it seems linear after trial 2. It might also help the computational model, which currently does not capture the quantitative aspects of human behavior. Furthermore, reanalyzing data from repetitions 2-10 would provide evidence for the robustness of the current results in the spirit of leave-one-(repetition-)out validation.

Response: Thank you for this suggestion. We have now re-run all our modelling and behavioural analyses using only data from repetitions 2-10. We can confirm that this did not change the age-related patterns we observed in our main results. The results of this analysis are presented in Table S2 (see above).

Modeling

Model does not capture human behavior adequately My central concern with the current study is that the model does not seem to capture human behavior adequately. As shown in the plot below from the supplemental material, one of the most noteworthy patterns in participant behavior is the initial increase in “P(go),” which is completely absent from the model behavior. Whereas the model thereafter shows a similar decline in go responses in the punishment condition to humans, it shows a steady increase of go responses in the reward condition, which is absent in the human data. (Indeed, the behavioral analyses show a lack of learning from reward, which directly contradicts the model.) This suggests that the model did not indeed capture human behavior.

Response: We have now conducted five further sets of analyses supporting the robustness of our winning model.

First, we have increased the number of simulated participants to match the number of participants in our sample, and used the median parameter values, at the request of a reviewer. The new simulated data more closely follows our model fitted data from participant behaviour.

Second, we have plotted the simulated and fitted data separately for the three different age groups and showed reasonable qualitative model fits across all three groups.

Third, we have repeated our model fitting and comparison procedure separately for the three age groups and again showed that the same model wins across all three age groups.

Fourth, we have repeated our behavioural analyses using only trials 2-10, and finally, we have repeated our model fitting and comparison procedure with trials 2-10. These analyses demonstrated exactly the same associations with age.

Therefore, our model accurately captures behaviour. Additionally, we have reanalysed our parameter recovery and model identifiability with model 5, without the low recovery rho parameter, and shown recovery is good and stronger than our original reported results using model 6. Furthermore, we can still confirm exactly the same associations with age regardless of whether model 5 or 6 is selected as the winning model (see Table S2).

Together, all these new analyses robustly show that our winning model is a good and robust model of participant behaviour.

Furthermore, the generate-and-recover analyses show poor recoverability for some parameters, especially parameter rho with a correlation of just $r=0.1$. That the model was not able to recover the underlying parameters of simulated data suggests that it also did not recover the parameters of human data very well. To determine this issue, it would be instructive to see scatterplots of the generated and recovered values of each parameter (e.g., see Wilson & Collins, 2019).

Response: In response to your earlier comment, we have now selected model 5 as the winning model throughout the manuscript and the parameters from this model recover well.

It would also be helpful to see a correlation matrix of all the model parameters, to see if parameters traded off with each other within the model. For example, learning rates and decision temperature parameters are known to trade off with each other, with important caveats for model interpretation.

Response: We have added these correlations to Supplementary materials. While there were significant correlations between our model parameters, the effect sizes of the correlations are mostly quite small. No correlations are substantial enough to cause concern, and in our study we find no association with the temperature parameter and age ($BF_{01} = 26.10$, strong evidence in support of the null), whilst robustly showing associations between learning rates and age, supporting that these two parameters are capturing separable latent cognitive processes.

Table S4. Correlations between model parameters (Spearman's r [95% confidence intervals])

	β	α_r	α_p	b_c
β	-	-0.58* [-0.63, -0.53]	-0.21* [-0.28, -0.14]	-0.36* [-0.42, -0.30]
α_r		-	-0.10* [-0.17, -0.03]	-0.18* [-0.25, -0.11]
α_p			-	-0.19* [-0.26, -0.12]
b_c				-

Notes: β : temperature parameter, α_r : reward learning rate, α_p : punishment learning rate, b_c : constant action initiation bias. * indicates $p < .05$

These patterns suggest that the models are misspecified and might have to be parameterized differently to adequately capture human behavior (e.g., Katahira, 2015, JoMP). The extremely large variability of parameter values between participants (despite hierarchical fitting) also supports the interpretation of a lack of model fit and potential model misspecification. I was also wondering if models fit participants of different ages equally well? If there are age-related differences in model fit, parameter differences over age would be hard to interpret. To answer this question, the authors could plot simulated behavior separately for participants of different ages, and assess the fit between the model and participants' behavior at each age group. For a more quantitative analyses, they could calculate the NLL for each participant's behavior under the winning model and parameters, and plot it over age.

Response: We hope that our responses to your previous comments have allayed these concerns. Specifically, we have repeated our modelling analyses separately in three age groups, including generating synthetic data based on median parameter values, and repeated all our analyses with only repetitions 2-10. The results from these additional analyses were the same as our main results. We have also selected model 5 as our winning model and demonstrated that our findings are virtually unchanged regardless of whether we select model 5 or model 6. We are therefore confident that our results are robust.

As for the behavioral analyses, I am wondering what the results would look like if the models were refit to the data starting at repetition 2. This would provide a (very mild) version of split-half reliability (i.e., fitting the model on half the dataset and validating it on the other half). If this leads to very different results, the validity of the current claims should be questioned. To capture the overall patterns better, I think the authors have two choices: If they think the prominent increase in "go" responses between repetition 1 and 2 is a crucial aspect of their dataset, they should aim to create a model that captures this increase. If they think that this sudden increase might be an artifact (e.g., related to a lack of previous training with the task), they could consider removing these trials and fitting the model to the remaining data. However, I don't think the model captures human behavior adequately in the current form. If this is the case, i.e., if the current model does not provide an adequate fit to the behavioral data, the fitted model parameters cannot be interpreted. In other words, even though the current model wins against other competing models in terms of numerical model comparison, as soon as at least two different models are compared, there always will be a winning model, and this does not show that the winning model fits the data adequately. Continued search of the model space will likely lead to a model that fits the data better and whose fitted parameters will be interpretable.

Response: Following your previous comments, we have now repeated our modelling analyses using only data from repetitions 2-10 and can confirm that this did not change our results (see Table S2).

Improvement of the model

There are several ways in which the model could be improved. First, additional parameters could help explain additional variance, and resolve issues of model misspecification (e.g., Katahira, 2015, JoMP). For example, similar tasks that involve a number of different stimuli whose presentation is spread out in time often involve additional memory parameters (e.g., episodic memory: Bornstein; working-memory: Collins), or forgetting parameters (e.g., Xia & Collins, 2021). If the authors find that the delay between presentations of the same stimulus explains variance in behavior (as suggested above), then adding a forgetting parameter will likely improve the fit of the computational model.

Response: Thank you for these suggestions. We investigated this possibility but found that there was substantial evidence in support of the null that the delay between stimulus repetitions did not explain participant behaviour, as we describe in our response to your earlier comment, which we reproduce below:

“Thank you for this suggestion.

We had considered adding a forgetting parameter to the model, but a behavioural analysis revealed no effect of the delay time between presentations on the number of correct responses (OR = 0.99 [0.97, 1.01], $z = -0.85$, $p = .40$, $BF_{01} = 161.00$, decisive evidence in support of the null), and no interaction between delay time and age (OR = 1.00 [0.99, 1.02], $z = 0.50$, $p = .61$, $BF_{01} = 203.00$, decisive evidence in support of the null). Therefore, delay time was not a significant predictor of behaviour.”

I was also wondering how the go bias parameters in the current model compares to the more common persistence / choice stickiness / perseverance parameter, which increases the chances of selecting the action that was shown on the previous trial. It seems like the go bias parameter is the same as a typical persistence parameter, except that it only applies to go trials, and that its always active, rather than just when the previous action was also a “go” action. I am wondering if a more traditional persistence parameter would potentially fare better in the current dataset. This idea is especially tempting given that most behavioral studies have found higher proportions of stay trials in children compared to adults behaviorally (and more persistence computationally). Have the authors explored this possibility I was wondering if the initiation bias was related to a behavioral measure of staying?

Response: The go bias parameters used in this task are standard implementations in the literature (e.g., Raab & Hartley, 2020: <https://doi.org/10.1038/s41598-020-72628-w>), and because we wanted to avoid excessive model complexity, we did not add additional parameters that would capture some of the same information as the ‘go’ bias. We also felt that a stickiness parameter would not be appropriate given the deterministic outcomes and no-go design of this task. We therefore felt that a stickiness parameter did not make as much conceptual sense as the simpler go bias. We have now mentioned choice stickiness parameters in the Discussion:

“In addition, while our task was able to separate action initiation biases from learning processes, there were several other possible parameters that it was not practical or theoretically meaningful to assess using this design. For example, we did not include variable learning rates⁴², choice stickiness parameters⁴³, or the role of forgetting⁴¹, which could be measured in future research using an adapted learning task.”

I was also wondering what the optimal strategy was for the current task, and whether there was a way to quantify how close participants were to this optimal strategy? Apparently, because feedback is deterministic, an optimal strategy would set both reward and punishment learning to the maximum values of 1. (Why are participants' parameters so different?) And what is the optimal setting of the go parameter? It seems like it should be 1 at the first iteration, so that the outcome of each stimulus can be observed once (and then memorized); after the first repetition, the go bias should then drop to 0, so that the correct action can be chosen consistently and without bias. If indeed, the optimal strategy requires an adjustment of the go bias over time, this might be a valid mechanism to integrate into the model. Also related to the go bias parameter, it seems like the main function of this parameter is to shift the balance between go/nogo toward more go responses. I was wondering if other options had been explored to capture this effect? For example, it might be more succinct to think of this shift as a shift of the entire softmax function.

Response: As you state, the optimal strategy would be to respond to all objects on the first presentation and then respond only to reward objects on the subsequent presentations, i.e., an initial go bias (which was estimated as a free parameter) and high learning rates. In line with this, we observe a positive correlation between task accuracy and learning rate in our data, confirming that higher learning rates were better, and thus more optimal:

“Overall task performance (proportion of correct responses) was positively correlated with reward learning rate (Spearman's $r_{(696)} = 0.40 [0.33, 0.46]$, $p < .001$) and punishment learning rate (Spearman's $r_{(696)} = 0.68 [0.63, 0.72]$, $p < .001$) and negatively correlated with action initiation bias (Spearman's $r_{(696)} = -0.26 [-0.33, -0.19]$, $p < .001$).”

The implementation of the go biases was based on previous models (esp. Raab & Hartley, 2020) and thus we followed this literature rather than shifting the softmax function. We also tested a declining go bias (see our response to your comment below) but this did not outperform the constant go bias. We added details of this additional control model to our supplementary materials and note this in the main text:

“An additional control model, with a declining rather than constant action initiation bias, underperformed relative to our winning model and was therefore not considered further (see Supplementary materials).”

“As a control analysis we considered an additional model that accounted for the action initiation bias as declining over time, rather than a constant action initiation bias as in our winning model. The amount of decline with time was set as a free parameter. The winning model still outperformed the declining action initiation bias model (exceedance probability = 0.99, $\Delta\text{BIC}_{\text{int}} = -1371.82$) so we did not consider the additional model further.”

Given the low recoverability of the current parameter ρ , the authors might consider either

removing it from the model (and it would be interesting if doing so would improve the recoverability of other parameters as well), or modeling reward magnitude sensitivity differently. For example, it seems like negative outcomes cluster together, positive outcomes cluster together, and outcomes of 1 are distinct. Would recoding the outcomes in this way improve the model fit?

Response: Thank you for this suggestion. We have now indeed removed the rho parameter and selected model 5 as the winning model (see our responses to your previous comments). We show good to excellent recovery of all model parameters from our new winning model (see new Figure 3 reproduced above):

We did also consider recoding the point magnitudes, but concluded that since this parameter is in any case not central to our hypotheses, it would be better to remove it (i.e., select model 5) than code it in such a way that it might be difficult to compare to other parameters in the literature.

Many studies have also reported that learning rates change over time (e.g., Jepma, 2020) or depending on previous trial outcomes. Have the authors explored this possibility in the current study?

Response: We considered but rejected this option for the current study, due to concerns about excessive model complexity and because our task was deterministic rather than probabilistic. However, we certainly agree that a dynamic learning rate would be worth investigating in future studies with a suitably designed behavioural paradigm. We have mentioned this as a future direction in the Discussion along with some of the other parameters that you have suggested:

“In addition, while our task was able to separate action initiation biases from learning processes, there were several other possible parameters that it was not practical or theoretically meaningful to assess using this design. For example, we did not include variable learning rates⁴¹, choice stickiness parameters⁴², or the role of forgetting⁴⁰, which could be measured in future research using a suitable probabilistic learning task, possibly with drifting rewards to capture behavioural variability.”

Overall, I was wondering how the authors determined the initial search space for their model, and the list of parameters to include. Was this based on a literature search of computational models for similar tasks? E.g., how is the current model related to the model in Guitart-Masip et al., (2012), which seems to provide a better fit to participant data in a similar task (see below)?

Response: We based our choice of parameters and their values on the previous literature where possible, especially Raab & Hartley (2020) and Guitart-Masip et al. (2012), given that these tasks were closest to our own in terms of the design (i.e., including the go/no-go element). Our constant action initiation bias was based on the one used by Raab & Hartley (2020) and Guitart-Masip et al. (2012), and our initial action initiation bias likewise follows Guitart-Masip et al. (2012). The crucial differences between the Raab & Hartley and Guitart-Masip tasks is that those tasks

are probabilistic (80/20% probability of outcomes given the correct action or inaction), rather than deterministic as in our task. The second crucial difference is that their tasks have full overlap between the action and valence possibilities, i.e., they have conditions where no-go can result in rewards and go can result in avoidance of punishment. Because of these differences we did not include certain parameters, like a Pavlovian bias, which would be appropriate only when the full action/valence space is manipulated in the task. It is certainly something we are very keen to evaluate in new studies. The qualitative model fits from Guitart-Masip in fact represent the data from just a single participant, selected by the authors, and therefore are unlikely to be a robust representation of the model fit across all participants (Figure 2A-D in Guitart-Masip et al., 2012). Therefore, we do not feel that it is accurate to claim that their data provide a better fit than ours, where our modelled and simulated data overlap substantially and are based on simulated data from the same number of participants who took part in our experiment. We have updated our discussion section:

“First, our learning task did not contain ‘no-go to gain reward’ and ‘go to avoid punishment’ conditions, meaning that we were unable to assess Pavlovian action biases²⁵. It would be interesting for future studies to implement probabilistic learning tasks with the full action-valence crossover.”

I am also wondering if the go bias might change over time rather than be a stable value, given that the ratio of go / nogo actions declines with repetitions. Have the authors tested this possibility?

Response: Thank you for this suggestion. The ratio of go/no-go responses does decline across the task, but this seems to be mainly driven by punishment learning. Similar to our response to your previous comment, we selected our go bias parameters based on the most relevant previous literature as well as theoretical considerations about our task. However, we have now created a new model using a declining go bias and compared this model to the one with the constant go bias. The constant go bias model still outperformed the declining go bias model (exceedance probability = 0.99, $\Delta\text{BIC}_{\text{int}} = -1371.82$). We have added this additional analysis to our supplementary materials (also in response to your point above):

“As a control analysis we considered an additional model that accounted for the action initiation bias as declining over time, rather than a constant action initiation bias as in our winning model. The winning model still outperformed the declining action initiation bias model (exceedance probability = 0.99, $\Delta\text{BIC}_{\text{int}} = -1371.82$) so we did not consider the additional model further.”

The authors also note that “one consideration is whether action initiation biases are themselves influenced by the prospect of a rewarding outcome, since there are forms of impulsivity that occur specifically in situations where a possible reward is anticipated.” I think this is a very interesting hypothesis—has this been tested in the model? E.g., the prospect of a rewarding outcomes seems to be the value (i.e., the expected long-term cumulative reward) of the “go” action. In other words, the authors hypothesize that the go bias is a function of action value. This could be explored both behaviorally and computationally.

Response: We are glad that you found this idea interesting. We are also keen to explore this in future work, but unfortunately we do not think it is possible to examine this hypothesis using the current task due to the go/no-go design. In order to test this, we would need a new task or set of tasks that would allow us to compare go biases in the current learning context with a task where participants were not predisposed by the task design to respond for reward (e.g., perhaps incorporating a task similar to Raab & Hartley, 2020, or one where the number of reward objects fluctuates across clearly distinct blocks). In other words, it is the overall learning context rather than the reward history itself that interests us here, and we would need to manipulate this overall context rather than looking at the parameters within this task.

Going beyond individual parameters, I was wondering if the structure of the model might have to be adapted to capture human behavior better. For example, the authors use a classic RL update when participants choose the “go” action, but no update is performed after the “nogo” action. In words, this model assumes that participants do not learn in “nogo” trials. I was wondering if the model space could be explored more to establish whether this assumption is true. To explore this idea, the authors could first determine whether participant behavior still changes even when participants choose the “nogo” action; if so, the model could capture behavioral changes, e.g., by including an update based on the outcome “0”.

Response: We discussed and tested these different possible options at some length when designing our models. There were three main problems with learning after no-go trials. First, updating values after a no-go response did not make theoretical sense given that participants had to respond to get feedback; the task design explicitly assumes that responses are necessary for learning to occur. Second, at the beginning of the task the outcome values are not known, so we could not plausibly interpret early non-responses as losses or gains. To get around this we would have to treat neutral outcomes as gains/losses only after the participant has received feedback on the object value, which means we would be making implicit assumptions about learning rate in coding the outcomes (i.e., that the participant remembered the value). Third, if we were to treat no-go on reward trials as loss and go on punishment trials as gain (throughout the whole task), there is a symmetrical effect so that while the expected values are inflated, the choice probabilities are the same as in our current coding.

Model fitting

I also have a few concerns regarding the model fitting procedure. For one, individual differences in fitted parameters are very large, and I was wondering if full hierarchical Bayesian fitting would work better than the iterative approximation chosen here. Why did the authors choose the hierarchical expectation maximization approach? There is now a huge literature that shows that models improve tremendously when fitted hierarchically (e.g., Lee, 2011, JoMP; Katahira, 2016; JoMP), and empirical evidence is steadily growing as well (e.g., Brown, 2020, Biological Psychiatry; Pratt, 2020). For example, the issues in parameter recoverability might be resolved by choosing this fitting method (e.g., see Eckstein et al., 2022 DCN, supplemental section, comparing ML fit to hierarchical fit). An additional advantage would be that age differences in model parameters could be tested in an

unbiased way using this approach. Furthermore, the issue of large between-participant parameter variability (Fig. 3 and 4), and the need to exclude 41 participants with values more than three standard deviations from the mean, might be alleviated using this approach.

Response: Thank you for the opportunity to clarify. We selected the hierarchical maximisation approach because this approach is recommended by several researchers in the field (Nathaniel Daw, Quentin Huys, Sam Gershman) whom we developed our code from. In particular, this approach has been recommended when samples are large but number of trials is low, such as in developmental studies like ours. We agree that there are limitations to using maximum likelihood estimation, which we did not use in our study, opting instead for a hierarchical approach as suggested. Regarding the recovery, we now show good recovery for all parameters in our winning model. Furthermore, rerunning our analyses with the outlier participants included did not change any of our results.

In terms of exposure of results, I was missing a plot that shows raw parameters over continuous age.

Response: We have added the following figure to Supplementary materials:

Figure S8. Associations between model parameter values and age as a continuous variable. (a) Associations between punishment learning and age. Punishment learning rates increased with age. (b) Associations between action initiation bias and age. Action initiation biases decreased with age. (c) Associations between reward learning and age. Reward learning remained stable with age.

Writing / presentation

Relationship to prior work

The authors state that: “very little is known [...] about the robustness of previous findings.” Several studies have come out just in the last year with regard to this question that might resolve the issue, including Nussenbaum & Hartley (2019) and Eckstein et al. (2021, COBS), which address this particular issue for developmental populations, and are already cited in other places. Another paper that addresses this issue in a developmental population is Eckstein et al., (2021, biorxiv). More studies targeting the reliability of RL models more generally include Weidinger & van den Bos (2019), Brown et al., (2020), Shahar et al., (2019), Pratt et al., (2020), and Waltmann et al. (2022)

Response: Thank you for highlighting this literature. The question of robustness of modelling results is different from robustness in the adolescent learning literature generally, due to the issues with parameter values depending heavily on task design as highlighted in the papers you cite. Because our comment was intended to address robustness based on sample sizes and general sparsity of literature rather than task designs, we now feel it was confusing to make a statement about robustness in this context, and have removed it from the text. We have also added the suggested references to our manuscript.

The authors state: “To our knowledge, only one study to date has measured learning in a task design that incorporates requirements both to learn and also to inhibit actions.” This is slightly misleading as it implies that no study at all has studied this, whereas it only refers to developing populations. E.g., Guitart-Masip et al. have published several studies using a similar task and model.

Response: Thank you for highlighting this. We have now amended the text to make clear that we are referring specifically to adolescents here:

“To our knowledge, only one study of adolescents to date has measured learning in a task design that incorporates requirements both to learn and also to inhibit actions²⁵ (see²⁶ for a study in adults).”

In a similar vein, the authors state: “There has been a relative paucity of computational modelling work focusing on learning in adolescence” and “reward- and punishment-guided behaviour in adolescence is not well understood.” I would soften these statements. It is unclear at what these behaviors would be “understood”, and there have now been several reviews on this topic (including Nussenbaum & Hartley), there is a dedicated journal (DCN), and a conference (Flux) that likely provide ample references.

Response: We have amended the text to make clear that ‘relative paucity’ is in comparison to the adult literature and to state that reward- and punishment-guided behaviour in adolescence is not “yet fully understood”.

“However, reward- and punishment-guided behaviour in adolescence is not yet fully understood.”

“There has been a relative paucity of computational modelling work focusing on learning in adolescence compared to the adult literature, and previous studies were usually not designed to distinguish between learning processes and action initiation biases (a bias to make a response, regardless of its expected outcome value).”

Another concern regards statements that seem to “oversell” computational modeling. For example: “Here, we use computational modelling to distinguish between learning processes (...) and action initiation or ‘go’ biases (...). We test whether these different mechanisms are separable...” Recent work has suggested that computational modeling might not be able to answer questions of this kind (e.g., Eckstein et al., 2021 COBH), and I would encourage the authors to provide more nuance here and in other parts of the paper that share this sentiment.

Response: Thank you for highlighting this. It is not our intention to oversell computational modelling, and we are aware that parameters are often specific to particular learning contexts. However, we do feel that these caveats are somewhat separate from the ability of a modelling approach to distinguish between different computational mechanisms within a task, which is what we wanted to express here. Since distinguishing the action initiation bias from learning is key to our study aim, we are reluctant to rephrase this sentence because it might reduce the clarity of the text. However, we have softened our language throughout the manuscript to emphasise the context-specific nature of our results, which we hope addresses your concerns.

Similarly, the paper sometimes seems to imply that the action initiation bias should replace previous modeling approaches or that previous studies were problematic because they did not include such a parameter (e.g., “They suggest that theoretical accounts positing heightened responses to reward in adolescence should consider differences in impulsive action initiation rather than reward sensitivity or learning.”). I would encourage the authors to consider that most prior RL studies did not include conditions in which such a parameter could be added (e.g., go versus nogo trials), just like the current study did not include a variety of conditions in which other parameters could be added that have been shown to be critical in prior studies (e.g., attention, episodic memory, model-based RL, working memory, etc.).

Response: Thank you for highlighting this. We agree with your assessment and did not want to imply that our task was superior to previous ones, which test learning in a different context. In addition to the changes that we made in response to earlier comments, which we hope also goes some way to addressing this point, we have amended the Discussion as follows:

“They suggest that theoretical accounts positing heightened responses to reward in adolescence should consider differences in impulsive action initiation rather than solely reward sensitivity or learning, since these can mimic reward learning in some contexts.”

I do not see how the conclusions follow from the premises in the following statements:

- “Moreover, for all parameters where we observed differences across development, we saw the same associations when considering pubertal stage. This further suggests that these differences are part of the developmental process, rather than only a reflection of chronological age.” If both models fit the data, how can one be ruled out at the cost of the other?

Response: We have rephrased as follows:

“Moreover, for all parameters where we observed differences across development, we saw the same associations when considering pubertal stage, suggesting that our findings were robust across different measures of development capturing pubertal stage as well as chronological age.”

- “First, in contrast to the classic go/no-go paradigm (where ‘go’ responses are required substantially more often than no-go responses), our task used equal numbers of go-for-reward and no-go-for-punishment trials. This means that ‘go’ responses were not particularly associated with reward in this context.” If “go” is the only response that can possibly lead to reward, why is there no association between go and reward?

Response: We apologise for the lack of clarity. It is of course true that ‘go’ actions lead to rewards, but this should be captured by the reward learning rate insofar as the behaviour is contingent on the outcome. We meant that simply making ‘go’ responses in this task, regardless of the stimulus, would not result in more rewards than punishments (as would be the case in the classic go/no-go task). We have rephrased as follows:

“First, in contrast to the classic go/no-go paradigm (where ‘go’ responses are required substantially more often than no-go responses), our task used equal numbers of go-for-reward and no-go-for-punishment trials. Participants who make ‘go’ responses blindly (i.e., in the absence of learning) are therefore equally likely to receive punishments as to receive rewards. Consequently, the classic go/no-go element of being ‘primed’ to make responses to gain points (because ‘go’ responses are more often correct) is missing from this task.”

Potential for more in-depth discussion

I was also interested to hear why the authors think that the go bias parameter in this study might be related to impulsivity. It seems that a go bias is necessary to perform well in this task (i.e., because only the go action can lead to feedback, it is rational and advisable to systematically prefer the go action over the nogo, especially in the beginning of the task).

Response: Thank you for the opportunity to expand. An initial go bias should indeed be an optimal strategy, but the constant go bias is not, as confirmed experimentally by a negative correlation between the constant go bias and task performance ($r = -0.26 [-0.33, -0.19], p < .001$). In fact, impulsive behaviour can be adaptive in some contexts (Burnett Heyes et al., 2012), but we considered this behaviour impulsive in a detrimental way because it is non-adaptive responding without regard to consequences.

In terms of the overall conclusion drawn about the results of this study, it seems inconsistent to state that there is learning from reward when behavioral analyses indicate no learning from reward. Improving the computational model will hopefully resolve this issue, such that behavioral and modeling analyses will lead to the same conclusion.

Response: Apologies for the lack of clarity. There is learning from reward, as demonstrated by participants correctly making go responses for reward significantly above chance in the experiment. We have now clarified this in the manuscript:

“We first examined whether learning occurred during the task. A generalised linear mixed model (GLMM) (predicting correct responses from age, stimulus repetition

number, outcome valence, and covariates; see Methods) revealed a significant main effect of stimulus repetition on the number of correct responses made, with performance improving throughout the task (Odds ratio (OR) = 1.19 [1.17, 1.21], $z = 18.56$, $p < .001$). To confirm that learning occurred for both reward and punishment stimuli, we repeated the analysis using only reward trials, and found a significant main effect of repetition on correct responses (OR = 1.13 [1.09, 1.16], $z = 8.48$, $p < .001$). We then repeated the analysis using only punishment trials and again found a significant main effect of repetition on correct responses (OR = 1.28 [1.26, 1.32], $z = 19.65$, $p < .001$). Thus, the task was able to capture learning behaviour in both reward and punishment conditions.”

There were a few section in which I wish the authors addressed the implications of their findings, e.g.:

- “However, the action initiation bias we observed appears to be a genuine action bias, rather than a deliberate strategy or an indirect effect of reward facilitating action. “ How did the authors reach that conclusion?

Response: This relates to our argument about the action initiation bias not appearing to be facilitated by reward, and the fact that the higher action biases were associated with poorer performance. We hope that the changes made in response to your earlier comment (copied below) has clarified how we reached this conclusion:

“First, in contrast to the classic go/no-go paradigm (where ‘go’ responses are required substantially more often than no-go responses), our task used equal numbers of go-for-reward and no-go-for-punishment trials. Participants who make ‘go’ responses blindly (i.e., in the absence of learning) are therefore equally likely to receive punishments as to receive rewards. Consequently, the classic go/no-go element of being ‘primed’ to make responses to gain points (because ‘go’ responses are more often correct) is missing from this task.”

- “Together, our findings suggest that the tendency to initiate actions and learn from punishment shifts from late childhood across adolescence” Can the authors say more about the mechanism behind this transition?

Response: This sentence is from the opening paragraph of the Discussion, and we hope that the main body of the Discussion addresses this point to some extent, especially with the changes we have made in response to previous comments. We were reluctant to speculate excessively about mechanisms behind this transition, given that we do not include measures that could speak directly to this. However, it is likely that life experience is an important factor alongside the developmental pathways we already mention, and we have now added this to the text:

“It is plausible that adolescent-onset psychopathologies, such as certain forms of conduct disorder, represent aberrant developmental pathways, in which these normative increases in punishment learning and declines in action initiation biases could be disrupted. This would be consistent with the symptoms of conduct disorder, which include aberrant learning from punishment and impulsivity³⁶ Life experience is also an important factor to consider here, alongside biological changes³⁷. Even in healthy adolescents, a better understanding of the difference between reward-

oriented and impulsive behaviour could potentially facilitate behavioural interventions designed to reduce risky behaviour. For example, there may be contexts where it is more beneficial to focus on planning and impulse control than on learned behaviour, even when risky behaviour appears superficially to be driven by desirable outcomes (e.g., social status or material goods).”

- “It is plausible that adolescent-onset psychopathologies represent aberrant developmental pathways, in which these normative increases in punishment learning and declines in action initiation biases are disrupted.” It would be interesting to explain more explicitly, or with examples, what the authors had in mind here. Which (neural) pathways? Which psychopathologies?

Response: We specifically had in mind behavioural disorders such as conduct disorder, which are associated with reduced punishment learning and often onset during adolescence. We have now mentioned this in the text:

“It is plausible that adolescent-onset psychopathologies, such as certain forms of conduct disorder, represent aberrant developmental pathways, in which these normative increases in punishment learning and declines in action initiation biases could be disrupted. This would be consistent with the symptoms of conduct disorder, which include aberrant learning from punishment and impulsivity³⁶.”

- “Our findings also demonstrate these associations robustly by testing a large and geographically diverse sample.” I am wondering if testing a large sample automatically implies robust findings. The error bars in model estimates might suggest otherwise. Including participants from many different backgrounds might also induce additional noise. Can the author explain this sentiment a little more?

Response: Thank you for the opportunity to expand. We meant that it is robust in the sense that it might be more representative of adolescents in general than smaller sample studies which might reach different conclusions amongst them. Even if more noise is introduced, we do show clear evidence for our effects and Bayesian evidence to support null effects. We have rephrased the below to highlight the point about representation:

“Our findings also demonstrate these associations robustly by testing a large and geographically diverse, and therefore potentially more representative, sample.”

Minor

In the introduction, can the authors define “action initiation biases”, just like they define RL?

Response: We have now added a clearer definition where we first introduce the term:

“... and action initiation or ‘go’ biases (initiating actions impulsively or ‘blindly’, without regard for consequences).”

I was wondering why the authors chose to divide participants into 3 age bins, but 5 puberty bins? Do the results depend on the number of bins chosen? It might aid direct comparison to have the same number of bins both analyses.

Response: Age was a continuous variable in all our analyses, so for presentation purposes, we selected three bins as the minimum needed to show any quadratic relationships that we might have found. By contrast, the pubertal development measure is itself scored according to these five bins (i.e., this is the 'raw' data for puberty), and we therefore chose to keep the standard scoring rather than collapse across categories and lose potentially important information.

We can however confirm that collapsing the pubertal status scores into three categories (pre/early, mid, and late/post pubertal) did not change our results. We still observed main effects of puberty on the number of correct responses (OR = 1.06 [1.03, 1.11], $z = 3.72$, $p < .001$), a puberty by repetition interaction (OR = 1.02 [1.01, 1.04], $z = 2.60$, $p = .009$), and a puberty by valence interaction (OR = 1.17 [1.13, 1.22], $z = 6.48$, $p < .001$), but no puberty by repetition by valence interaction (OR = 1.02 [0.98, 1.05], $z = 0.89$, $p = .37$). We have added a note to this effect in the main text and added these results to Supplementary materials:

“To confirm that the lack of pubertal stage by repetition by valence interaction was not a reflection of the number of pubertal categories, we repeated the analysis with pubertal stage collapsed into three categories (pre/early, mid, and late/post-pubertal), and observed no differences (see Supplementary materials).”

Supplementary materials:

“As an additional control analysis, we collapsed the pubertal status scores into three categories (pre/early, mid, and late/post pubertal) to reflect the three age categories used for visualisation. Collapsing the pubertal stage data in this way did not change our results. Consistent with the analysis using the original five stages of the PDS measure, we still observed main effects of puberty on the number of correct responses (OR = 1.06 [1.03, 1.11], $z = 3.72$, $p < .001$), a puberty by repetition interaction (OR = 1.02 [1.01, 1.04], $z = 2.60$, $p = .009$), and a puberty by valence interaction (OR = 1.17 [1.13, 1.22], $z = 6.48$, $p < .001$), but no puberty by repetition by valence interaction (OR = 1.02 [0.98, 1.05], $z = 0.89$, $p = .37$).”

In the sentence: “Part of this variability could be due to different task demands”, Eckstein et al. (2021, COBS) and Eckstein et al., (2021, biorxiv) might be other relevant citations.

Response: Thank you for these suggestions, which we have now added to the manuscript.

I did not fully understand which methods were used in the following analyses: “To confirm the strength of these associations, and obtain strength of evidence for any null effects, we calculated Bayes factors using the BIC method and linear mixed effects regression models, with age removed from the null model. We observed very strong evidence for the associations between age and punishment learning rate (BF₁₀ = 336.00, BF₀₁ = 0.003) and

between age and action initiation bias ($BF_{10} = 6545.00$, $BF_{01} = 0.0002$)” Could the authors provide additional explanation? Specifically, how did they test age effects after removing age?

Response: We apologise that this was not clear. We have amended the text as follows to make this clearer:

“To confirm the strength of these associations, and obtain strength of evidence for any null effects, we calculated Bayes factors using the BIC method³³. For each model parameter of interest, we compared two linear mixed effects regression models: our ‘standard’ model, predicting the model parameter from age and covariates, and a ‘null’ model, which predicted parameter values from the covariates only. We then calculated the BIC of each of these two models, and used the difference between the BICs to calculate a Bayes factor³³. We observed decisive evidence for the associations between age and punishment learning rate (i.e., the model including age was a better fit; $BF_{10} = 167.25$, $BF_{01} = 0.01$) and between age and action initiation bias ($BF_{10} = 8926.69$, $BF_{01} = 0.0001$). In contrast, there was no evidence for associations between age and reward learning rate ($BF_{10} = 0.07$, $BF_{01} = 13.60$, strong evidence in support of the null).”

For Figures 3 and 4, I was wondering if the data might be summarized better in terms of the median and bootstrapped confidence intervals, rather than the mean and standard deviation, given that data points are not distributed normally (see Figure 4 below)? This would also make the analysis more resistant to outlier values.

Response: We did explore plotting with medians, but the figures looked very similar, and thus we chose to continue using the means, since the age bins in these figures are in any case only for presentation purposes. We reproduce the alternative figure for punishment learning along with the figure in the main text to demonstrate:

Why does Table S1 use Pearson's correlation, but table S2 Spearman's?

Response: This is because Table S1 (now re-numbered as Table S3) uses data where the parameter values are not normally distributed. We have now clarified this in the table heading.

This sentence seems to miss a verb? "Internalising problems are likewise associated with difficulties in reinforcement learning, and social media use, which can become problematic for some adolescents, and has recently been linked to reward learning mechanisms."

Response: We have rephrased for clarity:

"Internalising problems are likewise associated with difficulties in reinforcement learning^{9,10}, and social media use (which can become problematic for some adolescents¹¹) has recently been linked to reward learning mechanisms¹²."

"Computational modelling of learning typically uses reinforcement learning models": There also is a huge (equally-sized?) literature on Bayesian Inference, e.g., Gopnik, Griffiths, Gershman

Response: We have amended the text as follows:

"Computational modelling of learning can be conducted using reinforcement learning models"

The above figure should include information about trial timing.

Response: The information about trial timing is included in the figure legend and we have now added this to the plot as well:

The reference list includes both “Sutton RS, Barto AG. Reinforcement Learning: An Introduction.

MIT Press; 2011.” and “Sutton RS, Barto AG. Reinforcement Learning, Second Edition: An Introduction. MIT Press; 2018. “

Response: Thank you for pointing this out. We have removed the duplicate reference.

The following figure might be better served using a continuous color scheme:

(This is the figure with the learning curves split by object value).

Response: This has been done.

REVIEWER COMMENTS

Reviewer #1 (Remarks to the Author):

I would like to thank the authors for answering all my questions with such care and in a detailed manner. Very much liked the paper, and now even more!

Reviewer #2 (Remarks to the Author):

The authors have been responsive to the previous comments.

Reviewer #3 (Remarks to the Author):

I appreciate the improvements the authors have made to their manuscript based on the reviewer comments. However, I believe that some of the most severe issues - raised by me as well as by other reviewers - have not been addressed adequately. For some of these issues, the language of the paper is vague - or even misleading - as to the obtained findings. In advance of publication, I believe that both the model and the coherence between statistical results and the verbal conclusions drawn from these results need to be improved.

In terms of modeling, there is a striking, qualitative discrepancy between human and model behavior (see first Fig. 6 [there are two Fig. 6's in the revised manuscript; I am referring to the first one here]). Even though it is impossible (and probably undesirable) to create a model that reproduces the to-be-modeled behavior identically in all its aspects, the main characteristics of the to-be-modeled data need to be captured for a model to be interpretable. If a model does not capture the data it is supposed to model, fitted model parameters are not interpretable because they do not reflect the modeled data.

In the current dataset, human endorsement of the “go” action shows a prominent inverse-U function over repetition number: On the first trial, on average, humans seem to endorse the “go” action in about 60 percent; this percentage jumps up to around 80 percent on the second trial; after

this, the responses to rewarded and unrewarded stimuli start diverging: For unrewarded stimuli, the percentage of “go” responses decreases continually until it reaches about 50 percent at the last, 10th repetition; however, for rewarded actions, the percentage stays relatively constant, around 75 percent all the way until the last repetition.

This pattern is fascinating and opens the possibility for a number of cognitively interesting questions (e.g., why does the percentage start low and then jump up abruptly? What cognitive process drives this increase, what is learned, and based on what information? Next, what drives the subsequent decrease in the non-rewarded condition? Based on which information and cognitive processes does this behavioral change take place? Furthermore, what does the relative lack of behavioral change in the rewarded condition signify? Is no learning taking place, or is learning taking place despite an absence of behavioral change?). The authors (verbally) evoke some interesting hypotheses to explain these patterns, but the cognitive model does not set out to test them, and also does not capture the human behavior: In the rewarded condition, the model’s percentage of “go” responses starts off low (around 70 percent), then linearly increases (up to about 90 percent in the last iteration), lacking the prominent inverse-U shape in the human data (the model does not have the capacity to create U-shapes). For the non-rewarded condition, the percentage of “go” responses starts high (in contrast to humans), and then linearly decreases (also in contrast to humans).

In other words, the model does not capture the fundamental features of human behavior. Such an initial result is not uncommon when creating a new computational model, or when applying an existing model to a new behavioral task (as the authors of this study did). What is required in this case as a next step is to iterate on the model until it captures human behavior. Until this step has been performed, the model should not be interpreted because it does not capture human behavior. In other words, the space of computational models (see second Fig. 6) that has been explored for this paper is not yet sufficient to have resulted in an appropriate model. I have provided several possible directions for improving the model in my last review (e.g., adding memory / forgetting parameters; outcome-independent choice stickiness; using the optimal strategy as an alternative starting point or addition for the computational model; including time-varying or counter-factual learning parameters), and am adding a couple more this time (e.g., a model that learns from nogo trials using an uncertainty or curiosity parameter that biases a future “go” choice). If the authors prefer not to implement the suggested changes themselves, it might also be an option to collaborate with a computational lab to finalize the computational model.

My other main concern is the verbal description of key findings in the paper that sometimes over- or understate statistical results. To provide just one example, correlations between parameters are described as follows: “While there were significant correlations between our model parameters, the effect sizes of the correlations are mostly quite small”, referring to Table S4. However, two of the six entries (i.e., a third) in Table S4 show correlations of $r=0.58$ and $r=0.36$. The language above (“mostly quite small”) therefore seems to severely misrepresent the table. This is aggravated by the fact that most readers will not invest the effort to look up the table in the supplemental materials, and hence

obtain a wrong impression of the data. Overall, statements like these undermined my trust in the authors' intent to represent their statistical findings in an unbiased and accurate way.

The paper also evoked the impression in several places that the study introduces a novel task and/or novel computational model and/or novel research direction. I would recommend the authors elaborate more on related previous studies and ground their findings more closely in the existing literature.

The following points of my initial review have not yet been addressed adequately or have:

Are younger participants excluded more often than older ones?

Could the authors provide a histogram of the number of participants excluded, binned in yearly increments?

Are 3 seconds enough for the youngest participants?

Could the authors provide a scatterplot of raw (i.e., not averaged) response times over trials, separately for each 1-year age bin?

Was there an association between age and sex?

I do not understand how a t-test test can assess this question ("There was no association between age and sex in the final sample (2-tailed t-test: $t(488) = 1.00$, $p = .20$, $BF_{01} = 5.03$, substantial evidence in support of the null).")

Could the authors provide a histogram that shows the number of male and female participants in each 1-year age bin?

It appears that the main difference between the reward and punishment conditions is that go responses stay high in the reward condition, whereas they drop in the punishment condition.

Could the authors assess whether there is evidence for this hypothesis? E.g., is there a significant positive slope of go responses in the reward condition, excluding trial 1? (Table S2, rightmost columns seems to indicate this, but I find it surprising given the slopes look negative for all age groups in Fig. 6)

The authors claim that "this age-related learning improvement was specific to learning from punishment outcomes" (i.e., no learning in reward trials). However, the computational model shows

reward learning rates greater than 0 (i.e., significant learning in reward trials). This seems contradictory. I see two possible causes for this: 1) The behavioral regression model treats repetition number as a linear predictor, while the behavior is non-linear, implying that a quadratic term might fit better. 2) The computational model does not fit the data.

Lastly, due to the qualitative difference in responding to repetition 1 compared to later repetitions, I am wondering if repetition 1 might be considered part of participants' learning phase, and if it would clean up some results if the analyses were restricted to repetitions 2-10 only. This would likely improve the fits of the GLMs that contain repetition as a predictor because in the current data, the effect of repetition is highly non-linear, but it seems linear after trial 2. It might also help the computational model, which currently does not capture the quantitative aspects of human behavior.

Authors response: "Thank you for this suggestion. We have now re-run all our modelling and behavioural analyses using only data from repetitions 2-10. We can confirm that this did not change the age-related patterns we observed in our main results. The results of this analysis are presented in Table S2 (see above)"

Could the authors confirm that they fitted the computational model to repetitions 2-10, and show the resulting simulated model behavior, model fits, and model parameters?

My central concern with the current study is that the model does not seem to capture human behavior adequately. As shown in the plot below from the supplemental material, one of the most noteworthy patterns in participant behavior is the initial increase in "P(go)," which is completely absent from the model behavior. Whereas the model thereafter shows a similar decline in go responses in the punishment condition to humans, it shows a steady increase of go responses in the reward condition, which is absent in the human data. (Indeed, the behavioral analyses show a lack of learning from reward, which directly contradicts the model.) This suggests that the model did not indeed capture human behavior.

"First, we have increased the number of simulated participants to match the number of participants in our sample, and used the median parameter values, at the request of a reviewer. The new simulated data more closely follows our model fitted data from participant behaviour. "

These changes did not address my initial concern.

Furthermore, simulated participants should be based on actual participants' parameter values (i.e., one simulation per participant, using that participant's parameter values), rather than the population median. Using the population median artificially smoothes the resulting simulations, and disregards existing variability and parameter correlations.

"Second, we have plotted the simulated and fitted data separately for the three different age groups and showed reasonable qualitative model fits across all three groups. "

The model fits are still not "reasonable", in my opinion. The non-linear behavioral pattern is not captured. The stability in go responses and initial increase in both conditions are not captured.

“Third, we have repeated our model fitting and comparison procedure separately for the three age groups and again showed that the same model wins across all three age groups. “

This does not address my initial point.

“Fourth, we have repeated our behavioural analyses using only trials 2-10, and finally, we have repeated our model fitting and comparison procedure with trials 2-10. These analyses demonstrated exactly the same associations with age. “

Could the authors show the results of these analyses?

“Therefore, our model accurately captures behaviour. Additionally, we have reanalysed our parameter recovery and model identifiability with model 5, without the low recovery rho parameter, and shown recovery is good and stronger than our original reported results using model 6. Furthermore, we can still confirm exactly the same associations with age regardless of whether model 5 or 6 is selected as the winning model (see Table S2).“

This is true, however does not address my initial point.

“Together, all these new analyses robustly show that our winning model is a good and robust model of participant behaviour. “

I disagree with this statement. The model does not capture the major patterns of human behavior.

To determine this issue, it would be instructive to see scatterplots of the generated and recovered values of each parameter (e.g., see Wilson & Collins, 2019).

Could the authors provide these scatterplots?

It would also be helpful to see a correlation matrix of all the model parameters, to see if parameters traded off with each other within the model. For example, learning rates and decision temperature parameters are known to trade off with each other, with important caveats for model interpretation.

“Response: We have added these correlations to Supplementary materials. While there were significant correlations between our model parameters, the effect sizes of the correlations are mostly quite small. No correlations are substantial enough to cause concern...”

I would not call correlations up to $r=0.58$ and $r=0.36$ “mostly quite small”; this statement seems misleading.

These patterns suggest that the models are misspecified and might have to be parameterized differently to adequately capture human behavior (e.g., Katahira, 2015, JoMP). The extremely large

variability of parameter values between participants (despite hierarchical fitting) also supports the interpretation of a lack of model fit and potential model misspecification.

In order to address this issue, the authors could try different parameterizations of their initial model (i.e., expand on the second Fig. 6)

There are several ways in which the model could be improved. First, additional parameters could help explain additional variance, and resolve issues of model misspecification (e.g., Katahira, 2015, JoMP).

Memory parameters (WM, episodic memory, forgetting, persistence)

Time-varying parameters (outcome-scaling learning rate; decaying go bias)

Uncertainty / curiosity (e.g., nogo would increase curiosity, which would trigger subsequent go response): I am wondering if the mechanism behind the increase in “go” responses from the first to the second iteration is caused by the realization that nogo actions do provide any information. Could the authors estimate an uncertainty parameter that captures how much participants want to learn about each stimulus, and positively contributes to selecting the go action? E.g., on each iteration that a nogo action is executed for a stimulus, its uncertainty could increase; and uncertainty and expected value could play together in determining action selection. The existing literature on the observe-or-bet task might be relevant here as well, as participants need to choose between the action of “observing” (i.e., seeing the outcome of an action, but without winning / losing points) or “betting” (i.e., Winning / losing points, but without observing the action)

Different kind of model (e.g., optimal Bayesian observer)

I was also wondering what the optimal strategy was for the current task, and whether there was a way to quantify how close participants were to this optimal strategy? Apparently, because feedback is deterministic, an optimal strategy would set both reward and punishment learning to the maximum values of 1. (Why are participants’ parameters so different?) And what is the optimal setting of the go parameter? It seems like it should be 1 at the first iteration, so that the outcome of each stimulus can be observed once (and then memorized); after the first repetition, the go bias should then drop to 0, so that the correct action can be chosen consistently and without bias. If indeed, the optimal strategy requires an adjustment of the go bias over time, this might be a valid mechanism to integrate into the model. Also related to the go bias parameter, it seems like the main function of this parameter is to shift the balance between go/nogo toward more go responses. I was wondering if other options had been explored to capture this effect? For example, it might be more succinct to think of this shift as a shift of the entire softmax function.

“The implementation of the go biases was based on previous models (esp. Raab & Hartley, 2020) and thus we followed this literature rather than shifting the softmax function. We also tested a declining go bias (see our response to your comment below) but this did not outperform the

constant go bias. We added details of this additional control model to our supplementary materials and note this in the main text:“

As far as I remember, the Raab & Hartley, 2020 task was different from the current paper because it was stochastic rather than deterministic? If so - and even if not - it is perfectly legitimate to try out different models, and I would encourage the authors to do so.

As for the declining go bias, I could not find any information about how the decline was implemented (e.g., linear decline? Quadratic decline? Exponential decay?). This choice could significantly impact the outcome. Given that the misfit of the current model to the data hinges on the fact that the model does not show non-linear trajectories, carefully trying out a parameterization that allows for non-linear behavior seems worthwhile.

Many studies have also reported that learning rates change over time (e.g., Jepma, 2020) or depending on previous trial outcomes. Have the authors explored this possibility in the current study?

Response: We considered but rejected this option for the current study, due to concerns about excessive model complexity and because our task was deterministic rather than probabilistic.

If a model does not fit the data, different options need to be tried. Model complexity need not necessarily increase if the newly added mechanism can replace others currently present in the model. And if a more complex model fits the data better, the additional complexity is not an issue. Hence, model complexity is not an argument to not try out more models.

Going beyond individual parameters, I was wondering if the structure of the model might have to be adapted to capture human behavior better. For example, the authors use a classic RL update when participants choose the “go” action, but no update is performed after the “nogo” action. In words, this model assumes that participants do not learn in “nogo” trials. I was wondering if the model space could be explored more to establish whether this assumption is true. To explore this idea, the authors could first determine whether participant behavior still changes even when participants choose the “nogo” action; if so, the model could capture behavioral changes, e.g., by including an update based on the outcome “0”.

Response: We discussed and tested these different possible options at some length when designing our models. There were three main problems with learning after nogo trials. First, updating values after a no-go response did not make theoretical sense given that participants had to respond to get feedback; the task design explicitly assumes that responses are necessary for learning to occur. Second, at the beginning of the task the outcome values are not known, so we could not plausibly interpret early non-responses as losses or gains. To get around this we would have to treat neutral outcomes as gains/losses only after the participant has received feedback on the object value, which means we would be making implicit assumptions about learning rate in coding the outcomes (i.e., that the participant remembered the value). Third, if we were to treat no-go on reward trials as loss and go on punishment trials as gain (throughout the whole task), there is a symmetrical effect so

that while the expected values are inflated, the choice probabilities are the same as in our current coding.

I would argue that NoGo actions could provide a different kind of information from Go actions. I would recommend the authors first do careful behavioral analyses to understand what effect NoGo actions have on behavior; after this understanding has been achieved, the model should be adapted to include the hypothesized underlying mechanism.

What cognitive processes underlie the behavior, and can these be tested behaviorally (in support of the computational model)?

For example, how large is the increase in responding between the first and second repetition? Is it significant? Is the overall shape linear or nonlinear (e.g., fitted better by a quadratic function than linear)? (How) do rewards and punishments affect the behavior? Do participants stay with responses (e.g., go for stim1) after observing positive outcomes (e.g., reward after go) while switching responses (e.g., nogo for stim2) after observing negative outcomes (e.g., punishment after go)? (This would be the “The other option, to conduct the same analysis at the level of individual stimuli (e.g., so that the ‘previous trial’ was the previous presentation of the same stimulus rather than the previous trial in time)”))

Response to Reviewers' comments R2: "Action initiation and punishment learning differ from childhood to adolescence while reward learning remains stable"

Reviewer #1 (Remarks to the Author):

R1.1. I would like to thank the authors for answering all my questions with such care and in a detailed manner. Very much liked the paper, and now even more!

Response: Thank you so much for your positive feedback about our response to your helpful suggestions. We are delighted that you like the paper even more. We really appreciate your time reviewing our work.

Reviewer #2 (Remarks to the Author):

R2.1. The authors have been responsive to the previous comments.

Response: Thank you so much for your time and positive feedback about our response to your helpful comments.

Reviewer #3 (Remarks to the Author):

I appreciate the improvements the authors have made to their manuscript based on the reviewer comments. However, I believe that some of the most severe issues - raised by me as well as by other reviewers - have not been addressed adequately. For some of these issues, the language of the paper is vague - or even misleading - as to the obtained findings. In advance of publication, I believe that both the model and the coherence between statistical results and the verbal conclusions drawn from these results need to be improved.

Response: Thank you for your time providing feedback on our work. We hope to have been able to address your remaining queries surrounding the model, which now closely reproduces participant behaviour including stimulus repetitions 2-10 as suggested, and we have worked to improve the coherence between statistical results and verbal conclusions.

R3.1 In terms of modeling, there is a striking, qualitative discrepancy between human and model behavior (see first Fig. 6 [there are two Fig. 6's in the revised manuscript; I am referring to the first one here]). Even though it is impossible (and probably undesirable) to create a model that reproduces the to-be-modeled behavior identically in all its aspects, the main characteristics of the to-be-modeled data need to be captured for a model to be interpretable. If a model does not capture the data it is supposed to model, fitted model parameters are not interpretable because they do not reflect the modeled data.

In the current dataset, human endorsement of the "go" action shows a prominent inverse-U function over repetition number: On the first trial, on average, humans seem to endorse the "go" action in about 60 percent; this percentage jumps up to around 80 percent on the second trial; after this, the responses to rewarded and unrewarded stimuli start diverging: For unrewarded stimuli, the percentage of "go" responses decreases continually until it reaches about 50 percent at the last, 10th repetition; however, for rewarded actions, the percentage stays relatively constant, around 75 percent all the way until the last repetition.

This pattern is fascinating and opens the possibility for a number of cognitively interesting questions (e.g., why does the percentage start low and then jump up abruptly? What cognitive process drives

this increase, what is learned, and based on what information? Next, what drives the subsequent decrease in the non-rewarded condition? Based on which information and cognitive processes does this behavioral change take place? Furthermore, what does the relative lack of behavioral change in the rewarded condition signify? Is no learning taking place, or is learning taking place despite an absence of behavioral change?). The authors (verbally) evoke some interesting hypotheses to explain these patterns, but the cognitive model does not set out to test them, and also does not capture the human behavior: In the rewarded condition, the model's percentage of "go" responses starts off low (around 70 percent), then linearly increases (up to about 90 percent in the last iteration), lacking the prominent inverse-U shape in the human data (the model does not have the capacity to create U-shapes). For the non-rewarded condition, the percentage of "go" responses starts high (in contrast to humans), and then linearly decreases (also in contrast to humans).

In other words, the model does not capture the fundamental features of human behavior. Such an initial result is not uncommon when creating a new computational model, or when applying an existing model to a new behavioral task (as the authors of this study did). What is required in this case as a next step is to iterate on the model until it captures human behavior. Until this step has been performed, the model should not be interpreted because it does not capture human behavior. In other words, the space of computational models (see second Fig. 6) that has been explored for this paper is not yet sufficient to have resulted in an appropriate model. I have provided several possible directions for improving the model in my last review (e.g., adding memory / forgetting parameters; outcome-independent choice stickiness; using the optimal strategy as an alternative starting point or addition for the computational model; including time-varying or counter-factual learning parameters), and am adding a couple more this time (e.g., a model that learns from nogo trials using an uncertainty or curiosity parameter that biases a future "go" choice). If the authors prefer not to implement the suggested changes themselves, it might also be an option to collaborate with a computational lab to finalize the computational model.

Response: Thank you for your detailed feedback on our computational model. We would like to note that several of the authors on the manuscript are experts in fitting computational models to behavioural data (e.g. Dr Pauli, Dr Klein-Flugge, Dr Lockwood, Dr Brazil), including writing highly cited tutorial based reviews on best practices (Lockwood & Klein-Flugge, 2021) and we hope that this assumption that we would need a 'computational lab' to finalise the model does not introduce any bias in the following comments.

As detailed in our previous response, we did try a range of plausible models, constrained based on the experimental task and the participant behaviour we observed. Of course, there has to be a limit to the model space, and it should be informed both theoretically and empirically. There were several converging reasons why we selected our winning model, and our selection was far from arbitrary:

- (1) Parameters from the model meaningfully correlated with task accuracy in the expected directions (higher learning rates were related to higher task accuracy, and lower go biases related to higher task accuracy).
- (2) The main parameters of interest from the model (learning rates and go biases) showed meaningful associations with age regardless of whether other parameters were included or omitted.
- (3) Regardless of whether we selected model 6, model 7, or modelled only stimulus repetitions 2-10, the resulting parameters had identical associations with age, suggesting that they can indeed be meaningfully interpreted.
- (4) We found that the same model provided the best explanation of the data regardless of whether we modelled all stimulus repetitions or only stimulus repetitions 2-10.
- (5) The winning model showed good parameter recovery and model identifiability.
- (6) The winning model had the highest exceedance probability and one of the lowest BIC values.

However, we agree that the correspondence between model simulated data and participant data could be improved, although we note many studies where the simulations were comparable to ours (e.g. Guitart-Masip, 2011; Callaway et al., 2021), and, as you say, it is “impossible (and probably undesirable) to create a model that reproduces the to-be-modelled behavior identically in all its aspects”.

We have now used the winning model to simulate behaviour from stimulus repetitions 2-10, as suggested in your comment R3.6. Thank you for the suggestion. As expected, based on the pattern from our initial simulations, considering stimulus presentation 1 as part of the learning phase significantly improves the correspondence between simulated and actual behaviour (Figure 6 below). We have already shown that the winning model with stimulus repetitions 2-10 has exactly the same associations with age as when we include all stimulus repetitions, and therefore this analysis further corroborates our choice of model that we selected for reasons 1-6 above.

We have now updated the manuscript to include this new analysis and figure, which we hope addresses your remaining concerns surrounding one aspect of model fit that could be improved. To reiterate, including only stimulus repetitions 2-10 does not change any of our main conclusions surrounding associations between age and model parameters, and the same model wins in the model space. The results of all analyses including stimulus repetitions 2-10 were previously shown in Table S2 and in Figure 7. In the revised manuscript, we have presented these results more fully; please see Figure 6, Figure S7, Figure S8, and Table S2. We have also reproduced Figure 6 and Table S2 again below. (Previously, this figure showed simulations separately by age category; these results have now been moved to Figure S6 in the supplement.)

The simulations using stimulus repetitions 2-10 only are now introduced as follows in the manuscript:

“To further confirm the accuracy of our model, we generated synthetic behavioural data for 742 ‘participants’ using the winning model and its median parameter values. As a control analysis, we also generated synthetic data using the version of the winning model with only stimulus repetitions 2-10 included (Figure 6).”

“In both the main 742-participant sample (with and without the first stimulation presentation) and the aged-based samples, the simulated responses fell within a similar range to the real responses. They followed a somewhat similar trajectory, particularly when omitting the first stimulus presentations and considering them as part of the practice phase (Figure 6 and Figure S11). Importantly, parameter associations with age were identical whether we included all trials or only repetitions 2-10 (Table S2 and Figure S8).”

Figure 6. Simulated probability of 'go' response to reward and punishment stimuli across stimulus repetitions. (a) Simulated and real probability of 'go' responses on reward trials, with simulated data generated using the winning model and its median parameter values for the full sample (N = 742) across all 10 stimulus repetitions. (b) Simulated and real probability of 'go' responses on punishment trials for the full sample (N = 742) across all 10 stimulus repetitions. (c) Simulated and real probability of 'go' responses on reward trials for the full sample (N = 742) across stimulus repetitions 2-10 only (d) Simulated and real probability of 'go' responses on punishment trials for the full sample (N = 742) across stimulus repetitions 2-10 only. This Figure relates to Figure S11.

Table S2. Summary of additional analyses

	Main analyses	With model 6	With all eligible participants	With all eligible and outliers	With repetitions 2-10 only
Sample size					
N	742	742	832	832	742
Behavioural analysis: GLMM predicting no. correct responses from age					
Main effect stimulus repetition	OR = 1.19 [1.17, 1.21], $z = 18.56$, $p < .001$)	N/A	OR = 1.17 [1.15, 1.19], $z = 19.01$, $p < .001$	N/A	OR = 1.15 [1.13, 1.17], $z = 13.87$, $p < .001$
Age*repetition	OR = 1.02 [1.01, 1.04], $z = 2.49$, $p = .01$)	N/A	OR = 1.03 [1.01, 1.04], $z = 2.97$, $p = .003$	N/A	OR = 1.02 [1.00, 1.04], $z = 2.06$, $p = .04$
Main effect age	(OR = 1.08 [1.04, 1.11], $z = 4.58$, $p < .001$)	N/A	OR = 1.07 [1.04, 1.10], $z = 4.92$, $p < .001$	N/A	OR = 1.08 [1.04, 1.12], $z = 4.48$, $p < .001$
Age*repetition*valence	OR = 1.09 [1.05, 1.13], $z = 4.65$, $p < .001$)	N/A	OR = 1.10 [1.06, 1.14], $z = 5.56$, $p < .001$	N/A	OR = 1.07 [1.03, 1.12], $z = 3.32$, $p < .001$
BF ₀₁ for (no) age difference in reward learning	57.80	N/A	29.11	N/A	112.00
Modelling: model comparison					
Overall winning model	5	N/A (6)	5	5	5
Modelling: GLMMs predicting parameter values from age					
Main effect of age on punishment learning	$\beta = 0.10$ [0.05, 0.15], $z = 4.12$, $p < .001$	$\beta = 0.10$ [0.05, 0.15], $z = 4.26$, $p < .001$	$\beta = 0.11$ [0.07, 0.15], $z = 4.86$, $p < .001$	$\beta = 0.05$ [0.03, 0.07], $z = 5.40$, $p < .001$	$\beta = 0.12$ [0.07, 0.17], $z = 4.59$, $p < .001$
Main effect of age on action initiation bias	$\beta = -0.20$ [-0.28, -0.12], $z = -4.91$, $p < .001$	$\beta = -0.20$ [-0.28, -0.12], $z = -4.78$, $p < .001$	$\beta = -0.20$ [-0.27, -0.12], $z = -5.30$, $p < .001$	$\beta = -0.17$ [-0.23, -0.10], $z = -5.14$, $p < .001$	$\beta = -0.15$ [-0.23, -0.06], $z = -3.44$, $p < .001$
Main effect of age on reward learning rate	$\beta = 0.01$ [-0.05, 0.07], $z = 0.30$, $p = .77$	$\beta = 0.01$ [-0.06, 0.07], $z = 0.17$, $p = .86$	$\beta = 0.02$ [-0.03, 0.08], $z = 0.80$, $p = .42$	$\beta = 0.03$ [-0.01, 0.07], $z = 1.29$, $p = .20$	$\beta = 0.03$ [-0.05, 0.11], $t = 0.73$, $p = .46^*$
BF ₀₁ for (no) age difference in reward learning rate	13.60	12.62	12.62	11.57	20.00
Main effect of age on temperature parameter	$\beta = 0.002$ [-0.07, 0.07], $z = 0.08$, $p = 0.94$	$\beta = 0.002$ [-0.07, 0.08], $z = 0.06$, $p = 0.95$	$\beta = -0.03$ [-0.10, 0.03], $z = -0.96$, $p = 0.34$	$\beta = -0.03$ [-0.10, 0.04], $t = -0.85$, $p = 0.39^*$	$\beta = 0.06$ [-0.01, 0.14], $z = 1.70$, $p = .09$

*These statistics are from a standard linear model without site, due to non-convergence of the mixed (GLMM) model due to model complexity

R3.2. My other main concern is the verbal description of key findings in the paper that sometimes over- or understate statistical results. To provide just one example, correlations between parameters are described as follows: “While there were significant correlations between our model parameters, the effect sizes of the correlations are mostly quite small”, referring to Table S4. However, two of the six entries (i.e., a third) in Table S4 show correlations of $r=0.58$ and $r=0.36$. The

language above (“mostly quite small”) therefore seems to severely misrepresent the table. This is aggravated by the fact that most readers will not invest the effort to look up the table in the supplemental materials, and hence obtain a wrong impression of the data. Overall, statements like these undermined my trust in the authors’ intent to represent their statistical findings in an unbiased and accurate way.

The paper also evoked the impression in several places that the study introduces a novel task and/or novel computational model and/or novel research direction. I would recommend the authors elaborate more on related previous studies and ground their findings more closely in the existing literature.

Response: Thank you for the opportunity to clarify. We certainly did not intend to overstate our results or the novelty of our task. We have removed any reference to the word ‘novel’ in our manuscript. We have also carefully gone through the manuscript to ensure that our claims are as measured and balanced as possible and made edits where appropriate.

In the manuscript, we pointed readers to Table S4, but in fact did not comment on the strength of correlations. Our description of these correlations was only in our response to reviewers, where we wanted to explain our reasoning to you. We have now also added to the manuscript that the correlations between model parameters were ‘small to moderate’:

“Correlations between model parameters and reward and punishment task performance are shown in Supplementary materials, Table S3, and correlations between model parameters are shown in Table S4. Correlations between model parameters were small to moderate.”

Moreover, also on advice of the editor, we have included a new section in the discussion where we further discuss the limitations of our model:

“In addition, while our task was able to separate action initiation biases from learning processes, there were several other possible parameters that it was not practical or theoretically meaningful to assess using this design. For example, we did not include variable learning rates⁴³, choice stickiness parameters⁴⁴, or the role of forgetting⁴², which could be measured in future research using a suitable probabilistic learning task, possibly with drifting rewards to capture behavioural variability. Such a task may also help to separate correlations between model parameters more clearly, which were in some cases moderately correlated, although the differential association with age supports their distinction.”

R3.3. The following points of my initial review have not yet been addressed adequately or have:
Are younger participants excluded more often than older ones?
Could the authors provide a histogram of the number of participants excluded, binned in yearly increments?

Response: Thank you for the opportunity to clarify. We ran several additional analyses to confirm that there was no age bias in the number of excluded participants. In our previous response we detailed how we ran a Bayesian analysis that showed substantial evidence for the null (i.e., no age differences in excluded versus included participants). We added this information to the Supplementary materials:

“We also note that there was no significant age difference between the excluded participants and those included in the final sample (2-tailed t-test: $t_{(110.94)} = 0.25$, $p = .81$, $BF_{01} = 7.87$, substantial evidence for the null).”

We have now in addition provided a histogram for the number of participants included and excluded in yearly increments (see Figure S15). This further confirms that there was no bias in exclusions by age group.

Figure S15. Number of participants included in or excluded from the final sample by age. Exclusions were due to poor data quality. The proportion of exclusions was similar across the age range.

We also note that exclusions did not have any impact on our results or conclusions. In R1 we repeated all of our analyses with any excluded data included (see Table S2).

R3.4. Are 3 seconds enough for the youngest participants?

Could the authors provide a scatterplot of raw (i.e., not averaged) response times over trials, separately for each 1-year age bin?

Response: Thank you for this suggestion. With 80 trials and 700+ participants, this is not the clearest way to represent the data. However, as you can see below, 3 seconds are more than enough even for the youngest participants, where individual RTs are generally far from the maximum RT. This is most clearly reflected in the fact that the mean RT for the youngest participants was just 1050ms. 1050ms is much faster than the maximum 3000ms participants had to make a response. We have also added this additional figure to our Supplementary materials (Figure S14). We also include our additional analyses from R1 which also confirmed 3 seconds was enough time, and we have now added standard deviations where we report mean reaction times:

“We checked that 3000ms was sufficient for participants to make a response if they elected to do so. The mean reaction time was 1000ms (SD = 231.47) for the whole sample, and 1050ms (SD = 252.61) in the youngest participants (aged 9-12 years). Therefore, all participants were very clearly able to respond in the allotted time (see Supplementary materials, Figure S10 for additional analysis of reaction times).”

Figure S14. Raw reaction times for ‘go’ responses across stimulus repetitions, by age. Data points represent raw reaction times for stimulus repetitions 1-10 (x-axis), presented separately by age in years (labelled in panels above each sub-figure). For each stimulus repetition, there are eight trials (one per stimulus) for each of the 742 participants. Participants had up to 3000ms to respond.

R3.5. Was there an association between age and sex?

I do not understand how a t-test test can assess this question (“There was no association between age and sex in the final sample (2-tailed t-test: $t(488) = 1.00$, $p = .20$, $BF_{01} = 5.03$, substantial evidence in support of the null).”)

Could the authors provide a histogram that shows the number of male and female participants in each 1-year age bin?

Response: Apologies for any confusion that may have arisen from the words ‘association between age and sex’, which should have read ‘difference in age between the sexes’. The t-test assesses whether the average age differs between the two groups (males and females). It is therefore equivalent to other tests such as a biserial correlation, which could be interpreted as assessing an association. We have updated the wording and also provided a histogram showing the proportions of male and female participants in each age bin as requested, also included as a supplementary figure (Figure S16):

Figure S16. Number of male and female participants by age. The greater proportion of females across the age range reflects a deliberate sampling strategy in the wider dataset.

R3.6 It appears that the main difference between the reward and punishment conditions is that go responses stay high in the reward condition, whereas they drop in the punishment condition. Could the authors assess whether there is evidence for this hypothesis? E.g., is there a significant positive slope of go responses in the reward condition, excluding trial 1? (Table S2, rightmost columns seems to indicate this, but I find it surprising given the slopes look negative for all age groups in Fig. 6). The authors claim that “this age-related learning improvement was specific to learning from punishment outcomes” (i.e., no learning in reward trials). However, the computational model shows reward learning rates greater than 0 (i.e., significant learning in reward trials). This seems contradictory. I see two possible causes for this: 1) The behavioral regression model treats repetition number as a linear predictor, while the behavior is non-linear, implying that a quadratic term might fit better. 2) The computational model does not fit the data.

Response: To reiterate, we are not claiming that there is no reward learning. It is clear that reward learning does occur. However, unlike punishment, there is no *age-related change* in reward learning. With regards to the fit between the model and the data, please see our next response below.

R3.7 (Related to R3.1 and R3.6). Lastly, due to the qualitative difference in responding to repetition 1 compared to later repetitions, I am wondering if repetition 1 might be considered part of participants’ learning phase, and if it would clean up some results if the analyses were restricted to repetitions 2-10 only. This would likely improve the fits of the GLMs that contain repetition as a predictor because in the current data, the effect of repetition is highly non-linear, but it seems linear after trial 2. It might also help the computational model, which currently does not capture the quantitative aspects of human behavior. Authors response: “Thank you for this suggestion. We have now re-run all our modelling and behavioural analyses using only data from repetitions 2-10. We can confirm that this did not change the age-related patterns we observed in our main results. The results of this analysis are presented in Table S2 (see above)”

Could the authors confirm that they fitted the computational model to repetitions 2-10, and show the resulting simulated model behavior, model fits, and model parameters?

My central concern with the current study is that the model does not seem to capture human behavior adequately. As shown in the plot below from the supplemental material, one of the most noteworthy patterns in participant behavior is the initial increase in “P(go),” which is completely absent from the model behavior. Whereas the model thereafter shows a similar decline in go responses in the punishment condition to humans, it shows a steady increase of go responses in the reward condition, which is absent in the human data. (Indeed, the behavioral analyses show a lack of learning from reward, which directly contradicts the model.) This suggests that the model did not indeed capture human behavior.

“First, we have increased the number of simulated participants to match the number of participants in our sample, and used the median parameter values, at the request of a reviewer. The new simulated data more closely follows our model fitted data from participant behaviour.”

These changes did not address my initial concern.

Furthermore, simulated participants should be based on actual participants’ parameter values (i.e., one simulation per participant, using that participant’s parameter values), rather than the population median. Using the population median artificially smoothes the resulting simulations, and disregards existing variability and parameter correlations.

Could the authors confirm that they fitted the computational model to repetitions 2-10, and show the resulting simulated model behavior, model fits, and model parameters?

“Second, we have plotted the simulated and fitted data separately for the three different age groups and showed reasonable qualitative model fits across all three groups.”

This does not address my initial point.

The model fits are still not “reasonable”, in my opinion. The non-linear behavioral pattern is not captured. The stability in go responses and initial increase in both conditions are not captured.

“Third, we have repeated our model fitting and comparison procedure separately for the three age groups and again showed that the same model wins across all three age groups. “

This does not address my initial point.

“Fourth, we have repeated our behavioural analyses using only trials 2-10, and finally, we have repeated our model fitting and comparison procedure with trials 2-10. These analyses demonstrated exactly the same associations with age. “

Could the authors show the results of these analyses?

“Therefore, our model accurately captures behaviour. Additionally, we have reanalysed our parameter recovery and model identifiability with model 5, without the low recovery rho parameter, and shown recovery is good and stronger than our original reported results using model 6. Furthermore, we can still confirm exactly the same associations with age regardless of whether model 5 or 6 is selected as the winning model (see Table S2).“

This is true, however does not address my initial point.

“Together, all these new analyses robustly show that our winning model is a good and robust model of participant behaviour. “

I disagree with this statement. The model does not capture the major patterns of human behavior. To determine this issue, it would be instructive to see scatterplots of the generated and recovered values of each parameter (e.g., see Wilson & Collins, 2019). Could the authors provide these scatterplots? It would also be helpful to see a correlation matrix of all the model parameters, to see if parameters traded off with each other within the model. For example, learning rates and decision temperature parameters are known to trade off with each other, with important caveats for model interpretation.

“Response: We have added these correlations to Supplementary materials. While there were significant correlations between our model parameters, the effect sizes of the correlations are mostly quite small. No correlations are substantial enough to cause concern...”

I would not call correlations up to $r=0.58$ and $r=0.36$ “mostly quite small”; this statement seems misleading.

These patterns suggest that the models are misspecified and might have to be parameterized differently to adequately capture human behavior (e.g., Katahira, 2015, JoMP). The extremely large variability of parameter values between participants (despite hierarchical fitting) also supports the interpretation of a lack of model fit and potential model misspecification. In order to address this issue, the authors could try different parameterizations of their initial model (i.e., expand on the second Fig. 6)

There are several ways in which the model could be improved. First, additional parameters could help explain additional variance, and resolve issues of model misspecification (e.g., Katahira, 2015, JoMP).

Memory parameters (WM, episodic memory, forgetting, persistence)

Time-varying parameters (outcome-scaling learning rate; decaying go bias)

Uncertainty / curiosity (e.g., nogo would increase curiosity, which would trigger subsequent go response): I am wondering if the mechanism behind the increase in “go” responses from the first to the second iteration is caused by the realization that nogo actions do provide any information. Could the authors estimate an uncertainty parameter that captures how much participants want to learn about each stimulus, and positively contributes to selecting the go action? E.g., on each iteration that a nogo action is executed for a stimulus, its uncertainty could increase; and uncertainty and expected value could play together in determining action selection. The existing literature on the observe-or-bet task might be relevant here as well, as participants need to choose between the action of “observing” (i.e., seeing the outcome of an action, but without winning / losing points) or “betting” (i.e., Winning / losing points, but without observing the action)

Different kind of model (e.g., optimal Bayesian observer)

I was also wondering what the optimal strategy was for the current task, and whether there was a way to quantify how close participants were to this optimal strategy? Apparently, because feedback is deterministic, an optimal strategy would set both reward and punishment learning to the maximum values of 1. (Why are participants’ parameters so different?) And what is the optimal

setting of the go parameter? It seems like it should be 1 at the first iteration, so that the outcome of each stimulus can be observed once (and then memorized); after the first repetition, the go bias should then drop to 0, so that the correct action can be chosen consistently and without bias. If indeed, the optimal strategy requires an adjustment of the go bias over time, this might be a valid mechanism to integrate into the model. Also related to the go bias parameter, it seems like the main function of this parameter is to shift the balance between go/nogo toward more go responses. I was wondering if other options had been explored to capture this effect? For example, it might be more succinct to think of this shift as a shift of the entire softmax function.

“The implementation of the go biases was based on previous models (esp. Raab & Hartley, 2020) and thus we followed this literature rather than shifting the softmax function. We also tested a declining go bias (see our response to your comment below) but this did not outperform the constant go bias. We added details of this additional control model to our supplementary materials and note this in the main text:“

As far as I remember, the Raab & Hartley, 2020 task was different from the current paper because it was stochastic rather than deterministic? If so - and even if not - it is perfectly legitimate to try out different models, and I would encourage the authors to do so.

As for the declining go bias, I could not find any information about how the decline was implemented (e.g., linear decline? Quadratic decline? Exponential decay?). This choice could significantly impact the outcome. Given that the misfit of the current model to the data hinges on the fact that the model does not show non-linear trajectories, carefully trying out a parameterization that allows for non-linear behavior seems worthwhile.

Many studies have also reported that learning rates change over time (e.g., Jepma, 2020) or depending on previous trial outcomes. Have the authors explored this possibility in the current study?

“Response: We considered but rejected this option for the current study, due to concerns about excessive model complexity and because our task was deterministic rather than probabilistic.”

If a model does not fit the data, different options need to be tried. Model complexity need not necessarily increase if the newly added mechanism can replace others currently present in the model. And if a more complex model fits the data better, the additional complexity is not an issue. Hence, model complexity is not an argument to not try out more models.

Going beyond individual parameters, I was wondering if the structure of the model might have to be adapted to capture human behavior better. For example, the authors use a classic RL update when participants choose the “go” action, but no update is performed after the “nogo” action. In words, this model assumes that participants do not learn in “nogo” trials. I was wondering if the model space could be explored more to establish whether this assumption is true. To explore this idea, the authors could first determine whether participant behavior still changes even when participants choose the “nogo” action; if so, the model could capture behavioral changes, e.g., by including an update based on the outcome “0”.

“Response: We discussed and tested these different possible options at some length when designing our models. There were three main problems with learning after nogo trials. First, updating values after a no-go response did not make theoretical sense given that participants had to respond to get feedback; the task design explicitly assumes that responses are necessary for learning to occur. Second, at the beginning of the task the outcome values are not known, so we could not plausibly

interpret early non-responses as losses or gains. To get around this we would have to treat neutral outcomes as gains/losses only after the participant has received feedback on the object value, which means we would be making implicit assumptions about learning rate in coding the outcomes (i.e., that the participant remembered the value). Third, if we were to treat no-go on reward trials as loss and go on punishment trials as gain (throughout the whole task), there is a symmetrical effect so that while the expected values are inflated, the choice probabilities are the same as in our current coding.”

I would argue that NoGo actions could provide a different kind of information from Go actions. I would recommend the authors first do careful behavioral analyses to understand what effect NoGo actions have on behavior; after this understanding has been achieved, the model should be adapted to include the hypothesized underlying mechanism.

What cognitive processes underlie the behavior, and can these be tested behaviorally (in support of the computational model)? For example, how large is the increase in responding between the first and second repetition? Is it significant? Is the overall shape linear or nonlinear (e.g., fitted better by a quadratic function than linear)? (How) do rewards and punishments affect the behavior? Do participants stay with responses (e.g., go for stim1) after observing positive outcomes (e.g., reward after go) while switching responses (e.g., nogo for stim2) after observing negative outcomes (e.g., punishment after go)? (This would be the “The other option, to conduct the same analysis at the level of individual stimuli (e.g., so that the ‘previous trial’ was the previous presentation of the same stimulus rather than the previous trial in time)”)

Response: Thank you for these additional queries. They related to your main query **R3.1** and **R3.6** regarding how our model fits behaviour, which we have now improved. We have reproduced our response here:

- (1) “Parameters from the model meaningfully correlated with task accuracy in the expected directions (higher learning rates were related to higher task accuracy, and lower go biases related to higher task accuracy).
- (2) The main parameters of interest from the model (learning rates and go biases) showed meaningful associations with age regardless of whether other parameters were included or omitted.
- (3) Regardless of whether we selected model 6, model 7, or modelled only stimulus repetitions 2-10, the resulting parameters had identical associations with age, suggesting that they can indeed be interpreted.
- (4) We found that the same model provided the best explanation regardless of whether we modelled all stimulus repetitions or only stimulus repetitions 2-10.
- (5) The winning model showed good parameter recovery and model identifiability.
- (6) The winning model had the highest exceedance probability and one of the lowest BIC values.

However, we agree that the correspondence between model simulated data and participant data could be improved, although we note many studies where the simulations were comparable to ours (e.g. Guitart-Masip, 2011; Callaway et al., 2021), and, as you say, it is “impossible (and probably undesirable) to create a model that reproduces the to-be-modelled behavior identically in all its aspects”.

We have now used the winning model to simulate behaviour from stimulus repetitions 2-10, as suggested in your comment R3.6. Thank you for the suggestion. As expected, based on the pattern from our initial simulations, considering stimulus presentation 1 as part of the learning phase significantly improves the correspondence between simulated and actual behaviour (Figure 6 below). We have already shown that the winning model with stimulus repetitions 2-10 has exactly

the same associations with age as when we include all stimulus repetitions, and therefore this analysis further corroborates our choice of model that we selected for reasons 1-6 above.

We have now updated the manuscript to include this new analysis and figure, which we hope addresses your remaining concerns surrounding one aspect of model fit that could be improved. To reiterate, including only stimulus repetitions 2-10 does not change any of our main conclusions surrounding associations between age and model parameters, and the same model wins in the model space. The results of all analyses including stimulus repetitions 2-10 were previously shown in Table S2 and in Figure 7. In the revised manuscript, we have presented these results more fully; please see Figure 6, Figure S7, Figure S8, and Table S2. We have also reproduced Figure 6 and Table S2 again below. (Previously, this figure showed simulations separately by age category; these results have now been moved to Figure S6 in the supplement.)

The simulations using stimulus repetitions 2-10 only are now introduced as follows in the manuscript:

“To further confirm the accuracy of our model, we generated synthetic behavioural data for 742 ‘participants’ using the winning model and its median parameter values. As a control analysis, we also generated synthetic data using the version of the winning model with only stimulus repetitions 2-10 included (Figure 6).”

“In both the main 742-participant sample (with and without the first stimulation presentation) and the aged-based samples, the simulated responses fell within a similar range to the real responses. They followed a somewhat similar trajectory, particularly when omitting the first trial and considering it as part of the practice phase (Figure 6 and Figure S11). Importantly, parameter associations with age were identical whether we included all trials or only repetitions 2-10 (Table S2 and Figure S8).”

Figure 6. Simulated probability of ‘go’ response to reward and punishment stimuli across stimulus repetitions. (a) Simulated and real probability of ‘go’ responses on reward trials, with simulated data generated using the winning model and its median parameter values for the full sample (N = 742) across all 10 stimulus repetitions. (b) Simulated and real probability of ‘go’ responses on punishment trials for the full sample (N = 742) across all 10 stimulus repetitions. (c) Simulated and real probability of ‘go’ responses on reward trials for the full sample (N = 742) across stimulus repetitions 2-10 only (d) Simulated and real probability of ‘go’ responses on punishment trials for the full sample (N = 742) across stimulus repetitions 2-10 only. This Figure relates to Figure S11.

Table S2. Summary of additional analyses

	Main analyses	With model 6	With all eligible participants	With all eligible and outliers	With repetitions 2-10 only
Sample size					
N	742	742	832	832	742
Behavioural analysis: GLMM predicting no. correct responses from age					
Main effect stimulus repetition	OR = 1.19 [1.17, 1.21], z = 18.56, p < .001)	N/A	OR = 1.17 [1.15, 1.19], z = 19.01, p < .001	N/A	OR = 1.15 [1.13, 1.17], z = 13.87, p < .001
Age*repetition	OR = 1.02 [1.01, 1.04], z = 2.49, p = .01)	N/A	OR = 1.03 [1.01, 1.04], z = 2.97, p = .003	N/A	OR = 1.02 [1.00, 1.04], z = 2.06, p = .04
Main effect age	(OR = 1.08 [1.04, 1.11], z = 4.58, p < .001)	N/A	OR = 1.07 [1.04, 1.10], z = 4.92, p < .001	N/A	OR = 1.08 [1.04, 1.12], z = 4.48, p < .001
Age*repetition* valence	OR = 1.09 [1.05, 1.13], z = 4.65, p < .001)	N/A	OR = 1.10 [1.06, 1.14], z = 5.56, p < .001	N/A	OR = 1.07 [1.03, 1.12], z = 3.32, p < .001

BF ₀₁ for (no) age difference in reward learning	57.80	N/A	29.11	N/A	112.00
Modelling: model comparison					
Overall winning model	5	N/A (6)	5	5	5
Modelling: GLMMs predicting parameter values from age					
Main effect of age on punishment learning	$\beta = 0.10$ [0.05, 0.15], $z = 4.12$, $p < .001$	$\beta = 0.10$ [0.05, 0.15], $z = 4.26$, $p < .001$	$\beta = 0.11$ [0.07, 0.15], $z = 4.86$, $p < .001$	$\beta = 0.05$ [0.03, 0.07], $z = 5.40$, $p < .001$	$\beta = 0.12$ [0.07, 0.17], $z = 4.59$, $p < .001$
Main effect of age on action initiation bias	$\beta = -0.20$ [-0.28, -0.12], $z = -4.91$, $p < .001$	$\beta = -0.20$ [-0.28, -0.12], $z = -4.78$, $p < .001$	$\beta = -0.20$ [-0.27, -0.12], $z = -5.30$, $p < .001$	$\beta = -0.17$ [-0.23, -0.10], $z = -5.14$, $p < .001$	$\beta = -0.15$ [-0.23, -0.06], $z = -3.44$, $p < .001$
Main effect of age on reward learning rate	$\beta = 0.01$ [-0.05, 0.07], $z = 0.30$, $p = .77$	$\beta = 0.01$ [-0.06, 0.07], $z = 0.17$, $p = .86$	$\beta = 0.02$ [-0.03, 0.08], $z = 0.80$, $p = .42$	$\beta = 0.03$ [-0.01, 0.07], $z = 1.29$, $p = .20$	$\beta = 0.03$ [-0.05, 0.11], $t = 0.73$, $p = .46^*$
BF ₀₁ for (no) age difference in reward learning rate	13.60	12.62	12.62	11.57	20.00
Main effect of age on temperature parameter	$\beta = 0.002$ [-0.07, 0.07], $z = 0.08$, $p = 0.94$	$\beta = 0.002$ [-0.07, 0.08], $z = 0.06$, $p = 0.95$	$\beta = -0.03$ [-0.10, 0.03], $z = -0.96$, $p = 0.34$	$\beta = -0.03$ [-0.10, 0.04], $t = -0.85$, $p = 0.39^*$	$\beta = 0.06$ [-0.01, 0.14], $z = 1.70$, $p = .09$

*These statistics are from a standard linear model without site, due to non-convergence of the mixed (GLMM) model due to model complexity

We have also, in response to your comments, conducted the parameter recovery and model identifiability procedures for the models with stimulus repetitions 2-10 only, which are now included in the updated model performance figure in Supplementary materials:

Figure S7. Model comparison with stimulus repetitions 2-10 only. (a) Exceedance probability for the four computational models that comprised the model space when including data from repetitions 2-10 only. The winning model on this measure was the same as when all stimulus repetitions were included ($2a\beta b_c$ model). (b) ΔBIC_{int} , relative to the winning model ($2a\beta b_c$). (c) ΔLME , relative to the winning model ($2a\beta b_c$). Note model 3 is not included in the model space since there can be no initial action initiation bias when omitting the first trial. (d) Parameter recovery. The confusion matrix represents Spearman correlations between simulated and fitted (recovered) parameters. Parameters for the winning $2a\beta b_c$ model were recoverable. (e) Exceedance probability from the model identifiability procedure. The diagonal represents the probability of each model having the best fit to its own synthetic data. (f) Number of runs where each model was selected as the best fit for data generated by each model in the model identifiability procedure. The diagonal represents the number of runs in which each model was selected as the best fit for its own data. Model identifiability was reasonable although less strong than for the models using all 10 stimulus repetitions.

In addition, we have implemented the additional requests beyond our updated analysis of stimulus repetitions 2-10 below.

“As for the declining go bias, I could not find any information about how the decline was implemented (e.g., linear decline? Quadratic decline? Exponential decay?). This choice could significantly impact the outcome. Given that the misfit of the current model to the data hinges on the fact that the model does not show non-linear trajectories, carefully trying out a parameterization that allows for non-linear behavior seems worthwhile. “

Response: This model used a linear decline – we apologise for not stating this in our earlier revision. We have updated the Supplementary materials with this information:

“As a control analysis, we considered an additional model that accounted for the action initiation bias as declining over time, rather than a constant bias as in our winning model. The amount of decline with time was set as a linear free parameter.”

I was also wondering what the optimal strategy was for the current task, and whether there was a way to quantify how close participants were to this optimal strategy?

Response: Thank you for this query. The optimal strategy is to have a high learning rate and a lower action initiation bias, as accuracy on the task positively correlates with learning rate and negatively correlates with action initiation bias. This information was previously in the revised manuscript, and we have now further elaborated on how these correlations suggest what would be an optimal strategy:

“Overall task performance (proportion of correct responses) was positively correlated with reward learning rate (Spearman’s $r_{(696)} = 0.40 [0.33, 0.46]$, $p < .001$) and punishment learning rate (Spearman’s $r_{(696)} = 0.68 [0.63, 0.72]$, $p < .001$) and negatively correlated with action initiation bias (Spearman’s $r_{(696)} = -0.26 [-0.33, -0.19]$, $p < .001$). Temperature parameter values were also negatively correlated with task performance (Spearman’s $r_{(696)} = -0.39 [-0.45, -0.32]$, $p < .001$). These correlations demonstrate that the optimal strategy for this task is captured by higher learning rates for both reward and punishment, combined with a lower action initiation bias.”

It would be instructive to see scatterplots of the generated and recovered values of each parameter (e.g., see Wilson & Collins, 2019). Could the authors provide these scatterplots? It would also be helpful to see a correlation matrix of all the model parameters, to see if parameters traded off with each other within the model. For example, learning rates and decision temperature parameters are known to trade off with each other, with important caveats for model interpretation.

Response: Thank you for this suggestion. We previously included a figure of our parameter recovery correlation matrix and a table showing the correlations between model parameters (see Figure 3 and Table S4). We have now in addition included the scatterplots as a supplementary figure:

Figure S9. Correlations between fitted and recovered values for model parameters from the winning model. All correlations were significant (reward learning rate $r = .71, p < .001$; punishment learning rate $r = .80, p < .001$; action initiation bias $r = .92, p < .001$; temperature parameter $r = .49, p < .001$). For presentation purposes, temperature parameter values are log-transformed. This Figure relates to Figure 3c in the main text.

“Second, we have plotted the simulated and fitted data separately for the three different age groups and showed reasonable qualitative model fits across all three groups.”

The model fits are still not “reasonable”, in my opinion. The non-linear behavioral pattern is not captured. The stability in go responses and initial increase in both conditions are not captured.

“Third, we have repeated our model fitting and comparison procedure separately for the three age groups and again showed that the same model wins across all three age groups.”
This does not address my initial point.

“Fourth, we have repeated our behavioural analyses using only trials 2-10, and finally, we have repeated our model fitting and comparison procedure with trials 2-10. These analyses demonstrated exactly the same associations with age.”

Could the authors show the results of these analyses?

Response: These results were in Table S2. As highlighted in response to your query **R3.1** and **R3.6**, we have now updated our model to include only stimulus repetitions 2-10 as an additional control. These analyses show that the same model wins as when including all stimulus repetitions 1-10, and the associations with age are identical. We have now added additional supplementary figures visualising the associations between model parameters and age, and we have pasted Table S2 again below (see the final column):

Figure S8. Age differences in action initiation bias, punishment learning, and reward learning from the winning model with stimulus 2-10 repetitions only. (a) Punishment learning rate across three age groups. Punishment learning rates increased linearly with age. **(b)** Action initiation bias across three age groups. Action initiation biases declined linearly with age. **(c)** Reward learning rates across three age groups. Reward learning rates remained stable with age. Points and errors bars represent means and 95% confidence intervals of the means for each group, with raw data represented by smaller points. Division into age groups is for presentation purposes only; age was treated as a continuous variable in all analyses. The relevant statistical analyses are provided in Table S2.

Table S2. Summary of additional analyses

	Main analyses	With model 6	With all eligible participants	With all eligible and outliers	With repetitions 2-10 only
Sample size					
N	742	742	832	832	742
Behavioural analysis: GLMM predicting no. correct responses from age					
Main effect stimulus repetition	OR = 1.19 [1.17, 1.21], $z = 18.56$, $p < .001$)	N/A	OR = 1.17 [1.15, 1.19], $z = 19.01$, $p < .001$	N/A	OR = 1.15 [1.13, 1.17], $z = 13.87$, $p < .001$
Age*repetition	OR = 1.02 [1.01, 1.04], $z = 2.49$, $p = .01$)	N/A	OR = 1.03 [1.01, 1.04], $z = 2.97$, $p = .003$	N/A	OR = 1.02 [1.00, 1.04], $z = 2.06$, $p = .04$
Main effect age	(OR = 1.08 [1.04, 1.11], $z = 4.58$, $p < .001$)	N/A	OR = 1.07 [1.04, 1.10], $z = 4.92$, $p < .001$	N/A	OR = 1.08 [1.04, 1.12], $z = 4.48$, $p < .001$
Age*repetition*valence	OR = 1.09 [1.05, 1.13], $z = 4.65$, $p < .001$)	N/A	OR = 1.10 [1.06, 1.14], $z = 5.56$, $p < .001$	N/A	OR = 1.07 [1.03, 1.12], $z = 3.32$, $p < .001$
BF ₀₁ for (no) age difference in reward learning	57.80	N/A	29.11	N/A	112.00
Modelling: model comparison					
Overall winning model	5	N/A (6)	5	5	5
Modelling: GLMMs predicting parameter values from age					
Main effect of age on punishment learning	$\beta = 0.10$ [0.05, 0.15], $z = 4.12$, $p < .001$	$\beta = 0.10$ [0.05, 0.15], $z = 4.26$, $p < .001$	$\beta = 0.11$ [0.07, 0.15], $z = 4.86$, $p < .001$	$\beta = 0.05$ [0.03, 0.07], $z = 5.40$, $p < .001$	$\beta = 0.12$ [0.07, 0.17], $z = 4.59$, $p < .001$
Main effect of age on action initiation bias	$\beta = -0.20$ [-0.28, -0.12], $z = -4.91$, $p < .001$	$\beta = -0.20$ [-0.28, -0.12], $z = -4.78$, $p < .001$	$\beta = -0.20$ [-0.27, -0.12], $z = -5.30$, $p < .001$	$\beta = -0.17$ [-0.23, -0.10], $z = -5.14$, $p < .001$	$\beta = -0.15$ [-0.23, -0.06], $z = -3.44$, $p < .001$
Main effect of age on reward learning rate	$\beta = 0.01$ [-0.05, 0.07], $z = 0.30$, $p = .77$	$\beta = 0.01$ [-0.06, 0.07], $z = 0.17$, $p = .86$	$\beta = 0.02$ [-0.03, 0.08], $z = 0.80$, $p = .42$	$\beta = 0.03$ [-0.01, 0.07], $z = 1.29$, $p = .20$	$\beta = 0.03$ [-0.05, 0.11], $t = 0.73$, $p = .46^*$
BF ₀₁ for (no) age difference in reward learning rate	13.60	12.62	12.62	11.57	20.00
Main effect of age on temperature parameter	$\beta = 0.002$ [-0.07, 0.07], $z = 0.08$, $p = 0.94$	$\beta = 0.002$ [-0.07, 0.08], $z = 0.06$, $p = 0.95$	$\beta = -0.03$ [-0.10, 0.03], $z = -0.96$, $p = 0.34$	$\beta = -0.03$ [-0.10, 0.04], $t = -0.85$, $p = 0.39^*$	$\beta = 0.06$ [-0.01, 0.14], $z = 1.70$, $p = .09$

*These statistics are from a standard linear model without site, due to non-convergence of the mixed (GLMM) model due to model complexity

REVIEWERS' COMMENTS

Reviewer #1 (Remarks to the Author):

I believe the authors have now addressed all queries of the reviewer adequately. I do believe that simply focusing on trails 2-10 is still somewhat of a hack, given that you would like to explain and replicate the full data, but in the sense of the message of the papers and the interpretation of the parameters of interest (and those that show developmental differences) this is a perfectly fine thing to do. I also want to state that the authors are clearly expert enough in computational modeling, and this paper shows high quality work in that area. We should always be aware of the sometimes qualitative nature of this type of work, and that the best fitting model, may not accurately describe the underlying mechanisms. I would love to see this paper in publication, and my apologies for the delays that I have caused in that process.

Reviewer #3 (Remarks to the Author):

I appreciate the care with which the authors have responded to my (many) comments on the paper. The additional analyses confirm that there are no reasons to believe that any of the confounds I mentioned could have explained the results, in support of the authors' initial conclusions. For some analyses that I suggested and that the authors decided not to perform, they have provided a clear rationale for not doing so. The paper has been updated to reflect all my key points.

At this point, I also want to thank the authors for their patience in this process, and apologize for the lack of care in phrasing some of the points in my initial review, which I noticed upon rereading.

Response to Reviewers' comments R3: "Action initiation and punishment learning differ from childhood to adolescence while reward learning remains stable"

REVIEWERS' COMMENTS

Reviewer #1 (Remarks to the Author):

I believe the authors have now addressed all queries of the reviewer adequately. I do believe that simply focusing on trails 2-10 is still somewhat of a hack, given that you would like to explain and replicate the full data, but in the sense of the message of the papers and the interpretation of the parameters of interest (and those that show developmental differences) this is a perfectly fine thing to do. I also want to state that the authors are clearly expert enough in computational modeling, and this paper shows high quality work in that area. We should always be aware of the the sometimes qualitative nature of this type of work, and that the best fitting model, may not accurately describe the underlying mechanisms. I would love to see this paper in publication, and my apologies for the delays that I have caused in that process.

Response: Thanks so much again for all your helpful feedback and time spent improving our manuscript. We are also very grateful for your acknowledgment of our expertise and support for publication of the manuscript.

Reviewer #3 (Remarks to the Author):

I appreciate the care with which the authors have responded to my (many) comments on the paper. The additional analyses confirm that there are no reasons to believe that any of the confounds I mentioned could have explained the results, in support of the authors' initial conclusions. For some analyses that I suggested and that the authors decided not to perform, they have provided a clear rationale for not doing so. The paper has been updated to reflect all my key points.

At this point, I also want to thank the authors for their patience in this process, and apologize for the lack of care in phrasing some of the points in my initial review, which I noticed upon rereading.

Response: Thank you for all your helpful feedback and time spent improving our manuscript. We are delighted that you are satisfied with our response to your additional queries, and are grateful for your acknowledgment of the care and patience we put into our response.